# Analysis of Error Feedback in Compressed Federated Non-Convex Optimization

## Abstract

Communication cost between the clients and the central server could be a bottleneck in real-world Federated Learning (FL) systems. In classical distributed learning, the method of Error Feedback (EF) has been a popular technique to remedy the downsides of biased gradient compression, but literature on applying EF to FL is still very limited. In this work, we propose a compressed FL scheme equipped with error feedback, named Fed-EF, with two variants depending on the global optimizer. We provide theoretical analysis showing that Fed-EF matches the convergence rate of the full-precision FL counterparts in non-convex optimization under data heterogeneity. Moreover, we initiate the first analysis of EF under partial client participation, which is an important scenario in FL, and demonstrate that the convergence rate of Fed-EF exhibits an extra slow down factor due to the "stale error compensation" effect. Experiments are conducted to validate the efficacy of Fed-EF in practical FL tasks and justify our theoretical findings.

## 1 Introduction

Federated Learning (FL) has seen numerous applications in, e.g., computer vision, language processing, public health, Internet of Things (IoT) [19; 44; 62; 39; 49; 29; 25]. A centralized FL system includes multiple clients each with local data, and one central server that coordinates the training process. The goal of FL is for $n$ clients to collaboratively find a global model, parameterized by $\theta$, such that

$$\theta^* = \arg\min_{\theta \in \mathbb{R}^d} f(\theta) := \arg\min_{\theta \in \mathbb{R}^d} \frac{1}{n} \sum_{i=1}^{n} f_i(\theta), \tag{1}$$

where $f_i(\theta) := \mathbb{E}_{D \sim \mathcal{D}_i}\big[F_i(\theta; D)\big]$ is a non-convex loss function for the $i$-th client w.r.t. the local data distribution $\mathcal{D}_i$. Taking standard local SGD [42; 53] as an example, in each training round, the server first broadcasts the model to the clients. Then, each client trains the model based on the local data, after which the updated local models are transmitted back to the server and aggregated. The number of clients, $n$, can be either tens/hundreds (*cross-silo* FL [41; 21], e.g., clients are companies) or as large as millions (*cross-device* FL [26; 29], e.g., clients are personal devices). There are two primary benefits of FL: (i) the clients train the model simultaneously, which is efficient in terms of computational resources; (ii) each client's data are kept local throughout training and never transmitted to other parties, which promotes data privacy. However, the efficiency and broad application scenarios also brings challenges for FL method design:

- **Communication cost:** In most FL algorithms, clients are allowed to conduct multiple training steps (e.g., local SGD updates) in each round. Though this has reduced the communication frequency, the one-time communication cost is still a challenge in real-world FL systems with limited bandwidth, e.g., portable devices at the wireless network edges [5; 61; 29].

- **Data Heterogeneity:** Unlike in the classical distributed training, the local data distribution in FL ($\mathcal{D}_i$ in (1)) can be different (non-iid), reflecting many real-world scenarios where the local data held by different clients (e.g., app/website users) are highly personalized. When multiple local training steps are taken, the local models could become "biased" towards minimizing the local losses, instead of the global loss. This data heterogeneity may hinder the global model to converge to a good solution [34; 67; 33].

- **Partial participation (PP):** Another practical issue, especially for cross-device FL, is the partial participation (PP) where the clients do not join training consistently, e.g., due to

unstable connection or user change. That is, only a fraction of clients are involved in each FL training round, which may also slow down the convergence of the global model [10; 12].

**FL under compression.** In order to overcome the main challenge of communication bottleneck, several works have considered federated learning with compressed message passing. Examples include FedPaQ [47], FedCOM [18] and FedZip [40]. All these algorithms are built upon directly compressing model updates communicated from clients to server. In particular, [47; 18] proposed to use unbiased stochastic compressors such as stochastic quantization [3] and sparsification [57], which showed that with considerable communication saving, applying unbiased compression in FL could approach the learning performance of un-compressed FL algorithms. However, unbiased (stochastic) compressors typically require additional computation (sampling) which is less efficient in real-world large training systems. Biased gradients/compressors are also common in many applications [2].

**Error feedback (EF) for distributed training.** One simpler and popular type of compressor is the deterministic compressor, including fixed quantization [13], TopK sparsification [54; 52; 35], SignSGD [50; 7; 8], etc., which belong to biased compression operators. In classical distributed learning literature, it has been shown that directly updating with the biased gradients may slow down the convergence or even lead to divergence [28; 2]. A popular remedy is the so-called *error feedback (EF)* strategy [54]: in each iteration, the local worker sends a compressed gradient to the server and records the local compression error, which is subsequently used to adjust the gradient computed in next iteration, conceptually "correcting the bias" due to compression. It is known that biased gradient compression with EF can achieve same convergence rate as the full-precision counterparts [28; 35].

**Our contributions.** Despite the rich literature on EF in classical distributed training, it has not been well explored in the context of federated learning. In this paper, we provide a thorough analysis of EF in FL. In particular, the three key features of FL, local steps, data heterogeneity and partial participation, pose interesting questions regarding the performance of EF in federated learning: *(i) Can EF still achieve the same convergence rate as full-precision FL algorithms, possibly with highly non-iid local data distribution? (ii) How does partial participation change the situation and the results?* We present new algorithm and results to address these questions:

- We study a FL framework with biased compression and error feedback, called Fed-EF, with two variants (Fed-EF-SGD and Fed-EF-AMS) depending on the global optimizer (SGD and adaptive AMSGrad [46], respectively). Under data heterogeneity, Fed-EF has asymptotic convergence rate $\mathcal{O}(\frac{1}{\sqrt{TKn}})$ where $T$ is the number of communication rounds, $K$ is the number of local training steps and $n$ is the number of clients. Our new analysis matches the convergence rate of full-precision FL counterparts, improving the previous convergence result [6] on error compensated FL (see detailed comparisons in Section 3). Moreover, Fed-EF-AMS is the first compressed adaptive FL algorithm in literature.

- Partial participation (PP) has not been considered for standard error feedback in distributed learning. We initiate new analysis of Fed-EF in this setting, with local steps and non-iid data all considered at the same time. We prove that under PP, Fed-EF exhibits a slow down of factor $\sqrt{n/m}$ compared with the best full-precision rate, where $m$ is the number of active clients per round, which is caused mainly by the *"delayed error compensation"* effect.

- Experiments are conducted to illustrate the effectiveness of the proposed methods, where we show that Fed-EF matches the performance of full-precision FL with significant reduction in communication, and compares favorably against the algorithms using unbiased compression without error feedback. Numerical examples are also provided to justify our theory.

## 2 BACKGROUND AND RELATED WORK

**Distributed SGD with compressed gradients.** In distributed SGD training systems, extensive works have considered compression applied to the communicated gradients. Unbiased stochastic compressors include stochastic rounding and QSGD [3; 66; 58; 38], and magnitude based random sparsification [57]. The works [50; 7; 8; 28; 24] analyzed communication compression using only the sign (1-bit) information of the gradients. Unbiased compressors can be combined with variance reduction techniques for acceleration, e.g., [17]. On the other hand, examples of popular biased compressors include TopK [37; 54; 52], which only transmits gradient coordinates with largest magnitudes, and fixed (or learned) quantization [13; 66]. See [9] for a summary of more biased

compressors. There exist other compression schemes, such as vector quantization [64; 40], low-rank approximation [56] and sketching [22], which will not be the focus of this paper.

**Error Feedback (EF) for biased compression.** It has been shown that directly implementing biased compression in distributed SGD leads to worse convergence and generalisation [2; 9]. Error feedback (EF), as proposed in [54], can fix this issue [28]. With EF, distributed SGD under biased compression can match the convergence rate of the full-precision distributed SGD, e.g., also achieving linear speedup ($\mathcal{O}(1/\sqrt{Tn})$) w.r.t. the number of workers $n$ in distributed SGD [23; 51; 4; 68; 55]. Recently, a variant of EF, called EF21, was proposed [48], which is different from EF in algorithm design. [15] applied EF21 to FL under several settings. Our work is different, where we analyze the standard EF in FL, design new algorithms and derive faster convergence rates with different analysis.

Finally, in federated learning, as mentioned earlier, several works have applied compression to the client-to-server communication, e.g., [47; 40; 18]. Among the limited related literature on adopting EF to FL, the most relevant method is QSparse-local-SGD [6] which is a special case of the Fed-EF framework studied in this paper. In Section 3 and Section 4, we will compare our proposed Fed-EF framework with these related methods in terms of both algorithm and theory. Our algorithm also exploits adaptive gradient method AMSGrad [46], which has been applied to distributed and federated learning [45; 35]. See, for instance, [14; 65; 30] for the series of works on adaptive gradient methods.

## 3 FED-EF: COMPRESSED FEDERATED LEARNING WITH ERROR FEEDBACK

In this paper, we consider deterministic compressors which are simple and computational efficient. Throughout the paper, $[n]$ will denote the integer set $\{1, ..., n\}$.

**Definition 1** ($q_{\mathcal{C}}$-deviate compressor). *The $q_{\mathcal{C}}$-deviate compressor $\mathcal{C} : \mathbb{R}^d \mapsto \mathbb{R}^d$ is defined such that for $\forall x \in \mathbb{R}^d$, $\exists\, 0 \le q_{\mathcal{C}} < 1$ s.t. $\|\mathcal{C}(x) - x\|^2 \le q_{\mathcal{C}}^2 \|x\|^2$. In particular, two examples are [54; 68]:*

- *Let $\mathcal{S} = \{i \in [d] : |x_i| \ge t\}$ where $t$ is the $(1 - k)$-quantile of $|x_i|$, $i \in [d]$. The **TopK** compressor with compression rate $k$ is defined as $\mathcal{C}(x)_i = x_i$, if $i \in \mathcal{S}$; $\mathcal{C}(x)_i = 0$ otherwise.*

- *Divide $[d]$ into $M$ groups (e.g., neural network layers) with index sets $\mathcal{I}_i$, $i = 1, ..., M$, and $d_i := |\mathcal{I}_i|$. The **(Grouped) Sign** compressor is defined as $\mathcal{C}(x) = \left[ \frac{\|x_{\mathcal{I}_1}\|_1}{d_1} sign(x_{\mathcal{I}_1}), ..., \frac{\|x_{\mathcal{I}_M}\|_1}{d_M} sign(x_{\mathcal{I}_M}) \right]$, with $x_{\mathcal{I}_i}$ the sub-vector of $x$ at indices $\mathcal{I}_i$.*

Larger $q_{\mathcal{C}}$ indicates heavier compression, and $q_{\mathcal{C}} = 0$ implies no compression, i.e. $\mathcal{C}(x) = x$. Additionally, these two compression operations can be combined to derive the so-called "**heavy-Sign**", where we first apply **TopK** and then **Sign**. This strategy is also $q$-deviate (see Appendix A for details) and will also be tested in our experiments in Section 5.

Can we simply use biased compressors in communication-efficient FL? As an example, in Figure 1, we report the test accuracy of a Multi-Layer Perceptron (MLP) trained on MNIST in non-iid FL environment (see Section 5 for more description), of Fed-SGD [53] with full communication v.s. **Sign** compression. We observe a catastrophic performance loss of using biased compression directly.

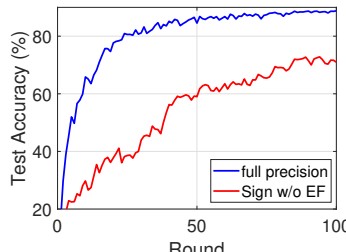

Figure 1: MLP trained by Fed-SGD on MNIST: full-precision vs. **Sign** compression, $n = 200$ non-iid clients.

**Fed-EF algorithm.** To resolve this problem, error feedback (EF), which is a popular tool in distributed training, can be adapted to federated learning. In Algorithm 1, we present a FL framework with biased compression, named Fed-EF, whose main steps are summarized below. In round $t$: 1) The server broadcast the global model $\theta_t$ to all clients (line 5); 2) The $i$-th client performs $K$ steps of local SGD updates to get local model $\theta_{t,i}^{(K)}$, compute the compressed *local model update* $\tilde{\Delta}_{t,i}$, updates the local error accumulator $e_{t,i}$, and sends the compressed $\tilde{\Delta}_{t,i}$ back to the server (line 6-12); 3) The server receives $\tilde{\Delta}_{t,i}$, $i \in [n]$ from all clients, takes the average, and perform a global model update using the averaged compressed local model updates (line 15-19).

Depending on the global model optimizer, we propose two variants: Fed-EF-SGD (green) which applies SGD global updates, and Fed-EF-AMS (blue), whose global optimizer is AMSGrad [46].

---

**Algorithm 1** Compressed Federated Learning with Error Feedback (Fed-EF)

---

1: **Input**: learning rates $\eta$, $\eta_l$, hyper-parameters $\beta_1, \beta_2, \epsilon$
2: **Initialize**: central server parameter $\theta_1 \in \mathbb{R}^d \subseteq \mathbb{R}^d$; $e_{1,i} = \mathbf{0}$ the accumulator for each worker; $m_0 = \mathbf{0}, v_0 = \mathbf{0}, \hat{v}_0 = \mathbf{0}$
3: **for** $t = 1, \ldots, T$ **do**
4:     **parallel for worker** $i \in [n]$ **do**:
5:         Receive model parameter $\theta_t$ from central server, set $\theta_{t,i}^{(1)} = \theta_t$
6:         **for** $k = 1, \ldots, K$ **do**
7:             Compute stochastic gradient $g_{t,i}^{(k)}$ at $\theta_{t,i}^{(k)}$
8:             Local update $\theta_{t,i}^{(k+1)} = \theta_{t,i}^{(k)} - \eta_l g_{t,i}^{(k)}$
9:         **end for**
10:         Compute the local model update $\Delta_{t,i} = \theta_{t,i}^{(K+1)} - \theta_t$
11:         Send compressed adjusted local model update $\widetilde{\Delta}_{t,i} = \mathcal{C}(\Delta_{t,i} + e_{t,i})$ to central server
12:         Update the error $e_{t+1,i} = e_{t,i} + \Delta_{t,i} - \widetilde{\Delta}_{t,i}$
13:     **end parallel**
14:     **Central server do:**
15:     Global aggregation $\overline{\widetilde{\Delta}}_t = \frac{1}{n} \sum_{i=1}^{n} \widetilde{\Delta}_{t,i}$
16:     Update the global model $\theta_{t+1} = \theta_t - \eta \overline{\widetilde{\Delta}}_t$                               $\triangleright$ Fed-EF-SGD
17:     $m_t = \beta_1 m_{t-1} + (1 - \beta_1)\overline{\widetilde{\Delta}}_t$                             $\triangleright$ Fed-EF-AMS
18:     $v_t = \beta_2 v_{t-1} + (1 - \beta_2)\overline{\widetilde{\Delta}}_t^2, \quad \hat{v}_t = \max(v_t, \hat{v}_{t-1})$
19:     Update the global model $\theta_{t+1} = \theta_t - \eta \frac{m_t}{\sqrt{\hat{v}_t} + \epsilon}$
20: **end for**

---

In Fed-EF-AMS, by the nature of adaptive gradient methods, we incorporate momentum ($m_t$) with different implicit dimension-wise learning rates $\eta/\hat{v}_t$. Additionally, for conciseness, the presented algorithm employs one-way compression (clients-to-server). In Appendix D, we also provide a two-way compressed Fed-EF framework and demonstrate that adding the server-to-clients compression would not affect the convergence rates.

**Comparison with prior work.** Compared with EF approaches in the classical distributed training, e.g., [54; 28; 68; 38; 16; 35], our algorithm allows local steps (more communication-efficient) and uses two-side learning rates. When $\eta \equiv 1$, the Fed-EF-SGD method reduces to QSparse-local-SGD [6]. In Section 4, we will demonstrate how the two-side learning rate schedule improves the convergence analysis of the one-side learning rate approach [6]. On the other hand, several recent works considered compressed FL using unbiased stochastic compressors (all of which use SGD as the global optimizer). FedPaQ [47] applied stochastic quantization without error feedback to local SGD, which is improved by [18] using a gradient tracking trick that, however, requires communicating an extra vector from server to clients, which is less efficient than Fed-EF. [40] provided an empirical study on directly compressing the local updates using various compressors in Fed-SGD, while we use EF to compensate for the bias. [43] proposed FedLin, which only uses compression for synchronizing a local memory term but still requires transmitting full-precision updates. Finally, to our knowledge, Fed-EF-AMS is the first compressed adaptive FL method in literature.

## 4 THEORETICAL RESULTS

**Assumption 1** (Smoothness). *For $\forall i \in [n]$, $f_i$ is $L$-smooth: $\|\nabla f_i(x) - \nabla f_i(y)\| \leq L \|x - y\|$.*

**Assumption 2** (Bounded variance). *For $\forall t \in [T]$, $\forall i \in [n]$, $\forall k \in [K]$: (i) the stochastic gradient is unbiased: $\mathbb{E}[g_{t,i}^{(k)}] = \nabla f_i(\theta_{t,i}^{(k)})$; (ii) the **local variance** is bounded: $\mathbb{E}[\|g_{t,i}^{(k)} - \nabla f_i(\theta_{t,i}^{(k)})\|^2] < \sigma^2$; (iii) the **global variance** is bounded: $\frac{1}{n} \sum_{i=1}^{n} \|\nabla f_i(\theta_t) - \nabla f(\theta_t)\|^2 \leq \sigma_g^2$.*

Both assumptions are standard in the convergence analysis of stochastic gradient methods. The global variance bound $\sigma_g^2$ in Assumption 2 characterizes the difference among local objective functions, which, is mainly caused by different local data distribution $\mathcal{X}_i$ in (1), i.e., data heterogeneity.

**Assumption 3** (Compression discrepancy). *There exists some $q_{\mathcal{A}} < 1$ such that $\mathbb{E}\big[\|\frac{1}{n}\sum_{i=1}^n \mathcal{C}(\Delta_{t,i} + e_{t,i}) - \frac{1}{n}\sum_{i=1}^n (\Delta_{t,i} + e_{t,i})\|^2\big] \leq q_{\mathcal{A}}^2 \mathbb{E}\big[\|\frac{1}{n}\sum_{i=1}^n (\Delta_{t,i} + e_{t,i})\|^2\big]$ in every round $t \in [T]$.*

In Assumption 3, if we replace "the average of compression", $\frac{1}{n}\sum_{i=1}^n \mathcal{C}(\Delta_{t,i} + e_{t,i})$, by "the compression of average", $\mathcal{C}(\frac{1}{n}\sum_{i=1}^n (\Delta_{t,i} + e_{t,i}))$, the statement immediately holds by Definition 1 with $q_{\mathcal{A}} = q_{\mathcal{C}}$. Thus, Assumption 3 basically says that the above two terms stay close during training. This is a common assumption in related work on compressed distributed learning, for example, a similar assumption is used in [4] analyzing sparsified SGD. In [18], for unbiased compression without EF, a similar condition is also assumed with an absolute bound. In Appendix B, we provide more discussion and empirical justification to validate this analytical assumption in practice.

### 4.1 CONVERGENCE OF FED-EF: LINEAR SPEEDUP UNDER DATA HETEROGENEITY

**Theorem 1** (Fed-EF-SGD). *Let $\theta^* = \arg\min f(\theta)$, and denote $q = \max\{q_{\mathcal{C}}, q_{\mathcal{A}}\}$, $C_1 := 2 + \frac{4q^2}{(1-q^2)^2}$. Under Assumptions 1 to 3, when $\eta_l \leq \frac{1}{2KL\cdot\max\{4, \eta(C_1+1)\}}$, the squared gradient norm of Fed-EF-SGD iterates in Algorithm 1 can be bounded by*

$$\frac{1}{T}\sum_{t=1}^T \mathbb{E}\big[\|\nabla f(\theta_t)\|^2\big] \lesssim \frac{f(\theta_1) - f(\theta^*)}{\eta\eta_l TK} + \frac{2\eta\eta_l C_1 L}{n}\sigma^2 + 10\eta\eta_l^3 C_1 K^2 L^3(\sigma^2 + 6K\sigma_g^2).$$

In Theorem 1, the LHS is the expected squared gradient norm at a uniformly chosen global model from $t = 1, ..., T$, which is a standard measure of convergence in non-convex optimization (i.e., "norm convergence"). The first term is dependent on the initialization, the second term $\sigma^2$ comes from the local stochastic variance, and the last term represents the influence of data heterogeneity. In general, we see that larger $q$ (i.e., higher compression) would slow down the convergence.

In our analysis of Fed-EF-AMS, we will make the following additional assumption of bounded stochastic gradients, which is common in the convergence analysis of adaptive methods, e.g., [46; 69; 11; 35]. Note that this assumption is only used for Fed-EF-AMS, but not for Fed-EF-SGD.

**Assumption 4.** *(Bounded gradients) It holds that $\|g_{t,i}^{(k)}\| \leq G$, $\forall t > 0$, $\forall i \in [n]$, $\forall k \in [K]$.*

We provide the first convergence analysis of compressed adaptive FL method as below.

**Theorem 2** (Fed-EF-AMS). *With same notations as in Theorem 1, let $C_1 := \frac{\beta_1}{1-\beta_1} + \frac{2q}{1-q^2}$. Under Assumptions 1 to 4, if the learning rates satisfy $\eta_l \leq \frac{\sqrt{\epsilon}}{8KL}\min\big\{\frac{1}{\sqrt{\epsilon}}, \frac{2(1-q^2)L}{(1+q^2)^{1.5}G}, \frac{1}{\max\{16, 32C_1^2\}\eta}, \frac{1}{3\eta^{1/3}}\big\}$, the Fed-EF-AMS iterates in Algorithm 1 satisfy*

$$\frac{1}{T}\sum_{t=1}^T \mathbb{E}\big[\|\nabla f(\theta_t)\|^2\big] \lesssim \frac{f(\theta_1) - f(\theta^*)}{\eta\eta_l TK} + \Big[\frac{5\eta_l^2 KL^2}{2\sqrt{\epsilon}} + \frac{\eta\eta_l^3(30 + 20C_1^2)K^2L^3}{\epsilon}\Big](\sigma^2 + 6K\sigma_g^2)$$

$$+ \frac{\eta\eta_l L(6 + 4C_1^2)}{n\epsilon}\sigma^2 + \frac{(C_1+1)G^2 d}{T\sqrt{\epsilon}} + \frac{3\eta\eta_l C_1^2 LKG^2 d}{T\epsilon}.$$

With some properly chosen learning rates, we have the following simplified results.

**Corollary 1** (Fed-EF, specific learning rates). *Suppose the conditions in Theorem 1 and Theorem 2 are satisfied respectively. Choosing $\eta_l = \Theta(\frac{1}{K\sqrt{T}})$ and $\eta = \Theta(\sqrt{Kn})$, Fed-EF-SGD satisfy*

$$\frac{1}{T}\sum_{t=1}^T \mathbb{E}\big[\|\nabla f(\theta_t)\|^2\big] = \mathcal{O}\Big(\frac{f(\theta_1) - f(\theta^*)}{\sqrt{TKn}} + \frac{1}{\sqrt{TKn}}\sigma^2 + \frac{\sqrt{n}}{T^{3/2}\sqrt{K}}(\sigma^2 + K\sigma_g^2)\Big),$$

*and for Fed-EF-AMS, it holds that*

$$\frac{1}{T}\sum_{t=1}^T \mathbb{E}\big[\|\nabla f(\theta_t)\|^2\big] = \mathcal{O}\Big(\frac{f(\theta_1) - f(\theta^*)}{\sqrt{TKn}} + \frac{1}{\sqrt{TKn}}\sigma^2 + (\frac{1}{TK} + \frac{\sqrt{n}}{T^{3/2}\sqrt{K}})(\sigma^2 + K\sigma_g^2)\Big).$$

**Discussion.** From Corollary 1, we see that when $T \geq K$, Fed-EF-AMS and Fed-EF-SGD have the same rate of convergence asymptotically. Therefore, our following discussion applies to the general Fed-EF scheme with both variants. In Corollary 1, when $T \geq Kn$, the global variance term $\sigma_g^2$ vanishes and the convergence rate becomes $\mathcal{O}(1/\sqrt{TKn})$. Thus, the proposed Fed-EF enjoys linear speedup w.r.t. the number of clients $n$, i.e., it reaches a $\delta$-stationary point (i.e., $\frac{1}{T}\sum_{t=1}^{T}\mathbb{E}\big[\|\nabla f(\theta_t)\|^2\big] \leq \delta$) as long as $TK = \Theta(1/n\delta^2)$, which matches the recent results of the full-precision counterparts [60; 45] ([45] only analyzed the special case $\beta_1 = 0$, while our analysis is more general). The condition $T \geq Kn$ to reach linear speedup considerably improves $\mathcal{O}(K^3n^3)$ of the federated momentum SGD analysis in [63]. In terms of communication complexity, by setting $K = \Theta(1/n\delta)$, Fed-EF only requires $T = \Theta(1/\delta)$ rounds of communication to converge. This matches one of the state-of-the-art FL communication complexity results of SCAFFOLD [27].

**Comparison with prior related results.** As a special case of Fed-EF-SGD ($\eta \equiv 1$) and the most relevant previous work, the analysis of QSparse-local-SGD [6] did not consider data heterogeneity, and their convergence rate $\mathcal{O}(1/\sqrt{TK})$ did not achieve linear speedup either. Our new analysis improves this result, showing that EF can also match the best rate of using full communication in federated learning. For FL with unbiased compression (without EF), the convergence rate of FedPaQ [47] is also $\mathcal{O}(1/\sqrt{TK})$. [18] refined the analysis and algorithm of FedPaQ, which matches our $\mathcal{O}(1/\delta)$ communication complexity. To sum up, both Fed-EF-SGD and Fed-EF-AMS are able to achieve the convergence rates of the corresponding full-precision FL counterparts, as well as the state-of-the-art rates of FL with unbiased compression.

## 4.2 ANALYSIS OF FED-EF UNDER PARTIAL CLIENT PARTICIPATION

Whilst being a popular strategy in classical distributed training, error feedback has rarely been analyzed under partial participation (PP), which is an important feature of FL. Next, we provide new analysis and results of EF under this setting, considering both local steps and data heterogeneity in federated learning. In each round $t$, assume only $m$ randomly chosen clients (without replacement) indexed by $\mathcal{M}_t \subseteq [n]$ are active and participate in training (i.e., changing $i \in [n]$ to $i \in \mathcal{M}_t$ at line 4 of Algorithm 1). For the remaining $(n - m)$ inactive clients, we simply set $e_{t,i} = e_{t-1,i}$, $\forall i \in [n] \setminus \mathcal{M}_t$. The convergence rate is given as below.

**Theorem 3** (Fed-EF, partial participation). *In each round, suppose $m$ randomly chosen clients in $\mathcal{M}_t$ participate in the training. Under Assumptions 1 to 3, suppose the learning rates satisfy* $\eta_l \leq \min\left\{\frac{1}{6}, \frac{m}{96C'\eta}, \frac{m^2}{53760(n-m)C_1\eta}, \frac{1}{4\eta}, \frac{1}{32C_1\eta}\right\}\frac{1}{KL}$. *Fed-EF-SGD admits*

$$\frac{1}{T}\sum_{t=1}^{T}\mathbb{E}\big[\|\nabla f(\theta_t)\|^2\big] \lesssim \frac{f(\theta_1) - f(\theta^*)}{\eta\eta_l TK} + \Big[\frac{\eta\eta_l L}{m} + \frac{8\eta\eta_l C_1 Ln}{m^2}\Big]\sigma^2 + \frac{3\eta\eta_l C'KL}{m}\sigma_g^2$$

$$+ \Big[\frac{5\eta_l^2 KL^2}{2} + \frac{15\eta\eta_l^3 C'K^2L^3}{m} + \frac{560\eta\eta_l C_1(n-m)L}{m^2}\Big](\sigma^2 + 6K\sigma_g^2),$$

*where $C_1 = \frac{q^2}{(1-q^2)^3}$ and $C' = \frac{n-m}{n-1}$. Choosing $\eta = \Theta(\sqrt{Km})$, $\eta_l = \Theta(\frac{\sqrt{m}}{K\sqrt{Tn}})$, we have*

$$\frac{1}{T}\sum_{t=1}^{T}\mathbb{E}\big[\|\nabla f(\theta_t)\|^2\big] = \mathcal{O}\Big(\frac{\sqrt{n}}{\sqrt{m}}\big(\frac{f(\theta_1) - f(\theta^*)}{\sqrt{TKm}} + \frac{1}{\sqrt{TKm}}\sigma^2 + \frac{\sqrt{K}}{\sqrt{Tm}}\sigma_g^2\big)\Big).$$

**Remark 1.** *We present Fed-EF-SGD for simplicity. With more complicated analysis, similar result applies to Fed-EF-AMS yielding the same asymptotic convergence rate as Fed-EF-SGD.*

**Remark 2.** *When $m = n$ (full participation), Theorem 3 recovers the $\mathcal{O}(1/\sqrt{TKn})$ rate in Corollary 1. When $q = 0$, we recover the $\mathcal{O}(\sqrt{K/Tm})$ rate of full-precision Fed-SGD under PP [60].*

**Effect of delayed error compensation.** The convergence rate in Theorem 3 involves $m$ in the denominator, instead of $n$ as in Corollary 1, which is a result of larger gradient estimation variance due to client sampling. Compared with the $\mathcal{O}(\sqrt{K/Tm})$ rate of [60] for full-precision local SGD under PP, Theorem 3 is slower by a factor of $\sqrt{n/m}$, which is a consequence of the mechanism of error feedback. Intuitively, with full participation where each client is active in every round, EF itself can, to a large extent, be regarded as subtly "delaying" the "untransmitted" gradient information

$(\mathcal{C}(\Delta_t) - \Delta_t)$ to the next iteration. However, under partial participation, in each round $t$, the error accumulator of a chosen client actually contains the latest information from round $t - s$, where $s$ can be viewed as the "lag" which follows a geometric distribution with $\mathbb{E}[s] = n/m$. In some sense, this shares similar spirit to the problem of asynchronous distributed optimization with delayed gradients (e.g., [1; 36]). The delayed error information in Fed-EF under PP is likely to pull the model away from heading towards a stationary point (i.e., slower down the norm convergence), especially for highly non-convex loss functions. In Section 5, we will propose a simple strategy to justify (and mitigate) the negative impact of this error staleness on the norm convergence empirically.

## 5  NUMERICAL STUDY

We provide numerical results to show the efficacy of Fed-EF in communication-efficient FL problems and justify our theoretical analysis. Due to space limitation, we include representative results here and place more results and experimental details in Appendix A.

**Datasets.** We present experiments on two popular FL datasets. The MNIST dataset [32] contains 60000 training examples and 10000 test samples of $28 \times 28$ gray-scale hand-written digits from 0 to 9. The FMNIST dataset [59] has the same input size and train/test split as MNIST, but the samples are fashion products (e.g., clothes and bags). More results on CIFAR dataset are included in Appendix A.

**Federated setting.** In our experiments, we test $n = 200$ clients. The clients' local data are set to be highly non-iid (heterogeneous), where we restrict the local data samples of each client to come from at most two classes. We run $T = 100$ rounds, where one FL training round is finished after all the clients have performed one epoch of local training. The local mini-batch size is 32, which means that the clients conduct 10 local iterations per round. Regarding partial participation, we uniformly randomly sample $m$ clients in each round. We present the results at multiple sampling proportion $p = m/n$ (e.g., $p = 0.1$ means choosing 20 active clients per round). To measure the communication cost, we report the accumulated number of bits transmitted from the client to server (averaged over all clients), assuming that full-precision gradients are 32-bit encoded.

**Methods and compressors.** For both Fed-EF variants, we implement **Sign** compressor, and **TopK** compressor with compression rate $k \in \{0.001, 0.01, 0.05\}$. We also employ a more compressive strategy **heavy-Sign** where **Sign** is applied after **TopK** (i.e., a further 32x compression over **TopK** under same sparsity). We test **hv-Sign** with $k \in \{0.01, 0.05, 0.1\}$. We compare our method with the analogue FL approach using full-precision updates, and the analogue algorithms of Fed-EF using unbiased stochastic quantization **"Stoc" without error feedback** [3]. For this compressor, we test parameter $b \in \{1, 2, 4\}$. For SGD, this algorithm is equivalent to FedCOM/FedPaQ [47; 18].

### 5.1  FED-EF MATCHES FULL-PRECISION FL WITH SUBSTANTIALLY LESS COMMUNICATION

Firstly, we demonstrate the superior performance of Fed-EF in practical FL tasks. For both datasets, we train a ReLU activated CNN with two convolutional layers followed by one max-pooling, one dropout and two fully-connected layers before the softmax output. In Figure 2 and Figure 3, we compare our method with **Stoc** without EF with $p = 0.5$ and $p = 0.1$, respectively. We have tested each compressor with multiple compression ratios, see Appendix A for the complete results. Here, we present curves (respective compression ratios) chosen by the following rule: for each method, we present the curve with highest compression level that achieves the best full-precision test accuracy; if the method does not match the full-precision performance, we present the curve with the highest test accuracy. From Figure 2 and Figure 3, we see that:

- In general, higher compression ratio leads to worse performance, as expected from the theory. The proposed Fed-EF (including both variants) is able to achieve the same performance as the full-precision methods with substantial communication reduction, e.g., **hv-Sign** and **TopK** reduce the communication by more than 100x without losing accuracy. **Sign** also provides 30x compression with matching accuracy as full-precision training.

- On MNIST, the loss and accuracy of **Stoc** (stochastic quantization without EF) tend to be slightly worse than Fed-EF-SGD with **hv-Sign**, yet requiring more communication.

- With more aggressive $p = 0.1$, with proper compressor, Fed-EF still matches the performance of full-precision algorithms. While **Sign** performs well on MNIST for both Fed-EF

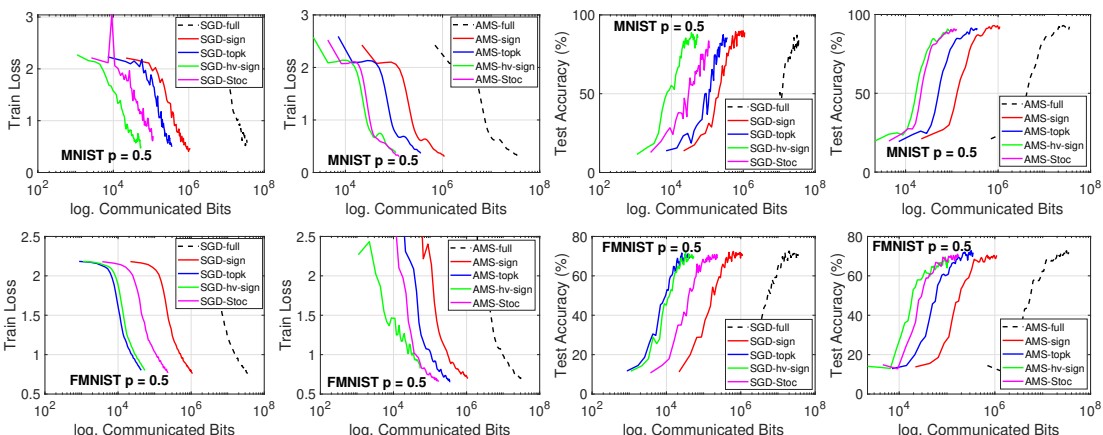

Figure 2: Training loss and test accuracy v.s. communicated bits, participation rate $p = 0.5$. "sign", "topk" and "hv-sign" are applied with Fed-EF, while "Stoc" is the stochastic quantization without EF.

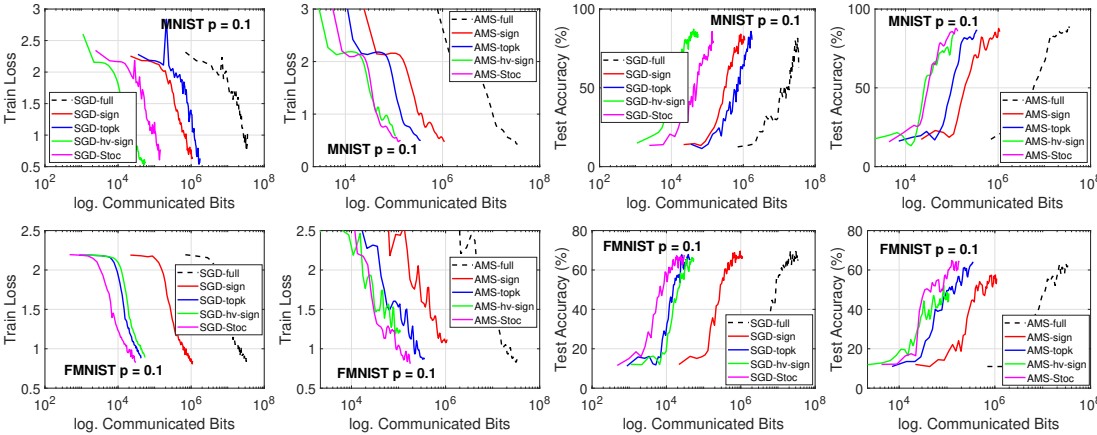

Figure 3: Training loss and test accuracy v.s. communicated bits, participation rate $p = 0.1$. "sign", "topk" and "hv-sign" are applied with Fed-EF, while "Stoc" is the stochastic quantization without EF.

variants, we notice that fixed sign-based compressors (**Sign** and **heavy-Sign**) are considerably outperformed by **TopK** for Fed-EF-AMS on FMNIST. We conjecture that this is because with small participation rate, sign-based compressors tend to assign a same implicit learning rate across coordinates (controlled by the second moment $\hat{v}$), making adaptive method less effective. In contrast, magnitude-preserving compressors (e.g., **TopK** and **Stoc**) may better exploit the adaptivity of AMSGrad.

## 5.2 ANALYSIS OF NORM CONVERGENCE AND DELAYED ERROR COMPENSATION

We empirically evaluate the norm convergence to verify the theoretical speedup properties of Fed-EF and the effect of delayed error compensation in partial participation (PP). Recall that from Theorem 1, in full participation case, reaching a $\delta$-stationary point requires running $\Theta(1/n\delta^2)$ rounds of Fed-EF (i.e., linear speedup). From Theorem 3, when $n$ is fixed, our result implies that the speedup should be super-linear against $m$, the number of participating clients, due to the additional stale error effect. In other words, altering $m$ is expected to have more impact on the convergence under PP.

We train an MLP (which is also used for Figure 1) with one hidden layer of 200 neurons. In Figure 4, we report the squared gradient norm and the training loss on MNIST under the same non-iid FL setting as above (the results on FMNIST are similar). In the full participation case, we implement Fed-EF-SGD with $n = 20, 40, 60, 100$ clients; for the partial participation case, we fix $n = 200$ and

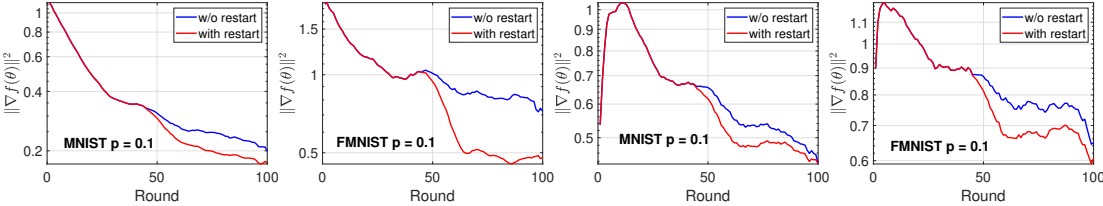

Figure 4: MLP on MNIST with **TopK-0.01** compressed Fed-EF: Squared gradient norm (left two) and train loss (right two) against the number of training rounds, averaged over 20 independent runs.

Figure 5: Squared gradient norm of Fed-EF (**TopK-0.01**) under PP with error restarting, $S = 10$. Left two panels: logistic regression. Right two panels: MLP. $n = 200$, $\eta = 1$, $\eta_l = 0.1$.

alter $m = 20, 40, 60, 100$. According to our theory, we set $\eta = 0.1\sqrt{n}$ (or $0.1\sqrt{m}$) and $\eta_l = 0.1$. We see that: 1) In general, the convergence of Fed-EF is faster with increasing $n$ or $m$, which confirms the speedup property; 2) The gaps among curves in the PP setting (the 2nd and 4th plot) is larger than those in the full-participation case (the 1st and 3rd plot), which suggests that the acceleration brought by increasing $m$ under PP is more significant than that of increasing $n$ in full-participation by a same proportion, which is consistent with our theoretical implications.

To further embody the intuitive impact of delayed error compensation under PP, we test a simple strategy called *"error restarting"*: for each client $i$ in round $t$, if the error accumulator was last updated more than $S$ (a threshold) rounds ago (i.e., before round $t - S$), we simply restart the error accumulator by setting $e_t = 0$, which effectively eliminates the error information that is "too old". In Figure 5, we first run Fed-EF for 50 rounds, and then apply error restarting with threshold $S = 10$. As we see, after prohibiting heavily delayed error information, the gradient norm is smaller than that of continuing running standard Fed-EF, i.e., the model finds a stationary point faster. These results illustrate the influence of stale error compensation in Fed-EF, and that properly handling this staleness might be a promising direction for improvement in the future.

## 6 DISCUSSION AND CONCLUSION

We propose Fed-EF, a Federated Learning (FL) framework with compressed communication and Error Feedback (EF). Two variants, Fed-EF-SGD and Fed-EF-AMS, are designed based on the choice of the global optimizer. Theoretically, we present convergence analysis in non-convex optimization showing that Fed-EF achieves the same convergence rate as the full-precision FL counterparts, which improves upon previous results. The Fed-EF-AMS variant is the first compressed adaptive FL method in literature. Moreover, we develop new analysis of error feedback in distributed training systems under the partial participation setting. We prove an additional slow down factor related to the participation rate due to the delayed error compensation of the EF mechanism. Experiments validate that compared with full-precision training, Fed-EF achieves significant communication reduction without performance drop. We also present numerical results to justify the theory and provide intuition regarding the impact of the delayed error feedback on the norm convergence of Fed-EF. Our work supports the effectiveness of error feedback in federated learning, and provide insight on its convergence under practical FL setting with partial participation. Our paper expands several interesting future directions, e.g., to improve Fed-EF by other tricks, especially under partial participation, and to study more closely the property of different compressors. More mechanisms in FL (e.g., variance reduction, fairness) can also be incorporated into our Fed-EF scheme.

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

# Appendices

CONTENTS

# A  EXPERIMENT DETAILS, ALGORITHMS AND MORE RESULTS

In this section, we provide more theoretical justification of the compressors, and more implementation details of the empirical results.

## A.1  BIASED COMPRESSION OPERATORS

In our Fed-EF, the biased compressors are implemented as follows. **Sign** is implemented exactly following Definition 1. For **TopK**, we also apply it in a "layer-wise" manner. Let $k$ denote the proportion of coordinates selected. For each layer with $d_i$ parameters, we pick $\max(1, \lfloor kd_i \rfloor)$ gradient dimensions. The maximum operator avoids the case where a layer is never updated. The **heavy-Sign** is implemented by first applying **TopK** (per layer) and then applying **Sign**.

For completeness, we provide more theoretical details of the biased compression operators, **TopK**, **Sign** and **heavy-Sign**. Recall in Definition 1 that $\|\mathcal{C}(x) - x\|^2 \leq q_\mathcal{C}^2 \|x\|^2$ for some $q_\mathcal{C} < 1$. We first justify that **TopK** and **Sign** are both valid compressors. In the sequel, $\|\cdot\|$ always denotes the $l_2$ norm and $\|\cdot\|_1$ is the $l_1$ norm.

Proposition A.1 is well-known (e.g., [54; 68]), and we provide the proof for clarity. Again, note that in **TopK**, $k$ is the compression rate, which is the fraction, instead of the number, of selected coordinates.

**Proposition A.1.** *For the **TopK** compressor which selects top $k$-percent of coordinates, we have $q_\mathcal{C}^2 = 1 - k$. For the (**Group) Sign** compressor, $q_\mathcal{C}^2 = 1 - \min_{i \in [M]} \frac{1}{d_i}$.*

*Proof.* For **TopK**, the proof is trivial: since $\mathcal{C}(x) - x$ only contain $(1-k)d$ coordinates with lowest magnitudes, we know $\|C(x) - x\|^2 / \|x^2\| \leq 1 - k$.

For **Sign**, recall that $\mathcal{I}_i$ is the index set of block (group) $i$. By definition, for the $i$-th block (group) $x_{\mathcal{I}_i} \in \mathbb{R}^{d_i}$, we have

$$\|\mathcal{C}(x_{\mathcal{I}_i}) - x_{\mathcal{I}_i}\|^2 = \|x_{\mathcal{I}_i} - \frac{\|x_{\mathcal{I}_i}\|_1}{d_i} sign(x_{\mathcal{I}_i})\|^2$$
$$= \|x_{\mathcal{I}_i}\|^2 + \frac{\|x_{\mathcal{I}_i}\|_1^2}{d_i^2} \cdot d_i - \frac{2\|x_{\mathcal{I}_i}\|_1^2}{d_i}$$
$$= \|x_{\mathcal{I}_i}\|^2 - \|x_{\mathcal{I}_i}\|_1^2 / d_i.$$

Since we have $M$ blocks, concatenating the blocks leads to

$$\|\mathcal{C}(x) - x\|^2 = \sum_{i=1}^{M} \left( \|x_{\mathcal{I}_i}\|^2 - \|x_{\mathcal{I}_i}\|_1^2 / d_i \right)$$
$$= \|x\|^2 - \sum_{i=1}^{M} \|x_{\mathcal{I}_i}\|_1^2 / d_i$$
$$= \left(1 - \frac{\sum_{i=1}^{M} \|x_{\mathcal{I}_i}\|_1^2 / d_i}{\|x\|^2}\right) \|x\|^2$$
$$\leq \left(1 - \min_{i \in [M]} \frac{\|x_{\mathcal{I}_i}\|_1^2}{d_i \|x_{\mathcal{I}_i}\|^2}\right) \|x\|^2 \leq \left(1 - \min_{i \in [M]} \frac{1}{d_i}\right) \|x\|^2,$$

where the last inequality is because $l_1$ norm is larger than $l_2$ norm. $\square$

We now show that **heavy-Sign** is also a valid compressor.

**Proposition A.2.** *The **heavy-Sign** compressor satisfies Definition 1 with $q_\mathcal{C}^2 = 1 - \min_{i \in [M]} \frac{k}{d_i}$.*

*Proof.* Let $\mathcal{C}_k$ denote the **TopK** compressor and $\mathcal{C}_s$ be the **Sign** operator. Thus, the **heavy-Sign** operator can be expressed as $\mathcal{C}(x) = \mathcal{C}_s(\mathcal{C}_k(x))$. It holds that

$$\|\mathcal{C}(x) - x\|^2 = \|\mathcal{C}_s(\mathcal{C}_k(x)) - \mathcal{C}_k(x) + \mathcal{C}_k(x) - x\|^2$$
$$= \|\mathcal{C}_s(\mathcal{C}_k(x)) - \mathcal{C}_k(x)\|^2 + \|\mathcal{C}_k(x) - x\|^2,$$

because **TopK** zeros out the unpicked coordinates. By Proposition A.1, we continue to obtain

$$\|\mathcal{C}(x) - x\|^2 \leq (1 - \min_{i \in [M]} \frac{1}{d_i})\|\mathcal{C}_k(x)\|^2 + \|\mathcal{C}_k(x) - x\|^2$$

$$= \|x\|^2 - \min_{i \in [M]} \frac{1}{d_i}\|\mathcal{C}_k(x)\|^2 \leq (1 - \min_{i \in [M]} \frac{k}{d_i})\|x\|^2,$$

where we use the fact that $\|\mathcal{C}_k(x)\|^2 + \|\mathcal{C}_k(x) - x\|^2 = \|x\|^2$, and $\|\mathcal{C}(x)\| \geq k\|x\|^2$ by Proposition A.1. This completes the proof. $\qquad\square$

## A.2 COMPETING METHODS: STOCHASTIC QUANTIZATION WITHOUT ERROR FEEDBACK

---

**Algorithm 2** Federated Learning with Unbiased Compression (Competing method **Stoc**)

---

1: **Input**: learning rates $\eta$, $\eta_l$, hyper-parameters $\beta_1$, $\beta_2$, $\epsilon$
2: **Initialize**: central server parameter $\theta_1 \in \mathbb{R}^d \subseteq \mathbb{R}^d$; $e_{1,i} = \mathbf{0}$ the accumulator for each worker; $m_0 = \mathbf{0}$, $v_0 = \mathbf{0}$, $\hat{v}_0 = \mathbf{0}$
3: **for** $t = 1, \ldots, T$ **do**
4:     **parallel for worker** $i \in [n]$ **do**:
5:         Receive model parameter $\theta_t$ from central server, set $\theta_{t,i}^{(1)} = \theta_t$
6:         **for** $k = 1, \ldots, K$ **do**
7:             Compute stochastic gradient $g_{t,i}^{(k)}$ at $\theta_{t,i}^{(k)}$
8:             Local update $\theta_{t,i}^{(k+1)} = \theta_{t,i}^{(k)} - \eta_l g_{t,i}^{(k)}$
9:         **end for**
10:        Compute the local model update $\Delta_{t,i} = \theta_{t,i}^{(K+1)} - \theta_t$
11:        Send quantized local model update $\widetilde{\Delta}_{t,i} = \mathcal{Q}(\Delta_{t,i})$ to central server using (2)
12:     **end parallel**
13:     **Central server do:**
14:     Global aggregation $\overline{\widetilde{\Delta}}_t = \frac{1}{n}\sum_{i=1}^n \widetilde{\Delta}_{t,i}$
15:     Update the global model $\theta_{t+1} = \theta_t - \eta\overline{\widetilde{\Delta}}_t$            ▷ **Stoc** with SGD
16:     $m_t = \beta_1 m_{t-1} + (1 - \beta_1)\overline{\widetilde{\Delta}}_t$            ▷ **Stoc** with AMSGrad
17:     $v_t = \beta_2 v_{t-1} + (1 - \beta_2)\overline{\widetilde{\Delta}}_t^2, \quad \hat{v}_t = \max(v_t, \hat{v}_{t-1})$
18:     Update the global model $\theta_{t+1} = \theta_t - \eta\frac{m_t}{\sqrt{\hat{v}_t} + \epsilon}$
19: **end for**

---

In Algorithm 2, for completeness, we give the details of the competing method, called **Stoc**, in our experiments. Instead of using the error feedback scheme, this method directly compresses the transmitted vector from clients to server by unbiased stochastic quantization $\mathcal{Q}(\cdot)$ proposed by [3]. For a vector $x \in \mathbb{R}^d$, the operator $\mathcal{Q}(\cdot)$ is defined as

$$\mathcal{Q}_b(x) = \|x\| \cdot sign(x) \cdot \xi(x, b), \tag{2}$$

where $b \geq 1$ is number of bits per non-zero entry of the compressed vector $\mathcal{Q}(x)$. Suppose $0 \leq l < 2^{b-1}$ is the integer such that $|x_i|/\|x\|$ is contained in the interval $[l/2^{b-1}, (l+1)/2^{b-1}]$. The random variable $\xi(x, b)$ is defined by

$$\xi(x, b) = \begin{cases} l/s, & \text{with probability } 1 - g(\frac{|x_i|}{\|x\|}, b), \\ (l+1)/s, & \text{otherwise,} \end{cases}$$

with $g(a, b) = a \cdot 2^{b-1} - l$ for $a \in [0, 1]$. Simply, 0 is always quantized to 0. The **Stoc** quantizer is unbiased, i.e., $\mathbb{E}[\mathcal{Q}(x)|x] = x$. In addition, it also introduces sparsity to the compressed vector in a probabilistic way, with $\mathbb{E}[\|\mathcal{Q}(x)\|_0] \leq 2^b + 2^{b-1}\sqrt{d}$.

**Stoc** also has two corresponding variants, one using SGD and one using AMSGrad as the global optimizer. For the SGD variant, **Stoc** is equivalent to the FedCOM method in [18], which is also the FedPaQ algorithm [47] with tunable global learning rate.

For the full-precision algorithms, we simply set $\widetilde{\Delta}_{t,i} = \Delta_{t,i}$ in line 11 of Algorithm 2. For SGD, it becomes the one studied in [60] which is the standard local SGD [42] with global learning rate. For adaptive optimizer, it becomes FedAdam [45]. Note that [45] used Adam, while we use AMSGrad (with the max operation). Empirically, the performance of these two options are fairly similar.

### A.3    RESULTS ON CIFAR-10 TRAINED BY RESNET-18

We present more experiment results of Fed-EF on the task of CIFAR-10 [31] image classification. This dataset contains 50000 natural images of size $32 \times 32$ each with 3 RGB channels. There are 10 classes, e.g., airplanes, cars, cats, etc. We follow a standard strategy for CIFAR-10 dataset to pre-process the training images by a random crop, a random horizontal flip and a normalization of the pixel values to have zero mean and unit variance. For test images, we only apply the normalization step. For this experiment, we train a ResNet-18 [20] network for 200 rounds. The clients local data are distributed in the same way as described in Section 5, which is highly non-iid.

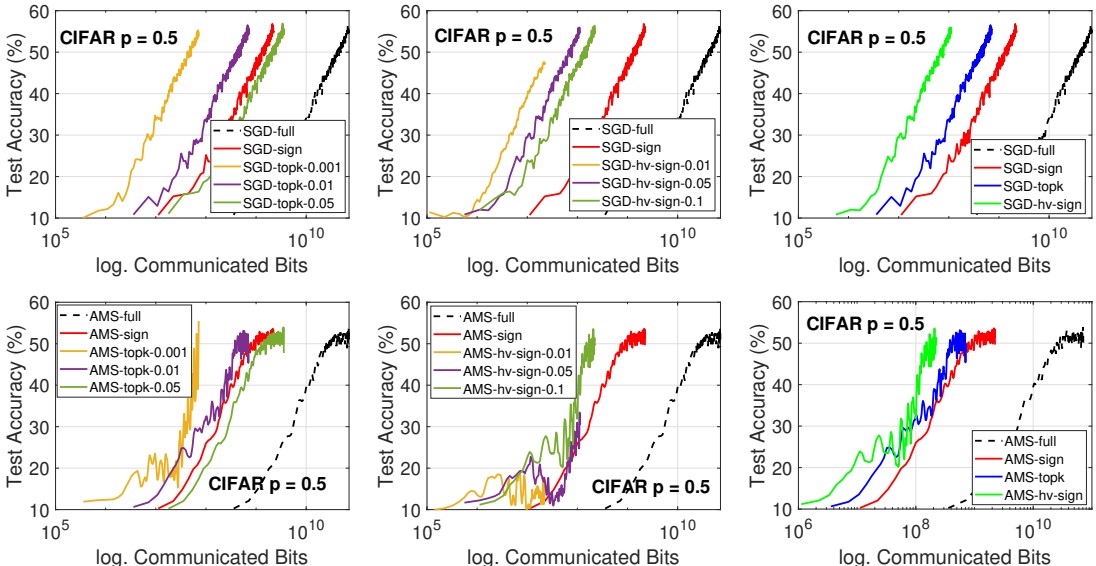

Figure 6: CIFAR-10 dataset trained by ResNet-18. Test accuracy of Fed-EF with **TopK**, **Sign** and **heavy-Sign** compressors. Participation rate $p = 0.5$, non-iid data. "sign", "topk" and "hv-sign" are applied with Fed-EF, while "Stoc" is the stochastic quantization without EF. 1st row: Fed-EF-SGD. 2nd row: Fed-EF-AMS. The last column presents the corresponding curves that achieve the full-precision accuracy using lowest communication.

In Figure 6, we plot the test accuracy of Fed-EF with different compressors, and **Stoc** without EF. Again, we see that Fed-EF (both variants) is able to attain the same accuracy level as the corresponding full-precision federated learning algorithms. For Fed-EF-SGD, the compression rate is around 32x for **Sign**, 100x for **TopK** and ∼300x for **heavy-Sign**. For Fed-EF-AMS, the compression ratio can also be around hundreds. Note that for Fed-EF-AMS, the training curve of **TopK-0.001** is not stable. Though it reaches a high accuracy, we still plot **TopK-0.01** in the third column for comparison.

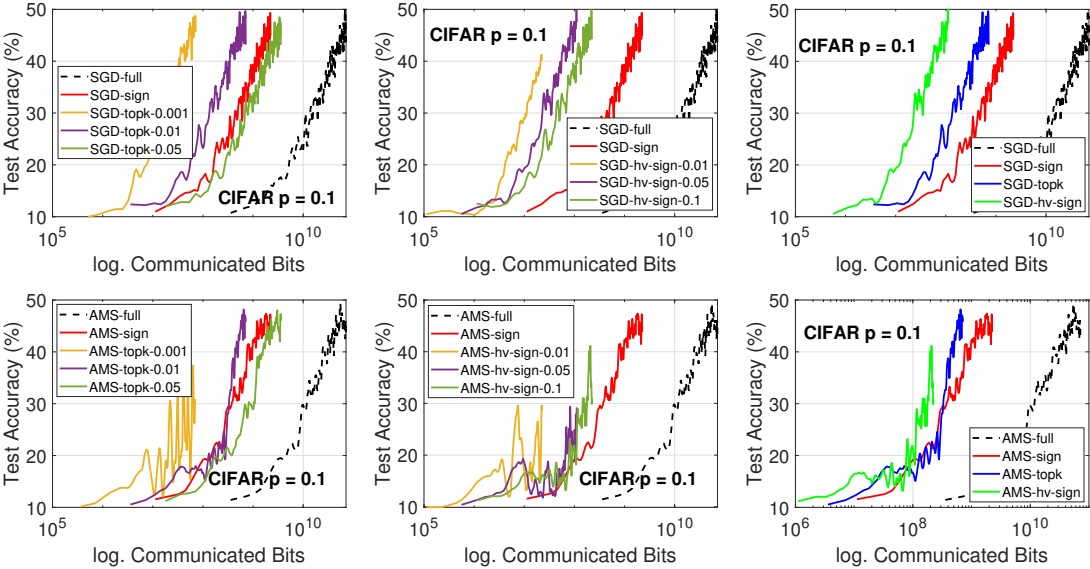

Figure 7: CIFAR-10 dataset trained by ResNet-18. Test accuracy of Fed-EF with **TopK**, **Sign** and **heavy-Sign** compressors. Participation rate $p = 0.1$, non-iid data. "sign", "topk" and "hv-sign" are applied with Fed-EF, while "Stoc" is the stochastic quantization without EF. 1st row: Fed-EF-SGD. 2nd row: Fed-EF-AMS. The last column presents the corresponding curves that achieve the full-precision accuracy using lowest communication.

In Figure 7 we report the results for aggressive partial participation with $p = 0.1$. Similarly, for SGD, all three compressors are able to match the full-precision accuracy, with significantly reduced number of communicated bits. For Fed-EF-AMS, similar to the observations on FMNIST, we see that **TopK** outperforms **Sign** and **heavy-Sign**, and matches the performance of full-precision method with 100x compression ratio. **Sign** also performs reasonably well.

In conclusion, our results on CIFAR-10 and ResNet again confirm that compared with standard full-precision FL algorithms, the proposed Fed-EF scheme can provide significant communication reduction without performance drop, under data heterogeneity and partial participation.

## A.4   MORE RESULTS ON MNIST AND FMNIST

We provide the complete set of experimental results on each method under various compression rates. In Table 1 - Table 4, for completeness we report the average test accuracy at the end of training and the standard deviations (over 5 independent runs), corresponding to the curves (compression parameters) in Figure 2 and Figure 3. Figure 8 to Figure 11 present the results for participation rate $p = 0.5$, and Figure 12 to Figure 15 report the results for $p = 0.1$.

|  | Fed-EF-SGD | | | No EF | |
|---|---|---|---|---|---|
|  | Sign | TopK | Hv-Sign | Stoc | Full-precision |
| MNIST | 90.87 ($\pm$0.84) | 91.04 ($\pm$1.05) | 91.18 ($\pm$1.10) | 90.16 ($\pm$0.96) | 90.85 ($\pm$0.89) |
| FMNIST | 71.13 ($\pm$0.68) | 71.16 ($\pm$0.77) | 71.07 ($\pm$0.83) | 71.26 ($\pm$0.87) | 71.20 ($\pm$0.71) |

Table 1: Test accuracy (%) with client participation rate $p = 0.5$, of Fed-EF-SGD with **Sign**, **TopK** and **heavy-Sign** compressor and **Stoc** (stochastic quantization) without EF. The compression parameters (i.e., $k$ and $b$) of the compressors are consistent with Figure 2.

| | Fed-EF-AMS | | | No EF | |
|---|---|---|---|---|---|
| | Sign | TopK | Hv-Sign | Stoc | Full-precision |
| MNIST | 92.32 ($\pm$0.98) | 92.74 ($\pm$0.84) | 91.77 ($\pm$1.22) | 92.36 ($\pm$0.93) | 92.23 ($\pm$0.73) |
| FMNIST | 71.35 ($\pm$0.61) | 71.90 ($\pm$0.78) | 70.73($\pm$1.03) | 71.94 ($\pm$0.95) | 71.97 ($\pm$0.86) |

Table 2: Test accuracy (%) with client participation rate $p = 0.5$, of Fed-EF-AMS with **Sign**, **TopK** and **heavy-Sign** compressor and **Stoc** (stochastic quantization) without EF. The compression parameters (i.e., $k$ and $b$) of the compressors are consistent with Figure 2.

| | Fed-EF-SGD | | | No EF | |
|---|---|---|---|---|---|
| | Sign | TopK | Hv-Sign | Stoc | Full-precision |
| MNIST | 90.15 ($\pm$1.06) | 90.61 ($\pm$0.93) | 90.42 ($\pm$1.09) | 90.27 ($\pm$1.18) | 90.22 ($\pm$0.82) |
| FMNIST | 67.69 ($\pm$0.73) | 67.47 ($\pm$0.80) | 67.72 ($\pm$0.55) | 67.71 ($\pm$0.78) | 67.50 ($\pm$0.85) |

Table 3: Test accuracy (%) with client participation rate $p = 0.1$, of Fed-EF-SGD with **Sign**, **TopK** and **heavy-Sign** compressor and **Stoc** (stochastic quantization) without EF. The compression parameters (i.e., $k$ and $b$) of the compressors are consistent with Figure 3.

| | Fed-EF-AMS | | | No EF | |
|---|---|---|---|---|---|
| | Sign | TopK | Hv-Sign | Stoc | Full-precision |
| MNIST | 88.67 ($\pm$1.11) | 88.97 ($\pm$1.16) | 77.49 ($\pm$1.53) | 88.76 ($\pm$1.22) | 89.05 ($\pm$1.04) |
| FMNIST | 57.60 ($\pm$2.34) | 64.09 ($\pm$0.91) | 50.77($\pm$2.87) | 64.35 ($\pm$1.06) | 64.18 ($\pm$0.90) |

Table 4: Test accuracy (%) with client participation rate $p = 0.1$, of Fed-EF-AMS with **Sign**, **TopK** and **heavy-Sign** compressor and **Stoc** (stochastic quantization) without EF. The compression parameters (i.e., $k$ and $b$) of the compressors are consistent with Figure 3.

## A.5 DATA SPLIT AND PARAMETER TUNING

In our experiments, for $n = 200$ clients, we first split the training samples into $2n = 400$ shards, where each shard contains samples from only one class. Then, each client is randomly assigned with two shards uniformly. This way, in expectation, around 180 clients would have samples from two classes, and about 20 clients would have samples from only one class. This corresponds to a strong data heterogeneity among the clients.

For the hyper-parameter of the compressors (i.e., the compression rate), we test $k \in \{0.001, 0.01, 0.05\}$ for **TopK**, $k \in \{0.01, 0.05, 0.1\}$ for **heavy-Sign** and $b \in \{1, 2, 4\}$ for **Stoc**. For the AMSGrad optimizer, we set $\beta_1 = 0.9$, $\beta_2 = 0.999$ and $\epsilon = 10^{-8}$ as the recommended default [46]. For each method, we tune $\eta$ over $\{10^{-4}, 10^{-3}, 10^{-2}, 10^{-1}, 1, 10\}$ and $\eta_l$ over $\{10^{-4}, 10^{-3}, 10^{-2}, 10^{-1}, 1\}$. We found that the compressed methods usually have same optimal learning rates as the full-precision training. The best learning rate combinations achieving highest test accuracy are given in Table 5.

| | Fed-EF-SGD | | Fed-EF-AMS | |
|---|---|---|---|---|
| | $\eta$ | $\eta_l$ | $\eta$ | $\eta_l$ |
| MNIST | 10 | $10^{-3}$ | $10^{-3}$ | $10^{-2}$ |
| FMNIST | 1 | $10^{-1}$ | $10^{-2}$ | $10^{-1}$ |
| CIFAR-10 | 1 | $10^{-1}$ | $10^{-3}$ | $10^{-2}$ |

Table 5: Optimal global ($\eta$) and local ($\eta_l$) learning rate combinations to attain highest test accuracy.

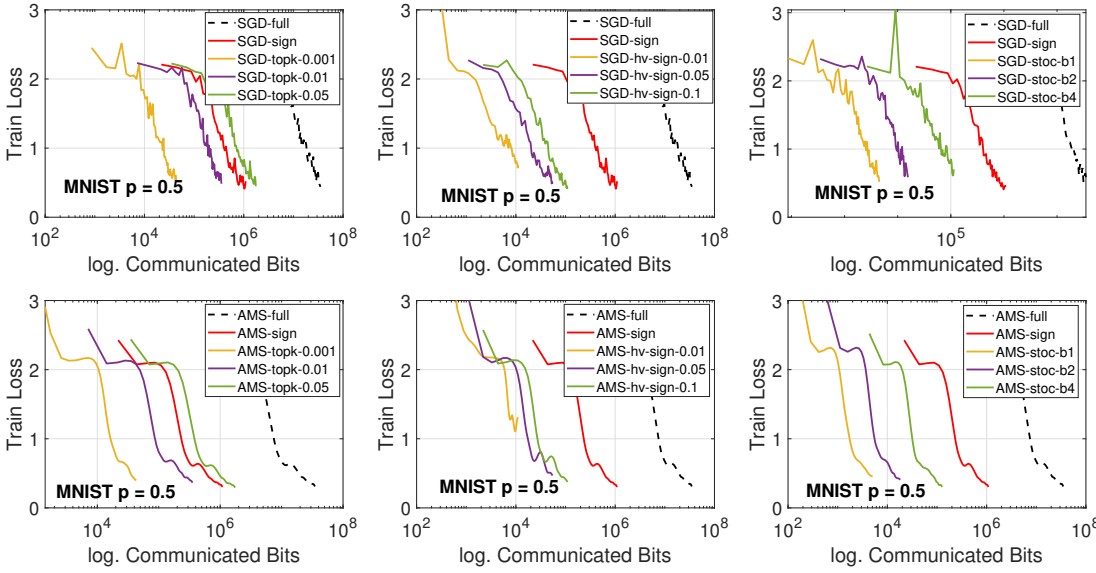

Figure 8: MNIST dataset trained by CNN. Training loss of Fed-EF with **TopK**, **Sign** and **heavy-Sign** compressors, and **Stoc** without EF. Participation rate $p = 0.5$, non-iid data. "sign", "topk" and "hv-sign" are applied with Fed-EF, while "Stoc" is the stochastic quantization without EF. 1st row: Fed-EF-SGD. 2nd row: Fed-EF-AMS.

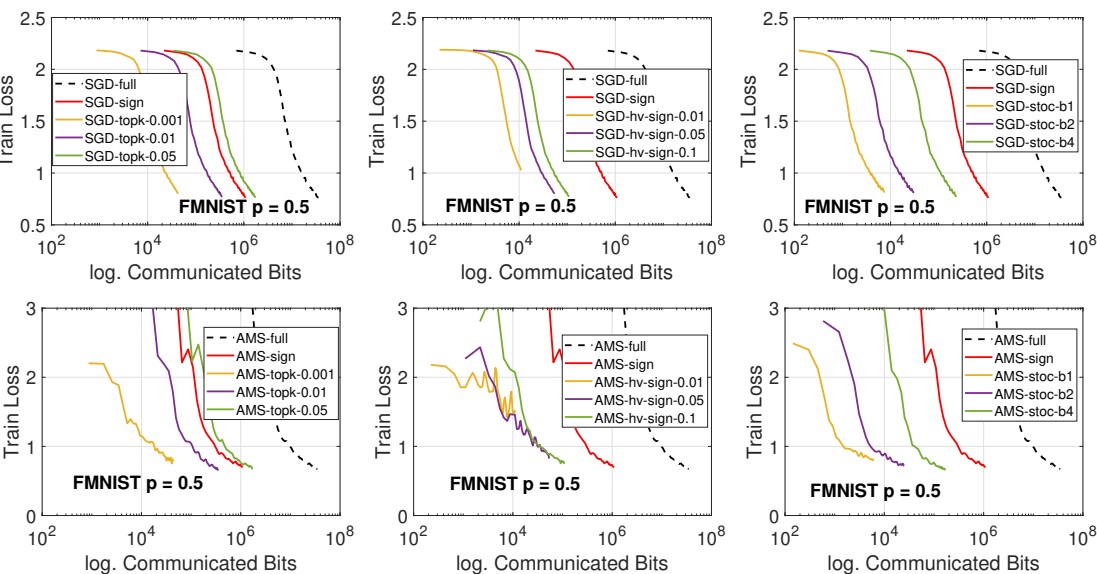

Figure 9: FMNIST dataset trained by CNN. Training loss of Fed-EF with **TopK**, **Sign** and **heavy-Sign** compressors, and **Stoc** without EF. Participation rate $p = 0.5$, non-iid data. "sign", "topk" and "hv-sign" are applied with Fed-EF, while "Stoc" is the stochastic quantization without EF. 1st row: Fed-EF-SGD. 2nd row: Fed-EF-AMS.

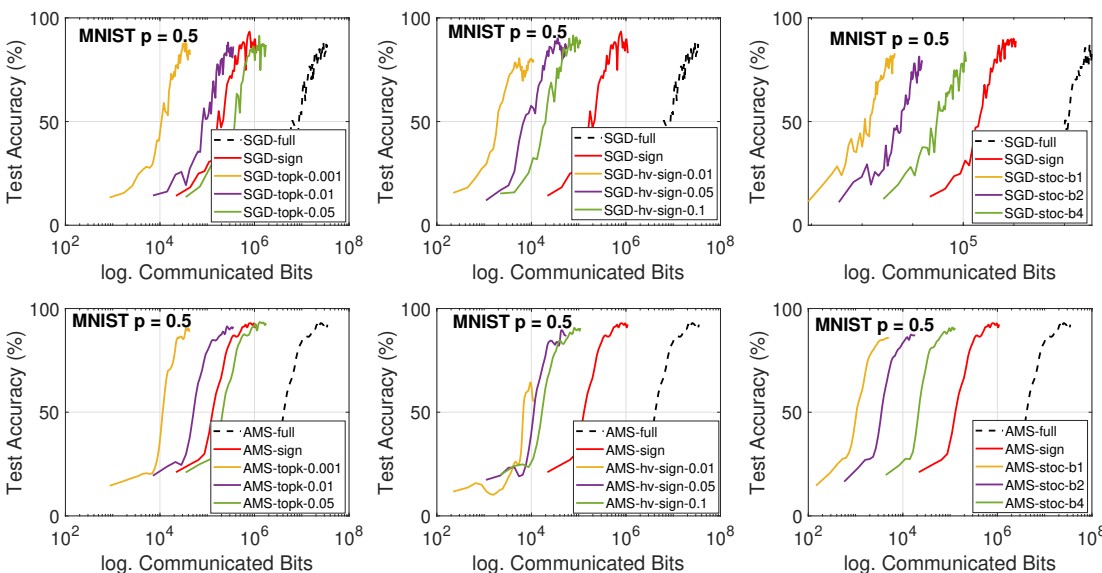

Figure 10: MNIST dataset trained by CNN. Test accuracy of Fed-EF with **TopK**, **Sign** and **heavy-Sign** compressors, and **Stoc** without EF. Participation rate $p = 0.5$, non-iid data. "sign", "topk" and "hv-sign" are applied with Fed-EF, while "Stoc" is the stochastic quantization without EF. 1st row: Fed-EF-SGD. 2nd row: Fed-EF-AMS.

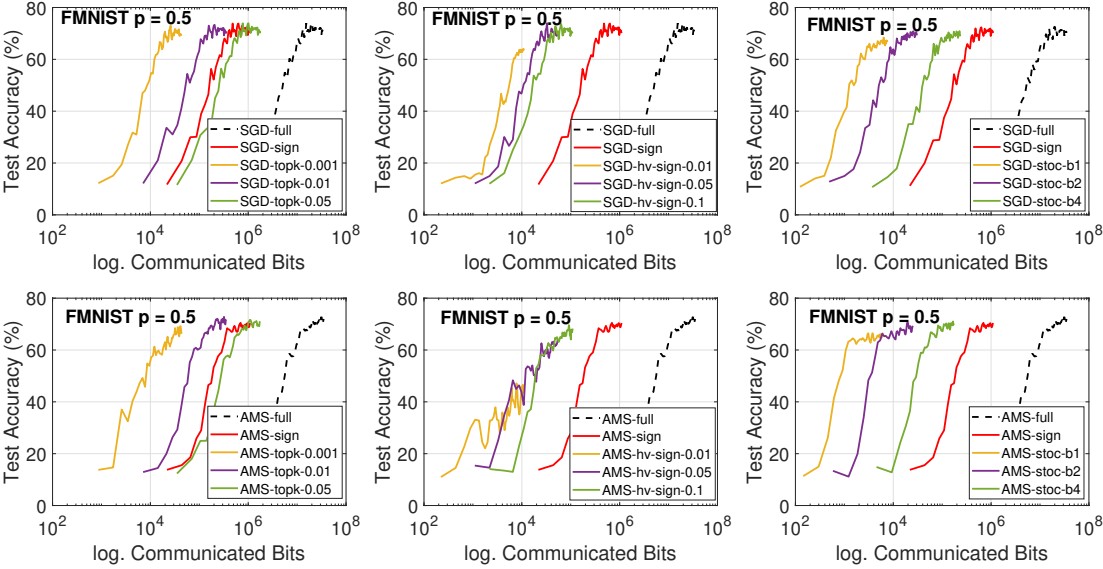

Figure 11: FMNIST dataset trained by CNN. Test accuracy of Fed-EF with **TopK**, **Sign** and **heavy-Sign** compressors, and **Stoc** without EF. Participation rate $p = 0.5$, non-iid data. "sign", "topk" and "hv-sign" are applied with Fed-EF, while "Stoc" is the stochastic quantization without EF. 1st row: Fed-EF-SGD. 2nd row: Fed-EF-AMS.

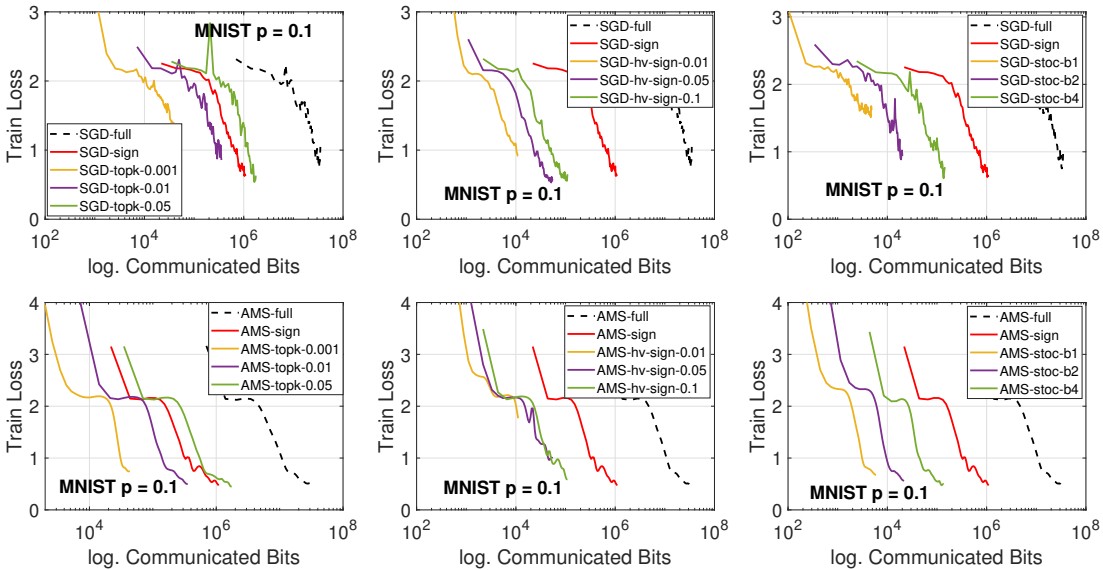

Figure 12: MNIST dataset trained by CNN. Training loss of Fed-EF with **TopK**, **Sign** and **heavy-Sign** compressors, and **Stoc** without EF. Participation rate $p = 0.1$, non-iid data. "sign", "topk" and "hv-sign" are applied with Fed-EF, while "Stoc" is the stochastic quantization without EF. 1st row: Fed-EF-SGD. 2nd row: Fed-EF-AMS.

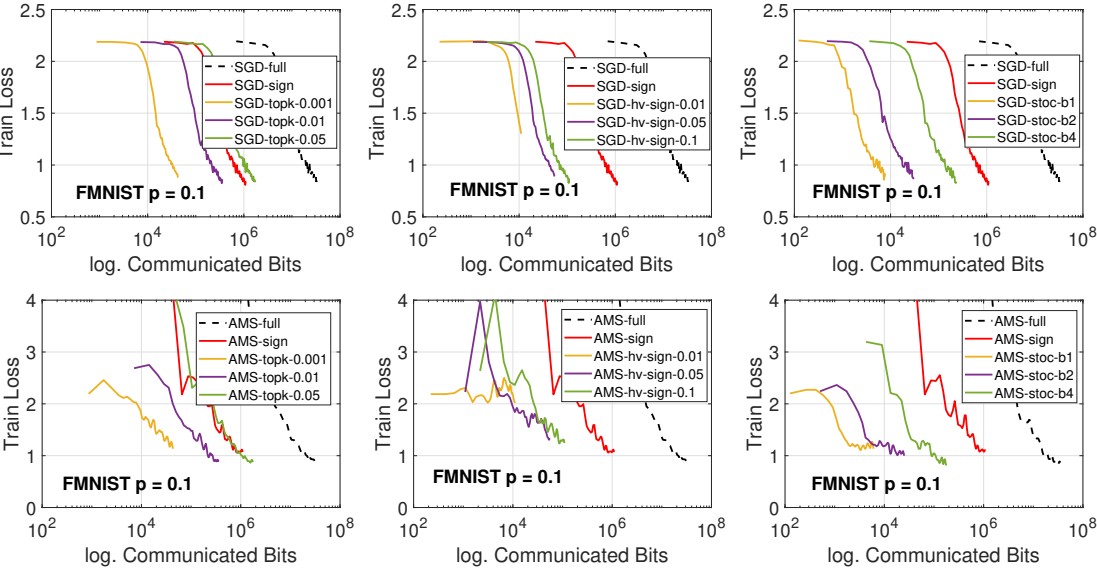

Figure 13: FMNIST dataset trained by CNN. Training loss of Fed-EF with **TopK**, **Sign** and **heavy-Sign** compressors, and **Stoc** without EF. Participation rate $p = 0.1$, non-iid data. "sign", "topk" and "hv-sign" are applied with Fed-EF, while "Stoc" is the stochastic quantization without EF. 1st row: Fed-EF-SGD. 2nd row: Fed-EF-AMS.

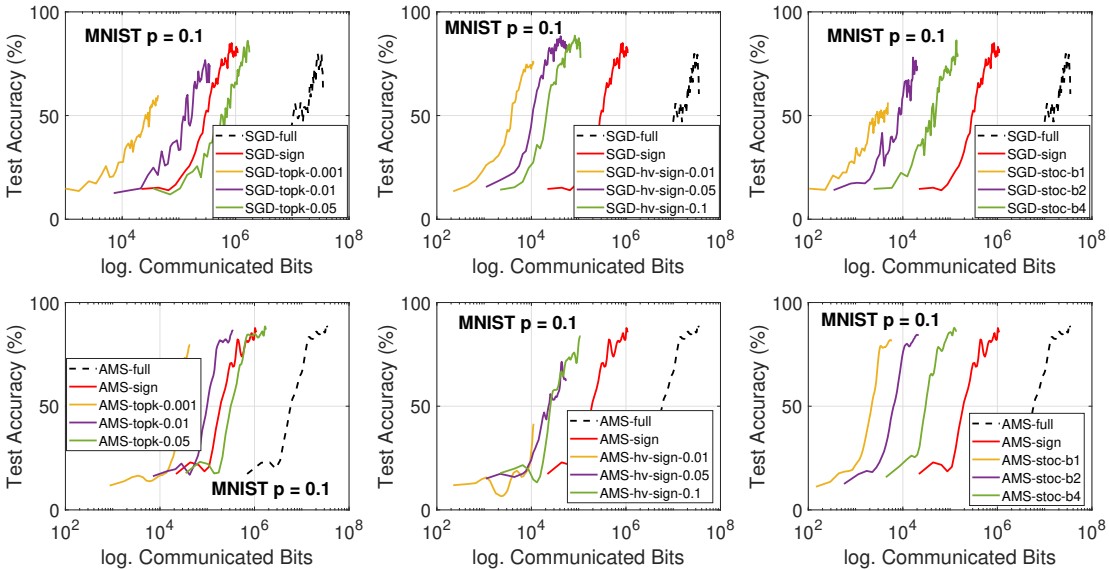

Figure 14: MNIST dataset trained by CNN. Test accuracy of Fed-EF with **TopK**, **Sign** and **heavy-Sign** compressors, and **Stoc** without EF. Participation rate $p = 0.1$, non-iid data. "sign", "topk" and "hv-sign" are applied with Fed-EF, while "Stoc" is the stochastic quantization without EF. 1st row: Fed-EF-SGD. 2nd row: Fed-EF-AMS.

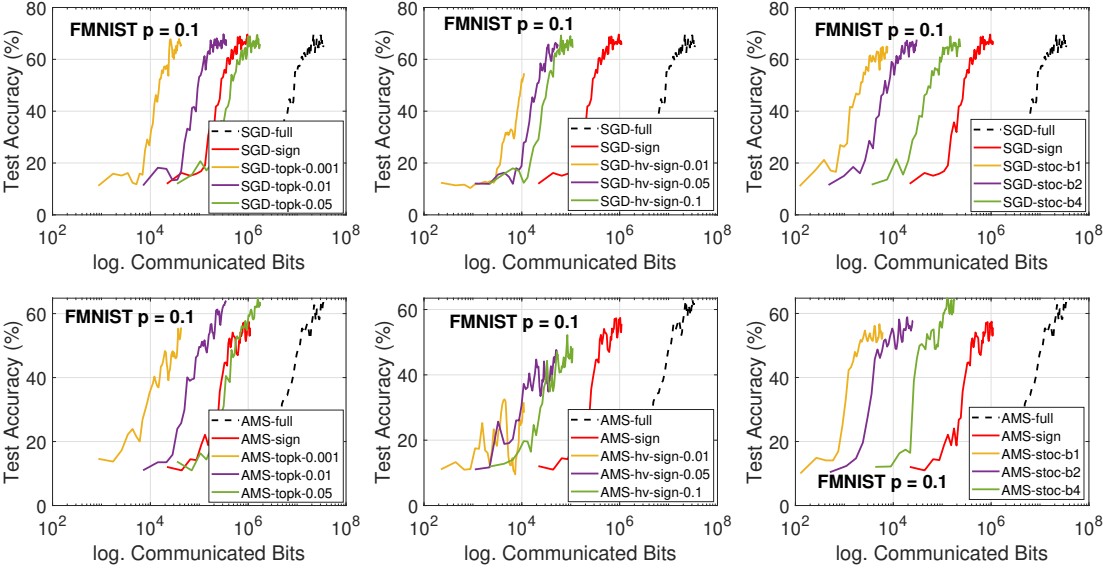

Figure 15: FMNIST dataset trained by CNN. Test accuracy of Fed-EF with **TopK**, **Sign** and **heavy-Sign** compressors, and **Stoc** without EF. Participation rate $p = 0.1$, non-iid data. "sign", "topk" and "hv-sign" are applied with Fed-EF, while "Stoc" is the stochastic quantization without EF. 1st row: Fed-EF-SGD. 2nd row: Fed-EF-AMS.

# B  COMPRESSION DISCREPANCY

In our theoretical analysis for Fed-EF, Assumption 3 is needed, which states that $\mathbb{E}[\|\frac{1}{n}\sum_{i=1}^{n}\mathcal{C}(\Delta_{t,i}+e_{t,i})-\frac{1}{n}\sum_{i=1}^{n}(\Delta_{t,i}+e_{t,i})\|^2] \leq q_{\mathcal{A}}^2\mathbb{E}[\|\frac{1}{n}\sum_{i=1}^{n}(\Delta_{t,i}+e_{t,i})\|^2]$ for some $q_{\mathcal{A}} < 1$ during training. In the following, we justify this assumption to demonstrate how it holds in practice. To study sparsified SGD, [4] also used a similar and stronger (uniform bound instead of in expectation) analytical assumption. As a result, our analysis and theoretical results are also valid under their assumption. Please see more related discussion therein.

## B.1  SIMULATED DATA

We first conduct a simulation to investigate how the two compressors, **TopK** and **Sign**, affect $q_{\mathcal{A}}$. In our presented results, for conciseness we use $n = 5$ clients and model dimensionality $d = 1100$. Similar conclusions hold for much larger $n$ and $d$. We simulate two types of gradients following normal distribution and Laplace distribution (more heavy-tailed), respectively. Examples of the simulated gradients are visualized in Figure 16 and Figure 17. To mimic non-iid data, we assume that each client has some strong signals (large gradients) in some coordinates, and we scale those gradients by a scaling factor $s = 2, 10, 100$. Conceptually, larger $s$ represents higher data heterogeneity.

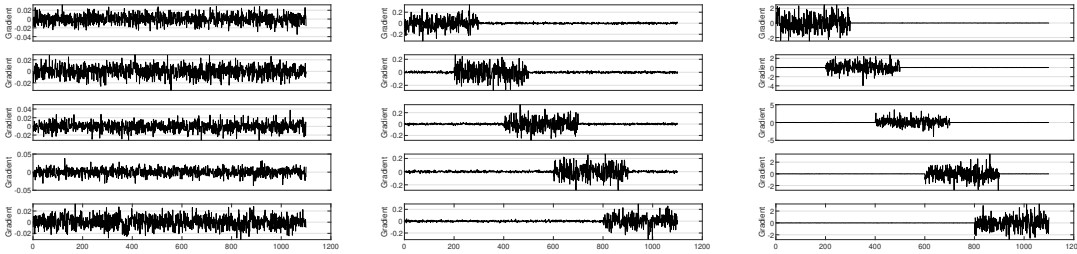

Figure 16: The simulated gradients of 5 heterogeneous clients, from $N(0, \gamma^2)$ with $\gamma = 0.01$. The gradient on each distinct client is scaled by $s = 2, 10, 100$ (left, mid, right), respectively. Larger $s$ implies higher data heterogeneity.

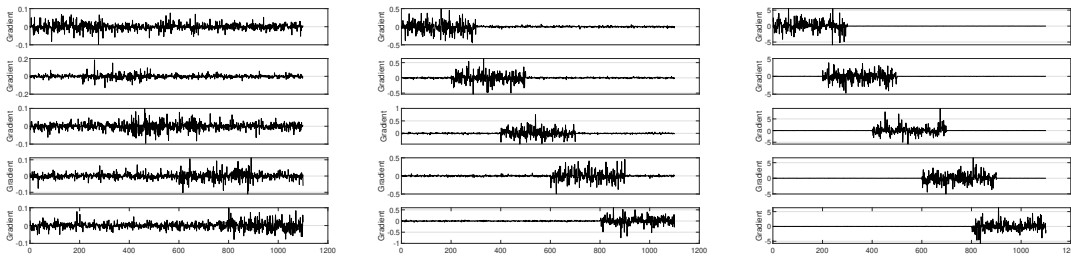

Figure 17: The simulated gradients of 5 heterogeneous clients, from $Lap(0, \lambda)$ with $\lambda = 0.01$. The $x$-axis is the dimension. The gradient on each distinct client is scaled by $s = 2, 10, 100$ (left, mid, right), respectively. Wee the gradients have heavier tail than normal distribution. Larger $s$ implies higher data heterogeneity.

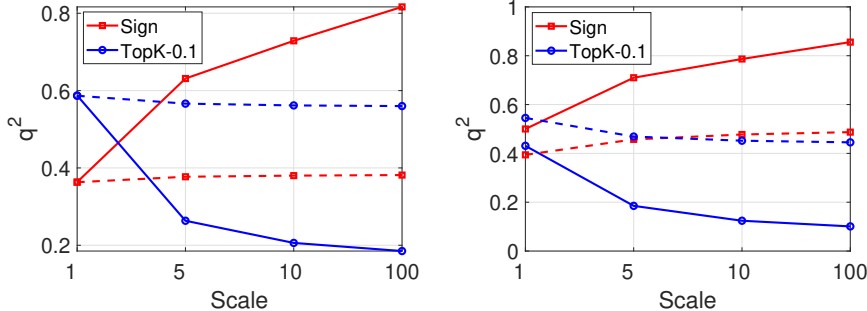

Figure 18: The compression coefficient $q_{\mathcal{A}}$ in Assumption 3 on simulated gradients. **TopK** is applied with sparsity $k = 0.1$. Left: Gaussian distribution. Right: Laplace distribution. $q_{\mathcal{A}}^2$ is computed by $q = \frac{\|\delta(x)-x\|^2}{\|x\|^2}$ where $\delta(x) = \frac{1}{n}\sum_{i=1}^n \mathcal{C}(\Delta_{t,i} + e_{t,i})$ and $x = \frac{1}{n}\sum_{i=1}^n (\Delta_{t,i} + e_{t,i})$. The dashed curves are respectively the compression coefficients $q_{\mathcal{C}}^2$ from Definition 1, which is calculated by replacing $\delta(x) = \mathcal{C}(\frac{1}{n}\sum_{i=1}^n \Delta_{t,i} + e_{t,i})$. We see that in all cases, $q_{\mathcal{A}} < 1$.

We apply the **TopK-0.1** and **Sign** compressor in Definition 1 to the simulated gradients, and compute the averaged $q_{\mathcal{A}}^2$ in Figure 18 over $10^5$ independent runs. The dashed curves are respectively the "ideal" compression coefficients $q_{\mathcal{C}}$ such that $\mathbb{E}[\|\mathcal{C}\left(\frac{1}{n}\sum_{i=1}^n \Delta_{t,i} + e_{t,i}\right) - \frac{1}{n}\sum_{i=1}^n (\Delta_{t,i} + e_{t,i})\|^2] \leq q_{\mathcal{C}}^2 \mathbb{E}[\|\frac{1}{n}\sum_{i=1}^n (\Delta_{t,i} + e_{t,i})\|^2]$ from Definition 1. We see that in all cases, $q_{\mathcal{A}}$ is indeed less than 1. This still holds even when the data heterogeneity increases to as large as 100.

## B.2 REAL-WORLD DATA

We report the empirical $q_{\mathcal{A}}$ values when training CNN on MNIST and FMNIST datasets. The experimental setup is the same as in Section 5. We present the result in Figure 19 with $\eta = 1$, $\eta_l = 0.01$ under the same heterogeneous setting where client data are highly non-iid. The plots for other learning rate combinations and iid data are similar. In particular, we see for both compressors and both datasets, the empirical $q_{\mathcal{A}}$ is well-bounded below 1 throughout the training process.

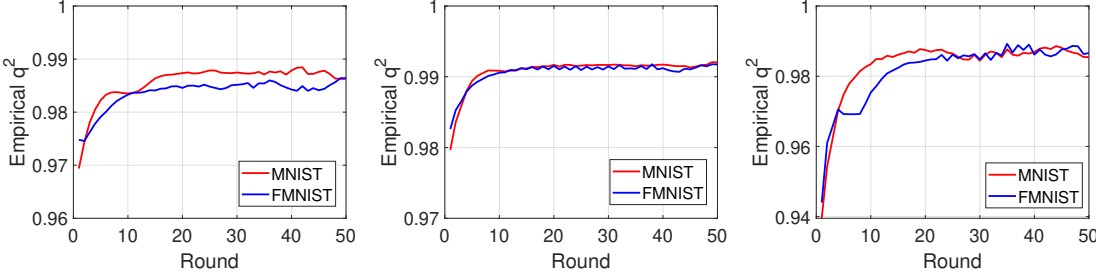

Figure 19: The compression coefficient $q_{\mathcal{A}}$ in Assumption 3 in our experiments (Section 5) for CNN trained on MNIST and FMNIST dataset, averaged over multiple runs. $\eta = 1$, $\eta_l = 0.01$, non-iid client data distribution. Left: **Sign** compression. Mid: **TopK** compression with $k = 0.01$. Right: **TopK** compression with $k = 0.1$.

## C  PROOF OF CONVERGENCE RESULTS

In this section, we provide the proof of the convergence rates of Fed-EF. For illustrative purpose, we first present the proof for the more complicated Fed-EF-AMS in Section C.1, and the proof of Fed-EF-SGD would follow in Section C.2. Section C.3 contains intermediary lemmas and Section C.4 provides the analysis of Fed-EF in partial participation.

### C.1  PROOF OF THEOREM 2: FED-EF-AMS

*Proof.* We first clarify some notations. At round $t$, let the full-precision local model update of the $i$-th worker be $\Delta_{t,i}$, the error accumulator be $e_{t,i}$, and denote $\widetilde{\Delta}_{t,i} = \mathcal{C}(g_{t,i} + e_{t,i})$. Define $\bar{\Delta}_t = \frac{1}{n}\sum_{i=1}^{n}\Delta_{t,i}$, $\overline{\widetilde{\Delta}}_t = \frac{1}{n}\sum_{i=1}^{n}\widetilde{\Delta}_{t,i}$ and $\bar{e}_t = \frac{1}{n}\sum_{i=1}^{n}e_{t,i}$. The second moment computed by the compressed local model updates is denoted as $v_t = \beta_2 v_{t-1} + (1-\beta_2)\overline{\widetilde{\Delta}}_t^2$, and $\hat{v}_t = \max\{\hat{v}_{t-1}, v_t\}$. Also, the first order moving average sequence

$$m_t = \beta_1 m_{t-1} + (1-\beta_1)\overline{\widetilde{\Delta}}_t \quad \text{and} \quad m'_t = \beta_1 m'_{t-1} + (1-\beta_1)\bar{\Delta}_t,$$

where $m'_t$ represents the first moment moving average sequence using the uncompressed local model updates. By construction we have $m'_t = (1-\beta_1)\sum_{\tau=1}^{t}\beta_1^{t-\tau}\bar{\Delta}_\tau$.

Our proof will use the following auxiliary sequences: for round $t = 1, ..., T$,

$$\mathcal{E}_{t+1} := (1-\beta_1)\sum_{\tau=1}^{t+1}\beta_1^{t+1-\tau}\bar{e}_\tau,$$

$$\theta'_{t+1} := \theta_{t+1} - \eta\frac{\mathcal{E}_{t+1}}{\sqrt{\hat{v}_t + \epsilon}}.$$

Then, we can write the evolution of $\theta'_t$ as

$$\begin{aligned}
\theta'_{t+1} &= \theta_{t+1} - \eta\frac{\mathcal{E}_{t+1}}{\sqrt{\hat{v}_t + \epsilon}} \\
&= \theta_t - \eta\frac{(1-\beta_1)\sum_{\tau=1}^{t}\beta_1^{t-\tau}\overline{\widetilde{\Delta}}_\tau + (1-\beta_1)\sum_{\tau=1}^{t+1}\beta_1^{t+1-\tau}\bar{e}_\tau}{\sqrt{\hat{v}_t + \epsilon}} \\
&= \theta_t - \eta\frac{(1-\beta_1)\sum_{\tau=1}^{t}\beta_1^{t-\tau}(\overline{\widetilde{\Delta}}_\tau + \bar{e}_{\tau+1}) + (1-\beta)\beta_1^t\bar{e}_1}{\sqrt{\hat{v}_t + \epsilon}} \\
&= \theta_t - \eta\frac{(1-\beta_1)\sum_{\tau=1}^{t}\beta_1^{t-\tau}\bar{e}_\tau}{\sqrt{\hat{v}_t + \epsilon}} - \eta\frac{m'_t}{\sqrt{\hat{v}_t + \epsilon}} \\
&= \theta_t - \eta\frac{\mathcal{E}_t}{\sqrt{\hat{v}_{t-1} + \epsilon}} - \eta\frac{m'_t}{\sqrt{\hat{v}_t + \epsilon}} + \eta(\frac{1}{\sqrt{\hat{v}_{t-1} + \epsilon}} - \frac{1}{\sqrt{\hat{v}_t + \epsilon}})\mathcal{E}_t \\
&\overset{(a)}{=} \theta'_t - \eta\frac{m'_t}{\sqrt{\hat{v}_t + \epsilon}} + \eta(\frac{1}{\sqrt{\hat{v}_{t-1} + \epsilon}} - \frac{1}{\sqrt{\hat{v}_t + \epsilon}})\mathcal{E}_t \\
&:= \theta'_t - \eta\frac{m'_t}{\sqrt{\hat{v}_t + \epsilon}} + \eta D_t\mathcal{E}_t,
\end{aligned}$$

where (a) uses the fact of error feedback that for every $i \in [n]$, $\widetilde{\Delta}_{t,i} + e_{t+1,i} = \Delta_{t,i} + e_{t,i}$, and $e_{t,1} = 0$ at initialization. Further define the virtual iterates:

$$x_{t+1} := \theta'_{t+1} - \eta\frac{\beta_1}{1-\beta_1}\frac{m'_t}{\sqrt{\hat{v}_t + \epsilon}},$$

which follows the recurrence:

$$
\begin{aligned}
x_{t+1} &= \theta'_{t+1} - \eta \frac{\beta_1}{1-\beta_1} \frac{m'_t}{\sqrt{\hat{v}_t + \epsilon}} \\
&= \theta'_t - \eta \frac{m'_t}{\sqrt{\hat{v}_t + \epsilon}} - \eta \frac{\beta_1}{1-\beta_1} \frac{m'_t}{\sqrt{\hat{v}_t + \epsilon}} + \eta D_t \mathcal{E}_t \\
&= \theta'_t - \eta \frac{\beta_1 m'_{t-1} + (1-\beta_1)\bar{\Delta}_t + \frac{\beta_1^2}{1-\beta_1}m'_{t-1} + \beta_1 \bar{\Delta}_t}{\sqrt{\hat{v}_t + \epsilon}} + \eta D_t \mathcal{E}_t \\
&= \theta'_t - \eta \frac{\beta_1}{1-\beta_1} \frac{m'_{t-1}}{\sqrt{\hat{v}_t + \epsilon}} - \eta \frac{\bar{\Delta}_t}{\sqrt{\hat{v}_t + \epsilon}} + \eta D_t \mathcal{E}_t \\
&= x_t - \eta \frac{\bar{\Delta}_t}{\sqrt{\hat{v}_t + \epsilon}} + \eta \frac{\beta_1}{1-\beta_1} D_t m'_{t-1} + \eta D_t \mathcal{E}_t.
\end{aligned}
$$

The general idea is to study the convergence of the sequence $x_t$, and show that the difference between $x_t$ and $\theta_t$ (of interest) is small. First, by the smoothness Assumption 1, we have

$$
f(x_{t+1}) \leq f(x_t) + \langle \nabla f(x_t), x_{t+1} - x_t \rangle + \frac{L}{2} \|x_{t+1} - x_t\|^2.
$$

Taking expectation w.r.t. the randomness at round $t$ and re-arranging terms, we obtain

$$
\begin{aligned}
&\mathbb{E}[f(x_{t+1})] - f(x_t) \\
&\leq -\eta \mathbb{E}\Big[\langle \nabla f(x_t), \frac{\bar{\Delta}_t}{\sqrt{\hat{v}_t + \epsilon}} \rangle\Big] + \eta \mathbb{E}\Big[\langle \nabla f(x_t), \frac{\beta_1}{1-\beta_1} D_t m'_{t-1} + D_t \mathcal{E}_t \rangle\Big] \\
&\qquad\qquad + \frac{\eta^2 L}{2} \mathbb{E}\Big[\|\frac{\bar{\Delta}_t}{\sqrt{\hat{v}_t + \epsilon}} - \frac{\beta_1}{1-\beta_1} D_t m'_{t-1} - D_t \mathcal{E}_t\|^2\Big] \\
&= \underbrace{-\eta \mathbb{E}\Big[\langle \nabla f(\theta_t), \frac{\bar{\Delta}_t}{\sqrt{\hat{v}_t + \epsilon}} \rangle\Big]}_{I} + \underbrace{\eta \mathbb{E}\Big[\langle \nabla f(x_t), \frac{\beta_1}{1-\beta_1} D_t m'_{t-1} + D_t \mathcal{E}_t \rangle\Big]}_{II} \\
&+ \underbrace{\frac{\eta^2 L}{2} \mathbb{E}\Big[\|\frac{\bar{\Delta}_t}{\sqrt{\hat{v}_t + \epsilon}} - \frac{\beta_1}{1-\beta_1} D_t m'_{t-1} - D_t \mathcal{E}_t\|^2\Big]}_{III} + \underbrace{\eta \mathbb{E}\Big[\langle \nabla f(\theta_t) - \nabla f(x_t), \frac{\bar{\Delta}_t}{\sqrt{\hat{v}_t + \epsilon}} \rangle\Big]}_{IV},
\end{aligned}
\tag{3}
$$

**Bounding term I.** We have

$$
\begin{aligned}
\boldsymbol{I} &= -\eta \mathbb{E}\Big[\langle \nabla f(\theta_t), \frac{\bar{\Delta}_t}{\sqrt{\hat{v}_{t-1} + \epsilon}} \rangle\Big] - \eta \mathbb{E}\Big[\langle \nabla f(\theta_t), (\frac{1}{\sqrt{\hat{v}_t + \epsilon}} - \frac{1}{\sqrt{\hat{v}_{t-1} + \epsilon}})\bar{\Delta}_t \rangle\Big] \\
&\leq -\eta \mathbb{E}\Big[\langle \nabla f(\theta_t), \frac{\bar{\Delta}_t}{\sqrt{\hat{v}_{t-1} + \epsilon}} \rangle\Big] + \eta \eta_l K G^2 \mathbb{E}[\|D_t\|_1],
\end{aligned}
\tag{4}
$$

where we use Assumption 4 on the stochastic gradient magnitude. The last inequality holds by simply bounding the aggregated local model update by

$$
\|\bar{\Delta}_t\| \leq \frac{1}{n} \sum_{i=1}^{n} \|\eta_l \sum_{k=1}^{K} g_{t,i}^{(k)}\| \leq \eta_l K G,
$$

and the fact that for any vector in $\mathbb{R}^d$, the $l_2$ norm is upper bounded by the $l_1$ norm.

Regarding the first term in (4), we have

$$
-\eta \mathbb{E}\Big[\langle \nabla f(\theta_t), \frac{\bar{\Delta}_t}{\sqrt{\hat{v}_{t-1}+\epsilon}}\rangle\Big]
$$

$$
= -\eta \mathbb{E}\Big[\langle \frac{\nabla f(\theta_t)}{\sqrt{\hat{v}_{t-1}+\epsilon}}, \bar{\Delta}_t - \eta_l K \nabla f(\theta_t) + \eta_l K \nabla f(\theta_t)\rangle\Big]
$$

$$
= -\eta\eta_l K \mathbb{E}\Big[\frac{\|\nabla f(\theta_t)\|^2}{\sqrt{\hat{v}_{t-1}+\epsilon}}\Big] + \eta \mathbb{E}\Big[\langle \frac{\nabla f(\theta_t)}{\sqrt{\hat{v}_{t-1}+\epsilon}}, -\bar{\Delta}_t + \eta_l K \nabla f(\theta_t)\rangle\Big]
$$

$$
\overset{(a)}{\leq} -\frac{\eta\eta_l K}{\sqrt{\frac{4\eta_l^2(1+q^2)^3 K^2}{(1-q^2)^2}G^2+\epsilon}}\mathbb{E}\big[\|\nabla f(\theta_t)\|^2\big] + \eta\langle \frac{\nabla f(\theta_t)}{\sqrt{\hat{v}_{t-1}+\epsilon}}, \mathbb{E}\big[-\frac{1}{n}\sum_{i=1}^{n}\sum_{k=1}^{K}\eta_l g_{t,i}^{(k)} + \eta_l K \nabla f(\theta_t)\big]\rangle
$$

$$
\overset{(b)}{=} -\frac{\eta\eta_l K}{\sqrt{\frac{4\eta_l^2(1+q^2)^3 K^2}{(1-q^2)^2}G^2+\epsilon}}\mathbb{E}\big[\|\nabla f(\theta_t)\|^2\big]
$$

$$
+ \eta\underbrace{\langle \frac{\sqrt{\eta_l}\nabla f(\theta_t)}{(\hat{v}_{t-1}+\epsilon)^{1/4}}, \mathbb{E}\Big[\frac{\sqrt{\eta_l}}{n(\hat{v}_{t-1}+\epsilon)^{1/4}}(-\sum_{i=1}^{n}\sum_{k=1}^{K}\nabla f_i(\theta_{t,i}^{(k)}) + K\nabla f(\theta_t))\Big]\rangle}_{V},
$$

where (a) uses Lemma C.6 and (b) is due to Assumption 2 that $g_{t,i}^{(k)}$ is an unbiased estimator of $\nabla f_i(\theta_{t,i}^{(k)})$. To bound term V, we note the inequality that $\langle a, b\rangle \leq \frac{\alpha}{2}a^2 + \frac{1}{2\alpha}b^2$ for any $a, b \in \mathbb{R}$ and $\alpha > 0$. Therefore, we have

$$
V \leq \frac{\eta_l K}{2\sqrt{\epsilon}}\mathbb{E}\big[\|\nabla f(\theta_t)\|^2\big] + \frac{\eta_l}{2K\sqrt{\epsilon}}\mathbb{E}\big[\|\frac{1}{n}\sum_{i=1}^{n}\sum_{k=1}^{K}(\nabla f_i(\theta_{t,i}^{(k)}) - \nabla f_i(\theta_t))\|^2\big]
$$

$$
\leq \frac{\eta_l K}{2\sqrt{\epsilon}}\mathbb{E}\big[\|\nabla f(\theta_t)\|^2\big] + \frac{\eta_l}{2nK\sqrt{\epsilon}}\mathbb{E}\big[\sum_{i=1}^{n}\|\sum_{k=1}^{K}(\nabla f_i(\theta_{t,i}^{(k)}) - \nabla f_i(\theta_t))\|^2\big]
$$

$$
\leq \frac{\eta_l K}{2\sqrt{\epsilon}}\mathbb{E}\big[\|\nabla f(\theta_t)\|^2\big] + \frac{\eta_l}{2n\sqrt{\epsilon}}\mathbb{E}\big[\sum_{i=1}^{n}\sum_{k=1}^{K}\|\nabla f_i(\theta_{t,i}^{(k)}) - \nabla f_i(\theta_t)\|^2\big]
$$

$$
\leq \frac{\eta_l K}{2\sqrt{\epsilon}}\mathbb{E}\big[\|\nabla f(\theta_t)\|^2\big] + \frac{\eta_l L^2}{2n\sqrt{\epsilon}}\mathbb{E}\big[\sum_{i=1}^{n}\sum_{k=1}^{K}\|\theta_{t,i}^{(k)} - \theta_t\|^2\big],
$$

where the last inequality is a result of the $L$-smoothness assumption on the loss function $f_i(x)$. Applying Lemma C.1 to the consensus error, we can further bound term $V$ by

$$
V \leq \frac{\eta_l K}{2\sqrt{\epsilon}}\mathbb{E}\big[\|\nabla f(\theta_t)\|^2\big] + \frac{\eta_l K L^2}{2\sqrt{\epsilon}}\big[5\eta_l^2 K(\sigma^2 + 6K\sigma_g^2) + 30\eta_l^2 K^2\mathbb{E}[\|\nabla f(\theta_t)\|^2]\big]
$$

$$
\leq \frac{47\eta_l K}{64\sqrt{\epsilon}}\mathbb{E}\big[\|\nabla f(\theta_t)\|^2\big] + \frac{5\eta_l^3 K^2 L^2}{2\sqrt{\epsilon}}(\sigma^2 + 6K\sigma_g^2),
$$

when we choose $\eta_l \leq \frac{1}{8KL}$. Further, if we set $\eta_l \leq \frac{\sqrt{15}(1-q^2)\sqrt{\epsilon}}{14(1+q^2)^{1.5}KG}$, we have

$$
\frac{4\eta_l^2(1+q^2)^3 K^2}{(1-q^2)^2}G^2 + \epsilon \leq \frac{60}{196}\epsilon + \epsilon = \frac{64}{49}\epsilon.
$$

Hence, we can establish from (4) that

$$
I \leq -\frac{\eta\eta_l K}{8\sqrt{\epsilon}}\mathbb{E}\big[\|\nabla f(\theta_t)\|^2\big] + \frac{5\eta\eta_l^3 K^2 L^2}{2\sqrt{\epsilon}}(\sigma^2 + 6K\sigma_g^2) + \eta\eta_l KG^2\mathbb{E}[\|D_t\|_1]. \tag{5}
$$

**Bounding term II.** By Lemma C.5, we know that $\|\mathcal{E}_t\| \leq \frac{2\eta_l qKG}{1-q^2}$, and by Lemma C.3, $\|m'_t\| \leq \eta_l KG$. Thus, we have

$$
\begin{aligned}
\boldsymbol{II} &\leq \eta\Big(\mathbb{E}\big[\langle\nabla f(\theta_t), \frac{\beta_1}{1-\beta_1}D_t m'_{t-1} + D_t\mathcal{E}_t\rangle\big] + \mathbb{E}\big[\langle\nabla f(x_t) - \nabla f(\theta_t), \frac{\beta_1}{1-\beta_1}D_t m'_{t-1} + D_t\mathcal{E}_t\rangle\big]\Big) \\
&\leq \eta\mathbb{E}\big[\|\nabla f(\theta_t)\|\|\frac{\beta_1}{1-\beta_1}D_t m'_{t-1} + D_t\mathcal{E}_t\|\big] \\
&\qquad + \eta^2\,L\mathbb{E}\big[\|\frac{\frac{\beta_1}{1-\beta_1}m'_{t-1} + \mathcal{E}_t}{\sqrt{\hat{v}_{t-1} + \epsilon}}\|\|\frac{\beta_1}{1-\beta_1}D_t m'_{t-1} + D_t\mathcal{E}_t\|\big] \\
&\leq \eta\eta_l C_1 KG^2\mathbb{E}[\|D_t\|_1] + \frac{\eta^2\eta_l^2 C_1^2 LK^2 G^2}{\sqrt{\epsilon}}\mathbb{E}[\|D_t\|_1], \qquad\qquad (6)
\end{aligned}
$$

where $C_1 := \frac{\beta_1}{1-\beta_1} + \frac{2q}{1-q^2}$, and the second inequality is due to the smoothness of $f(\theta)$.

**Bounding term III.** This term can be bounded as follows:

$$
\begin{aligned}
\boldsymbol{III} &\leq \eta^2 L\mathbb{E}\big[\|\frac{\bar{\Delta}_t}{\sqrt{\hat{v}_t + \epsilon}}\|^2\big] + \eta^2 L\mathbb{E}\big[\|\frac{\beta_1}{1-\beta_1}D_t m'_{t-1} - D_t\mathcal{E}_t\|^2\big] \\
&\leq \frac{\eta^2 L}{\epsilon}\mathbb{E}\big[\|\bar{\Delta}_t\|^2\big] + \eta^2 L\mathbb{E}\big[\|D_t(\frac{\beta_1}{1-\beta_1}m'_{t-1} - \mathcal{E}_t)\|^2\big] \\
&\leq \frac{\eta^2 L(2\eta_l^2 K^2 + 120\eta_l^4 K^4 L^2)}{\epsilon}\mathbb{E}\big[\|\nabla f(\theta_t)\|^2\big] + \frac{4\eta^2\eta_l^2 KL}{n\epsilon}\sigma^2 \\
&\qquad + \frac{20\eta^2\eta_l^4 K^3 L^3}{\epsilon}(\sigma^2 + 6K\sigma_g^2) + \eta^2\eta_l^2 C_1^2 LK^2 G^2\mathbb{E}[\|D_t\|^2], \qquad (7)
\end{aligned}
$$

where we apply Lemma C.2 and use similar argument as in bounding term II.

**Bounding term IV.** Lastly, for term IV, we have for some $\rho > 0$,

$$
\boldsymbol{IV} \qquad\qquad\qquad\qquad\qquad\qquad\qquad\qquad\qquad\qquad\qquad\qquad\qquad (8)
$$

$$
\begin{aligned}
&= \eta\mathbb{E}\Big[\langle\nabla f(\theta_t) - \nabla f(x_t), \frac{\bar{\Delta}_t}{\sqrt{\hat{v}_{t-1} + \epsilon}}\rangle\Big] + \eta\mathbb{E}\Big[\langle\nabla f(\theta_t) - \nabla f(x_t), (\frac{1}{\sqrt{\hat{v}_t + \epsilon}} - \frac{1}{\sqrt{\hat{v}_{t-1} + \epsilon}})\bar{\Delta}_t\rangle\Big] \\
&\overset{(a)}{\leq} \frac{\eta\rho}{2\epsilon}\mathbb{E}\big[\|\bar{\Delta}_t\|^2\big] + \frac{\eta}{2\rho}\mathbb{E}\big[\|\nabla f(\theta_t) - \nabla f(x_t)\|^2\big] + \eta^2 L\mathbb{E}\Big[\|\frac{\frac{\beta_1}{1-\beta_1}m'_{t-1} + \mathcal{E}_t}{\sqrt{\hat{v}_{t-1} + \epsilon}}\|\|D_t\Delta_t\|\Big] \\
&\overset{(b)}{\leq} \frac{\rho\eta(\eta_l^2 K^2 + 60\eta_l^4 K^4 L^2)}{\epsilon}\mathbb{E}\big[\|\nabla f(\theta_t)\|^2\big] + \frac{2\rho\eta\eta_l^2 K}{\epsilon n}\sigma^2 + \frac{10\rho\eta\eta_l^4 K^3 L^2}{\epsilon}(\sigma^2 + 6K\sigma_g^2) \\
&\qquad + \frac{\eta^3 L^2}{2\rho}\mathbb{E}\Big[\|\frac{\frac{\beta_1}{1-\beta_1}m'_{t-1} + \mathcal{E}_t}{\sqrt{\hat{v}_{t-1} + \epsilon}}\|^2\Big] + \frac{\eta^2\eta_l^2 C_1 LK^2 G^2}{\sqrt{\epsilon}}\mathbb{E}[\|D_t\|] \\
&\leq \frac{\rho\eta\eta_l^2 K^2(60\eta_l^2 K^2 L^2 + 1)}{\epsilon}\mathbb{E}\big[\|\nabla f(\theta_t)\|^2\big] + \frac{2\rho\eta\eta_l^2 K}{\epsilon n}\sigma^2 + \frac{10\rho\eta\eta_l^4 K^3 L^2}{\epsilon}(\sigma^2 + 6K\sigma_g^2) \\
&\qquad + \frac{\eta^3 L^2}{\rho\epsilon}\Big[\frac{\beta_1^2}{(1-\beta_1)^2}\mathbb{E}\big[\|m'_t\|^2\big] + \mathbb{E}\big[\|\mathcal{E}_t\|^2\big]\Big] + \frac{\eta^2\eta_l^2 C_1 LK^2 G^2}{\sqrt{\epsilon}}\mathbb{E}[\|D_t\|_1], \qquad (9)
\end{aligned}
$$

where (a) is a consequence of Young's inequality ($\rho$ will be specified later) and the smoothness Assumption 1, and (b) is based on Lemma C.2.

Now that we have bounded all four terms in (3), the next step is to gather the ingredients by taking the telescope summation over $t = 1, ..., T$. Before moving on, for the ease of presentation, we first do this for the third term in (9). For this term, according to Lemma C.3 and Lemma C.5, summing

over $t = 1, ..., T$, we conclude

$$\sum_{t=1}^{T} \frac{\eta^3 L^2}{\rho \epsilon} \Big[ \frac{\beta_1^2}{(1-\beta_1)^2} \mathbb{E}[\|m_t'\|^2] + \mathbb{E}[\|\mathcal{E}_t\|^2] \Big]$$

$$\leq \frac{\eta^3 \beta_1^2 L^2}{\rho(1-\beta_1)^2 \epsilon} \Big[ 2\eta_l^2 K^2 (60\eta_l^2 K^2 L^2 + 1) \sum_{t=1}^{T} \mathbb{E}\big[\|\nabla f(\theta_t)\|^2\big] + 4\frac{T\eta_l^2 K}{n} \sigma^2 + 20T\eta_l^4 K^3 L^2 (\sigma^2 + 6K\sigma_g^2) \Big]$$

$$+ \frac{\eta^3 q^2 L^2}{\rho(1-q^2)^2 \epsilon} \Big[ 8\eta_l^2 K^2 (60\eta_l^2 K^2 L^2 + 1) \sum_{t=1}^{T} \mathbb{E}\big[\|\nabla f(\theta_\tau)\|^2\big] + \frac{16T\eta_l^2 K}{n} \sigma^2 + 80T\eta_l^4 K^3 L^2 (\sigma^2 + 6K\sigma_g^2) \Big]$$

$$\leq \frac{2\eta^3 \eta_l^2 C_2 K^2 L^2}{\rho \epsilon} (60\eta_l^2 K^2 L^2 + 1) \sum_{t=1}^{T} \mathbb{E}\big[\|\nabla f(\theta_t)\|^2\big]$$

$$+ \frac{4T\eta^3 \eta_l^2 C_2 K L^2}{\rho n \epsilon} \sigma^2 + \frac{20T\eta^3 \eta_l^4 C_2 K^3 L^4}{\rho \epsilon} (\sigma^2 + 6K\sigma_g^2), \tag{10}$$

with $C_2 := \frac{\beta_1^2}{(1-\beta_1)^2} + \frac{4q^2}{(1-q^2)^2}$.

**Putting together.** We are in the position to combine pieces together to get our final result by integrating (5), (6), (7), (9) and (10) into (3) and taking the telescoping sum over $t = 1, ..., T$. After re-arranging terms, when $\eta_l \leq \min\left\{ \frac{1}{8KL}, \frac{(1-q^2)\sqrt{\epsilon}}{4(1+q^2)^{1.5} KG} \right\}$, we have

$$\mathbb{E}[f(x_{T+1}) - f(x_1)]$$

$$\leq -\frac{\eta \eta_l K}{8\sqrt{\epsilon}} \sum_{t=1}^{T} \mathbb{E}\big[\|\nabla f(\theta_t)\|^2\big] + \frac{5T\eta \eta_l^3 K^2 L^2}{2\sqrt{\epsilon}} (\sigma^2 + 6K\sigma_g^2) + \eta \eta_l K G^2 \sum_{t=1}^{T} \mathbb{E}[\|D_t\|_1]$$

$$+ \eta \eta_l C_1 K G^2 \sum_{t=1}^{T} \mathbb{E}[\|D_t\|_1] + \frac{\eta^2 \eta_l^2 C_1^2 L K^2 G^2}{\sqrt{\epsilon}} \sum_{t=1}^{T} \mathbb{E}[\|D_t\|_1]$$

$$+ \frac{\eta^2 L (2\eta_l^2 K^2 + 120\eta_l^4 K^4 L^2)}{\epsilon} \sum_{t=1}^{T} \mathbb{E}\big[\|\nabla f(\theta_t)\|^2\big] + \frac{4T\eta^2 \eta_l^2 K L}{n \epsilon} \sigma^2$$

$$+ \frac{20T\eta^2 \eta_l^4 K^3 L^3}{\epsilon} (\sigma^2 + 6K\sigma_g^2) + \eta^2 \eta_l^2 C_1^2 L K^2 G^2 \sum_{t=1}^{T} \mathbb{E}[\|D_t\|^2]$$

$$+ \frac{\rho \eta \eta_l^2 K^2 (60\eta_l^2 K^2 L^2 + 1)}{\epsilon} \sum_{t=1}^{T} \mathbb{E}\big[\|\nabla f(\theta_t)\|^2\big] + \frac{2T\rho \eta \eta_l^2 K}{\epsilon n} \sigma^2 + \frac{10T\rho \eta \eta_l^4 K^3 L^2}{\epsilon} (\sigma^2 + 6K\sigma_g^2)$$

$$+ \frac{2\eta^3 \eta_l^2 C_2 K^2 L^2}{\rho \epsilon} (60\eta_l^2 K^2 L^2 + 1) \sum_{t=1}^{T} \mathbb{E}\big[\|\nabla f(\theta_t)\|^2\big] + \frac{4T\eta^3 \eta_l^2 C_2 K L^2}{\rho n \epsilon} \sigma^2$$

$$+ \frac{20T\eta^3 \eta_l^4 C_2 K^3 L^4}{\rho \epsilon} (\sigma^2 + 6K\sigma_g^2) + \frac{\eta^2 \eta_l^2 C_1 L K^2 G^2}{\sqrt{\epsilon}} \sum_{t=1}^{T} \mathbb{E}[\|D_t\|_1]$$

$$= \Upsilon_1 \cdot \sum_{t=1}^{T} \mathbb{E}\big[\|\nabla f(\theta_t)\|^2\big] + \Upsilon_2 \cdot (\sigma^2 + 6K\sigma_g^2) + \Upsilon_3 \cdot \sigma^2$$

$$+ \Upsilon_4 \cdot \sum_{t=1}^{T} \mathbb{E}[\|D_t\|_1] + \eta^2 \eta_l^2 C_1^2 L K^2 G^2 \sum_{t=1}^{T} \mathbb{E}[\|D_t\|^2], \tag{11}$$

where

$$\Upsilon_1 = -\frac{\eta\eta_l K}{8\sqrt{\epsilon}} + \frac{\eta^2 L(2\eta_l^2 K^2 + 120\eta_l^4 K^4 L^2)}{\epsilon}$$

$$+ \frac{\rho\eta\eta_l^2 K^2(60\eta_l^2 K^2 L^2 + 1)}{\epsilon} + \frac{2\eta^3\eta_l^2 C_2 K^2 L^2}{\rho\epsilon}(60\eta_l^2 K^2 L^2 + 1)$$

$$\leq -\frac{\eta\eta_l K}{8\sqrt{\epsilon}} + \frac{2\eta^2\eta_l^2 K^2 L}{\epsilon} + \frac{120\eta^2\eta_l^4 K^4 L^3}{\epsilon} + \frac{2\rho\eta\eta_l^2 K^2}{\epsilon} + \frac{4\eta^3\eta_l^2 C_2 K^2 L^2}{\rho\epsilon}, \tag{12}$$

$$\Upsilon_2 = \frac{5T\eta\eta_l^3 K^2 L^2}{2\sqrt{\epsilon}} + \frac{20T\eta^2\eta_l^4 K^3 L^3}{\epsilon} + \frac{10T\rho\eta\eta_l^4 K^3 L^2}{\epsilon} + \frac{20T\eta^3\eta_l^4 C_2 K^3 L^4}{\rho\epsilon},$$

$$\Upsilon_3 = \frac{4T\eta^2\eta_l^2 KL}{n\epsilon} + \frac{2T\rho\eta\eta_l^2 K}{n\epsilon} + \frac{4T\eta^3\eta_l^2 C_2 KL^2}{\rho n\epsilon},$$

$$\Upsilon_4 = \eta\eta_l(C_1 + 1)KG^2 + \frac{\eta^2\eta_l^2 C_1^2 LK^2 G^2}{\sqrt{\epsilon}} + \frac{\eta^2\eta_l^2 C_1 LK^2 G^2}{\sqrt{\epsilon}},$$

where to bound $\Upsilon_1$ we use the fact that $\eta_l \leq \frac{1}{8KL}$. We now look at the upper bound (12) of $\Upsilon_1$ which contains 5 terms. In the following, we choose $\rho \equiv L\eta$ in (9) and (10). Suppose we choose $\epsilon < 1$. Then, when the local learning rate satisfies

$$\eta_l \leq \frac{1}{K}\min\left\{\frac{1}{8L}, \frac{(1-q^2)\sqrt{\epsilon}}{4(1+q^2)^{1.5}G}, \frac{\sqrt{\epsilon}}{128\eta L}, \frac{\sqrt{\epsilon}}{256C_2\eta L}, \left(\frac{\sqrt{\epsilon}}{7680\eta}\right)^{1/3}\frac{1}{L}\right\}$$

$$\leq \frac{\sqrt{\epsilon}}{8KL}\min\left\{\frac{1}{\sqrt{\epsilon}}, \frac{2(1-q^2)L}{(1+q^2)^{1.5}G}, \frac{1}{\max\{16, 32C_2\}\eta}, \frac{1}{3\eta^{1/3}}\right\},$$

each of the last four terms can be bounded by $\frac{\eta\eta_l K}{48\sqrt{\epsilon}}$. Thus, under this learning rate setting,

$$\Upsilon_1 \leq -\frac{\eta\eta_l K}{16\sqrt{\epsilon}}.$$

Taking the above into (11), we arrive at

$$\frac{\eta\eta_l K}{16\sqrt{\epsilon}}\sum_{t=1}^{T}\mathbb{E}\big[\|\nabla f(\theta_t)\|^2\big] \leq f(x_1) - \mathbb{E}[f(x_{T+1})] + \Upsilon_2 \cdot (\sigma^2 + 6K\sigma_g^2)$$

$$+ \Upsilon_3 \cdot \sigma^2 + \Upsilon_4 \cdot \frac{d}{\sqrt{\epsilon}} + \frac{\eta^2\eta_l^2 C_1^2 LK^2 G^2 d}{\epsilon},$$

where Lemma C.7 on the difference sequence $D_t$ is applied. Consequently, we have

$$\frac{1}{T}\sum_{t=1}^{T}\mathbb{E}\big[\|\nabla f(\theta_t)\|^2\big] \lesssim \frac{f(x_1) - \mathbb{E}[f(x_{T+1})]}{\eta\eta_l TK} + \widetilde{\Upsilon}_2 \cdot (\sigma^2 + 6K\sigma_g^2) + \widetilde{\Upsilon}_3 \cdot \sigma^2$$

$$+ \frac{(C_1 + 1)G^2 d}{T\sqrt{\epsilon}} + \frac{2\eta\eta_l C_1^2 LKG^2 d}{T\epsilon} + \frac{\eta\eta_l C_1^2 LKG^2 d}{T\epsilon}$$

$$\leq \frac{f(x_1) - \mathbb{E}[f(x_{T+1})]}{\eta\eta_l TK} + \widetilde{\Upsilon}_2 \cdot (\sigma^2 + 6K\sigma_g^2) + \widetilde{\Upsilon}_3 \cdot \sigma^2$$

$$+ \frac{(C_1 + 1)G^2 d}{T\sqrt{\epsilon}} + \frac{3\eta\eta_l C_1^2 LKG^2 d}{T\epsilon},$$

where we make simplification at the second inequality using the fact that $C_1 \leq C_1^2$ since $C_1 \geq 1$. Moreover, $\widetilde{\Upsilon}_2$ and $\widetilde{\Upsilon}_3$ is defined as (recall that we have chosen $\rho \equiv L\eta$)

$$\widetilde{\Upsilon}_2 = \frac{5\eta_l^2 KL^2}{2\sqrt{\epsilon}} + \frac{20\eta\eta_l^3 K^2 L^3}{\epsilon} + \frac{10\eta\eta_l^3 K^2 L^3}{\epsilon} + \frac{20\eta\eta_l^3 C_2 K^2 L^3}{\epsilon}$$

$$\leq \frac{5\eta_l^2 KL^2}{2\sqrt{\epsilon}} + \frac{\eta\eta_l^3(30 + 20C_2)K^2 L^3}{\epsilon},$$

$$\widetilde{\Upsilon}_3 = \frac{4\eta\eta_l L}{n\epsilon} + \frac{2\eta\eta_l L}{n\epsilon} + \frac{4\eta\eta_l C_2 L}{n\epsilon} \leq \frac{\eta\eta_l L(6 + 4C_2)}{n\epsilon}.$$

Finally, to connect the virtual iterates $x_t$ with the actual iterates $\theta_t$, note that $x_1 = \theta_1$, and $f(x_{T+1}) \geq f(\theta^*)$ since $\theta^* = \arg\min_\theta f(\theta)$. Replacing $\widetilde{\Upsilon}_2$ and $\widetilde{\Upsilon}_3$ with above upper bounds, this eventually leads to the bound

$$\frac{1}{T}\sum_{t=1}^{T}\mathbb{E}\big[\|\nabla f(\theta_t)\|^2\big] \lesssim \frac{f(\theta_1) - f(\theta^*)}{\eta\eta_l TK} + \left[\frac{5\eta_l^2 KL^2}{2\sqrt{\epsilon}} + \frac{\eta\eta_l^3(30 + 20C_1^2)K^2 L^3}{\epsilon}\right](\sigma^2 + 6K\sigma_g^2)$$
$$+ \frac{\eta\eta_l L(6 + 4C_1^2)}{n\epsilon}\sigma^2 + \frac{(C_1 + 1)G^2 d}{T\sqrt{\epsilon}} + \frac{3\eta\eta_l C_1^2 LKG^2 d}{T\epsilon},$$

which gives the desired result. Here we use the fact that $C_2 \leq C_1^2$. This completes the proof. $\qquad\square$

## C.2 PROOF OF THEOREM 1: FED-EF-SGD

*Proof.* Now, we prove the variant of Fed-EF with SGD as the central server update rule. The proof follows the same routine as the one for Fed-EF-AMS, but is simpler since there are no moving average terms that need to be handled. Note that for this algorithm, we do not need Assumption 4 that the stochastic gradients are uniformly bounded.

For Fed-EF-SGD, consider the virtual sequence

$$x_{t+1} = \theta_{t+1} - \eta\bar{e}_{t+1}$$
$$= \theta_t - \eta\overline{\widetilde{\Delta}}_t - \eta\bar{e}_{t+1}$$
$$= \theta_t - \frac{\eta}{n}\sum_{i=1}^{n}(\tilde{\Delta}_{t,i} + e_{t+1,i})$$
$$= \theta_t - \eta\bar{\Delta}_t - \eta\bar{e}_t$$
$$= x_t - \eta\bar{\Delta}_t, \tag{13}$$

where the second last equality follows from the update rule that $\tilde{\Delta}_{t,i} + e_{t+1,i} = \Delta_{t,i} + e_{t,i}$ for all $i \in [n]$ and $t \in [T]$.

By the smoothness Assumption 1, we have

$$f(x_{t+1}) \leq f(x_t) + \langle\nabla f(x_t), x_{t+1} - x_t\rangle + \frac{L}{2}\|x_{t+1} - x_t\|^2.$$

Taking expectation w.r.t. the randomness at round $t$ gives

$$\mathbb{E}[f(x_{t+1})] - f(x_t)$$
$$\leq -\eta\mathbb{E}\big[\langle\nabla f(x_t), \bar{\Delta}_t\rangle\big] + \frac{\eta^2 L}{2}\mathbb{E}\big[\|\bar{\Delta}_t\|^2\big]$$
$$= -\eta\mathbb{E}\big[\langle\nabla f(\theta_t), \bar{\Delta}_t\rangle\big] + \frac{\eta^2 L}{2}\mathbb{E}\big[\|\bar{\Delta}_t\|^2\big] + \eta\mathbb{E}\big[\langle\nabla f(\theta_t) - \nabla f(x_t), \bar{\Delta}_t\rangle\big] \tag{14}$$

We can bound the first term in (14) using similar technique as bounding term **I** in the proof of Fed-EF-AMS. Specifically, we have

$$-\eta\mathbb{E}\big[\langle\nabla f(\theta_t), \bar{\Delta}_t\rangle\big] = -\eta\mathbb{E}\big[\langle\nabla f(\theta_t), \bar{\Delta}_t - \eta_l K\nabla f(\theta_t) + \eta_l K\nabla f(\theta_t)\rangle\big]$$
$$= -\eta\eta_l K\mathbb{E}\big[\|\nabla f(\theta_t)\|^2\big] + \eta\mathbb{E}\big[\langle\nabla f(\theta_t), -\bar{\Delta}_t + \eta_l K\nabla f(\theta_t)\rangle\big].$$

Note that the second term in the above can be bounded in the same way as bounding term **V** in Fed-EF-AMS (without the $\sqrt{\epsilon}$ factor). Thus, with $\eta_l \leq \frac{1}{8KL}$ we have

$$-\eta\mathbb{E}\big[\langle\nabla f(\theta_t), \bar{\Delta}_t\rangle\big] \leq -\eta\eta_l K\mathbb{E}\big[\|\nabla f(\theta_t)\|^2\big] + \frac{3\eta\eta_l K}{4}\mathbb{E}\big[\|\nabla f(\theta_t)\|^2\big] + \frac{5\eta\eta_l^3 K^2 L^2}{2}(\sigma^2 + 6K\sigma_g^2)$$
$$= -\frac{\eta\eta_l K}{4}\mathbb{E}\big[\|\nabla f(\theta_t)\|^2\big] + \frac{5\eta\eta_l^3 K^2 L^2}{2}(\sigma^2 + 6K\sigma_g^2).$$

The second term in (14) can be bounded using Lemma C.2 as

$$\frac{\eta^2 L}{2}\mathbb{E}\big[\|\bar{\Delta}_t\|^2\big] \leq \eta^2\eta_l^2 K^2 L(60\eta_l^2 K^2 L^2 + 1)\mathbb{E}\big[\|\nabla f(\theta_t)\|^2\big]$$
$$+ \frac{2\eta^2\eta_l^2 KL}{n}\sigma^2 + 10\eta^2\eta_l^4 K^3 L^3(\sigma^2 + 6K\sigma_g^2).$$

The last term in (14) can be bounded similarly as **VI** in Fed-EF-AMS by

$$
\begin{aligned}
&\eta \mathbb{E}\big[\langle \nabla f(\theta_t) - \nabla f(x_t), \bar{\Delta}_t\rangle\big]\\
&\leq \frac{\eta\rho}{2}\mathbb{E}\big[\|\bar{\Delta}_t\|^2\big] + \frac{\eta}{2\rho}\mathbb{E}\big[\|\nabla f(\theta_t) - \nabla f(x_t)\|^2\big]\\
&\overset{(a)}{\leq} \frac{\eta^2}{2}\mathbb{E}\big[\|\bar{\Delta}_t\|^2\big] + \frac{\eta^2 L^2}{2}\mathbb{E}\big[\|\bar{e}_t\|^2\big]\\
&\overset{(b)}{\leq} \frac{\eta^2 L}{2}\Big[2\eta_l^2 K^2(60\eta_l^2 K^2 L^2 + 1)\mathbb{E}\big[\|\nabla f(\theta_t)\|^2\big] + 4\frac{\eta_l^2 K}{n}\sigma^2 + 20\eta_l^4 K^3 L^2(\sigma^2 + 6K\sigma_g^2)\Big]\\
&\quad + \frac{\eta^2 L}{2}\Big[\frac{4q^2\eta_l^2 K^2(60\eta_l^2 K^2 L^2 + 1)}{1 - q^2}\sum_{\tau=1}^{t}(\frac{1 + q^2}{2})^{t-\tau}\mathbb{E}\big[\|\nabla f(\theta_\tau)\|^2\big]\\
&\qquad\qquad + \frac{16\eta_l^2 q^2 K}{(1 - q^2)^2 n}\sigma^2 + \frac{80\eta_l^4 q^2 K^3 L^2}{(1 - q^2)^2}(\sigma^2 + 6K\sigma_g^2)\Big],
\end{aligned}
\tag{15}
$$

where (a) uses Young's inequality and (b) uses Lemma C.2 and Lemma C.4. When taking telescoping sum of this term over $t = 1, ..., T$, again using the geometric summation trick, we further obtain

$$
\begin{aligned}
&\eta\sum_{t=1}^{T}\mathbb{E}\big[\langle\nabla f(\theta_t) - \nabla f(x_t), \bar{\Delta}_t\rangle\big]\\
&\leq \eta^2\eta_l^2 C_1 K^2 L(60\eta_l^2 K^2 L^2 + 1)\sum_{t=1}^{T}\mathbb{E}\big[\|\nabla f(\theta_t)\|^2\big]\\
&\qquad\qquad + \frac{2T\eta^2\eta_l^2 C_1 KL}{n}\sigma^2 + 10T\eta^2\eta_l^4 C_1 K^3 L^3(\sigma^2 + 6K\sigma_g^2),
\end{aligned}
$$

where $C_1 = 1 + \frac{4q^2}{(1-q^2)^2}$. Now, taking the summation over all terms in (14), we get

$$
\begin{aligned}
\mathbb{E}[f(x_{t+1})] - f(x_1) &\leq \Big(-\frac{\eta\eta_l K}{4} + \eta^2\eta_l^2(C_1 + 1)K^2 L(60\eta_l^2 K^2 L^2 + 1)\Big)\sum_{t=1}^{T}\mathbb{E}\big[\|\nabla f(\theta_t)\|^2\big]\\
&\quad + \frac{2T\eta^2\eta_l^2(C_1 + 1)KL}{n}\sigma^2 + 10T\eta^2\eta_l^4(C_1 + 1)K^3 L^3(\sigma^2 + 6K\sigma_g^2).
\end{aligned}
$$

Since $\eta_l \leq \frac{1}{8KL}$, we know that $60\eta_l^2 K^2 L^2 + 1 < 2$. Therefore, provided that the local learning rate is such that

$$
\eta_l \leq \frac{1}{2KL \cdot \max\{4, \eta(C_1 + 1)\}},
$$

we have

$$
\begin{aligned}
\frac{\eta\eta_l K}{8}\sum_{t=1}^{T}\mathbb{E}\big[\|\nabla f(\theta_t)\|^2\big] &\leq f(x_1) - \mathbb{E}[f(x_{t+1})] + \frac{2T\eta^2\eta_l^2(C_1 + 1)KL}{n}\sigma^2\\
&\quad + 10T\eta^2\eta_l^4(C_1 + 1)K^3 L^3(\sigma^2 + 6K\sigma_g^2),
\end{aligned}
$$

leading to

$$
\begin{aligned}
\frac{1}{T}\sum_{t=1}^{T}\mathbb{E}\big[\|\nabla f(\theta_t)\|^2\big] &\lesssim \frac{f(x_1) - \mathbb{E}[f(x_{t+1})]}{\eta\eta_l TK} + \frac{2\eta\eta_l(C_1 + 1)L}{n}\sigma^2\\
&\quad + 10\eta\eta_l^3(C_1 + 1)K^2 L^3(\sigma^2 + 6K\sigma_g^2)\\
&\leq \frac{f(\theta_1) - f(\theta^*)}{\eta\eta_l TK} + \frac{2\eta\eta_l(C_1 + 1)L}{n}\sigma^2\\
&\quad + 10\eta\eta_l^3(C_1 + 1)K^2 L^3(\sigma^2 + 6K\sigma_g^2),
\end{aligned}
$$

which concludes the proof. $\qquad\square$

## C.3 INTERMEDIATE LEMMAS

In our analysis, we will make use of the following lemma on the consensus error. Note that this is a general result holding for algorithms (both Fed-EF-SGD and Fed-EF-AMS) with local SGD steps.

**Lemma C.1** ([45]). *For $\eta_l \leq \frac{1}{8LK}$, for any round $t$, local step $k \in [K]$ and client $i \in [n]$, under Assumption 1 to Assumption 2, it holds that*

$$\mathbb{E}\big[\|\theta_{t,i}^{(k)} - \theta_t\|^2\big] \leq 5\eta_l^2 K(\sigma^2 + 6K\sigma_g^2) + 30\eta_l^2 K^2 \mathbb{E}\big[\|\nabla f(\theta_t)\|^2\big].$$

We then state some results that bound several key ingredients in our analysis.

**Lemma C.2.** *Recall $\bar{\Delta}_t = \frac{1}{n} \sum_{i=1}^n \Delta_{t,i}$. Under Assumption 1 to Assumption 2, for $\forall t$, the following bounds hold:*

*1. Bound by local gradients:*

$$\mathbb{E}\big[\|\bar{\Delta}_t\|^2\big] \leq \frac{\eta_l^2}{n^2} \mathbb{E}\big[\|\sum_{i=1}^n \sum_{k=1}^K \nabla f_i(\theta_{t,i}^{(k)})\|^2\big] + \frac{\eta_l^2 K}{n}\sigma^2.$$

*2. Bound by global gradient:*

$$\mathbb{E}\big[\|\bar{\Delta}_t\|^2\big] \leq (2\eta_l^2 K^2 + 120\eta_l^4 K^4 L^2)\mathbb{E}\big[\|\nabla f(\theta_t)\|^2\big] + 4\frac{\eta_l^2 K}{n}\sigma^2 + 20\eta_l^4 K^3 L^2(\sigma^2 + 6K\sigma_g^2).$$

*Proof.* By definition, we have

$$\mathbb{E}\big[\|\bar{\Delta}_t\|^2\big] = \mathbb{E}\big[\|\frac{1}{n} \sum_{i=1}^n \sum_{k=1}^K \eta_l g_{t,i}^{(k)}\|^2\big]$$

$$\leq \frac{\eta_l^2}{n^2} \mathbb{E}\big[\|\sum_{i=1}^n \sum_{k=1}^K (g_{t,i}^{(k)} - \nabla f_i(\theta_{t,i}^{(k)}))\|^2\big] + \frac{\eta_l^2}{n^2} \mathbb{E}\big[\|\sum_{i=1}^n \sum_{k=1}^K \nabla f_i(\theta_{t,i}^{(k)})\|^2\big]$$

$$\leq \frac{\eta_l^2 K}{n}\sigma^2 + \frac{\eta_l^2}{n^2} \mathbb{E}\big[\|\sum_{i=1}^n \sum_{k=1}^K \nabla f_i(\theta_{t,i}^{(k)})\|^2\big],$$

where the second line is due to the variance decomposition, and the last line is a result of Assumption 2 that the stochastic gradients are independent and unbiased. This proves the first part. For the second part, note that

$$\mathbb{E}\big[\|\bar{\Delta}_t\|^2\big] = \mathbb{E}\big[\|\frac{1}{n} \sum_{i=1}^n \sum_{k=1}^K \eta_l g_{t,i}^{(k)} - K\eta_l \nabla f(\theta_t) + K\eta_l \nabla f(\theta_t)\|^2\big]$$

$$\leq 2\eta_l^2 K^2 \mathbb{E}\big[\|\nabla f(\theta_t)\|^2\big] + 2\eta_l^2 \mathbb{E}\big[\|\frac{1}{n} \sum_{i=1}^n \sum_{k=1}^K g_{t,i}^{(k)} - \frac{K}{n} \sum_{i=1}^n \nabla f_i(\theta_t)\|^2\big]$$

$$= 2\eta_l^2 K^2 \mathbb{E}\big[\|\nabla f(\theta_t)\|^2\big] + \frac{2\eta_l^2}{n^2} \mathbb{E}\big[\|\sum_{i=1}^n \sum_{k=1}^K (g_{t,i}^{(k)} - \nabla f_i(\theta_t))\|^2\big]$$

$$\leq 2\eta_l^2 K^2 \mathbb{E}\big[\|\nabla f(\theta_t)\|^2\big] + \frac{2\eta_l^2}{n^2} \mathbb{E}\big[\|\sum_{i=1}^n \sum_{k=1}^K (g_{t,i}^{(k)} - \nabla f_i(\theta_{t,i}^{(k)}) + \nabla f_i(\theta_{t,i}^{(k)}) - \nabla f_i(\theta_t))\|^2\big]$$

$$\leq 2\eta_l^2 K^2 \mathbb{E}\big[\|\nabla f(\theta_t)\|^2\big] + \frac{2\eta_l^2}{n^2} \underbrace{\mathbb{E}\big[\|\sum_{i=1}^n \sum_{k=1}^K (g_{t,i}^{(k)} - \nabla f_i(\theta_{t,i}^{(k)}) + \nabla f_i(\theta_{t,i}^{(k)}) - \nabla f_i(\theta_t))\|^2\big]}_{A}.$$

The expectation $A$ can be further bounded as

$$A \leq 2\mathbb{E}\big[\|\sum_{i=1}^{n}\sum_{k=1}^{K}(g_{t,i}^{(k)} - \nabla f_i(\theta_{t,i}^{(k)}))\|^2\big] + 2\mathbb{E}\big[\|\sum_{i=1}^{n}\sum_{k=1}^{K}(\nabla f_i(\theta_{t,i}^{(k)}) - \nabla f_i(\theta_t))\|^2\big]$$

$$\overset{(a)}{\leq} 2nK\sigma^2 + 2nK\sum_{i=1}^{n}\sum_{k=1}^{K}\mathbb{E}\big[\|\nabla f_i(\theta_{t,i}^{(k)}) - \nabla f_i(\theta_t)\|^2\big]$$

$$\overset{(b)}{\leq} 2nK\sigma^2 + 2nKL^2\sum_{i=1}^{n}\sum_{k=1}^{K}\mathbb{E}\big[\|\theta_{t,i}^{(k)} - \theta_t\|^2\big]$$

$$\overset{(c)}{\leq} 60\eta_l^2 n^2 K^4 L^2 \mathbb{E}\big[\|\nabla f(\theta_t)\|^2\big] + 2nK\sigma^2 + 10\eta_l^2 n^2 K^3 L^2(\sigma^2 + 6K\sigma_g^2),$$

where (a) is implied by Assumption 2 that each local stochastic gradient $g_{t,i}^{(k)}$ can be written as $g_{t,i}^{(k)} = \nabla f_i(\theta_{t,i}^{(k)}) + \xi_{t,i}^{(k)}$, where $\xi_{t,i}^k$ is a zero-mean random noise with bounded variance $\sigma^2$, and all the noises for $t \in [T], i \in [n], k \in [K]$ are independent. The inequality (b) is due to the smoothness Assumption 1, and (c) follows from Lemma C.1. Therefore, we obtain

$$\mathbb{E}[\|\bar{\Delta}_t\|^2] \leq (2\eta_l^2 K^2 + 120\eta_l^4 K^4 L^2)\mathbb{E}[\|\nabla f(\theta_t)\|^2] + 4\frac{\eta_l^2 K}{n}\sigma^2 + 20\eta_l^4 K^3 L^2(\sigma^2 + 6K\sigma_g^2),$$

which completes the proof of the second claim. $\qquad\square$

**Lemma C.3.** *Under Assumption 1 to Assumption 2 we have:*

$$\|m_t'\| \leq \eta_l KG, \quad for \ \forall t,$$
$$\sum_{t=1}^{T}\mathbb{E}\big[\|m_t'\|^2\big] \leq (2\eta_l^2 K^2 + 120\eta_l^4 K^4 L^2)\sum_{t=1}^{T}\mathbb{E}\big[\|\nabla f(\theta_t)\|^2\big]+$$
$$+ 4\frac{T\eta_l^2 K}{n}\sigma^2 + 20T\eta_l^4 K^3 L^2(\sigma^2 + 6K\sigma_g^2).$$

*Proof.* For the first part, by Assumption 4 we know that

$$\|m_t'\| = (1 - \beta_1)\|\sum_{\tau=1}^{t}\beta_1^{t-\tau}\bar{\Delta}_t\|$$

$$= (1 - \beta_1)\sum_{\tau=1}^{t}\beta_1^{t-\tau}\frac{\eta_l}{n}\sum_{i=1}^{n}\sum_{k=1}^{K}\|g_{t,i}^{(k)}\|$$

$$\leq \eta_l KG.$$

For the second claim, by Lemma C.2 we know that

$$\mathbb{E}\big[\|\bar{\Delta}_t\|^2\big] \leq (2\eta_l^2 K^2 + 120\eta_l^4 K^4 L^2)\mathbb{E}\big[\|\nabla f(\theta_t)\|^2\big] + 4\frac{\eta_l^2 K}{n}\sigma^2 + 20\eta_l^4 K^3 L^2(\sigma^2 + 6K\sigma_g^2).$$

Let $\bar{\Delta}_{t,j}$ denote the $j$-th coordinate of $\bar{\Delta}_t$. By the updating rule of Fed-EF, we have

$$
\begin{aligned}
\mathbb{E}\big[\|m_t'\|^2\big] &= \mathbb{E}\big[\|(1-\beta_1)\sum_{\tau=1}^{t}\beta_1^{t-\tau}\bar{\Delta}_\tau\|^2\big] \\
&\le (1-\beta_1)^2\sum_{j=1}^{d}\mathbb{E}\big[(\sum_{\tau=1}^{t}\beta_1^{t-\tau}\bar{\Delta}_{\tau,j})^2\big] \\
&\overset{(a)}{\le} (1-\beta_1)^2\sum_{j=1}^{d}\mathbb{E}\big[(\sum_{\tau=1}^{t}\beta_1^{t-\tau})(\sum_{\tau=1}^{t}\beta_1^{t-\tau}\bar{\Delta}_{\tau,j}^2)\big] \\
&\le (1-\beta_1)\sum_{\tau=1}^{t}\beta_1^{t-\tau}\mathbb{E}\big[\|\bar{\Delta}_\tau\|^2\big] \\
&\le (2\eta_l^2 K^2 + 120\eta_l^4 K^4 L^2)(1-\beta_1)\sum_{\tau=1}^{t}\beta_1^{t-\tau}\mathbb{E}\big[\|\nabla f(\theta_t)\|^2\big] \\
&\qquad\qquad + 4\frac{\eta_l^2 K}{n}\sigma^2 + 20\eta_l^4 K^3 L^2(\sigma^2 + 6K\sigma_g^2),
\end{aligned}
$$

where (a) is due to Cauchy-Schwartz inequality. Summing over $t = 1, ..., T$, we obtain

$$
\begin{aligned}
\sum_{t=1}^{T}\mathbb{E}\big[\|m_t'\|^2\big] &\le (2\eta_l^2 K^2 + 120\eta_l^4 K^4 L^2)(1-\beta)\sum_{t=1}^{T}\sum_{\tau=1}^{t}\beta_1^{t-\tau}\mathbb{E}\big[\|\nabla f(\theta_t)\|^2\big] + \\
&\qquad\qquad + 4\frac{T\eta_l^2 K}{n}\sigma^2 + 20T\eta_l^4 K^3 L^2(\sigma^2 + 6K\sigma_g^2) \\
&\le (2\eta_l^2 K^2 + 120\eta_l^4 K^4 L^2)\sum_{t=1}^{T}\mathbb{E}\big[\|\nabla f(\theta_t)\|^2\big] + \\
&\qquad\qquad + 4\frac{T\eta_l^2 K}{n}\sigma^2 + 20T\eta_l^4 K^3 L^2(\sigma^2 + 6K\sigma_g^2),
\end{aligned}
$$

which concludes the proof. $\qquad\square$

**Lemma C.4.** *Under Assumption 2, we have for $\forall t$ and each local worker $\forall i \in [n]$,*

$$
\|e_{t,i}\|^2 \le \frac{4\eta_l^2 q^2 K^2 G^2}{(1-q^2)^2}, \ \forall t,
$$

$$
\begin{aligned}
\mathbb{E}[\|\bar{e}_{t+1}\|^2] &\le \frac{4q^2\eta_l^2 K^2(60\eta_l^2 K^2 L^2 + 1)}{1-q^2}\sum_{\tau=1}^{t}(\frac{1+q^2}{2})^{t-\tau}\mathbb{E}\big[\|\nabla f(\theta_\tau)\|^2\big] \\
&\qquad + \frac{16\eta_l^2 q^2 K}{(1-q^2)^2 n}\sigma^2 + \frac{80\eta_l^4 q^2 K^3 L^2}{(1-q^2)^2}(\sigma^2 + 6K\sigma_g^2).
\end{aligned}
$$

*Proof.* To prove the second claim, we start by using Assumption 3 and Young's inequality to get

$$
\begin{aligned}
\|\bar{e}_{t+1}\|^2 &= \|\bar{\Delta}_t + \bar{e}_t - \frac{1}{n}\sum_{i=1}^{n}\mathcal{C}(\Delta_{t,i} + e_{t,i})\|^2 \\
&\le q^2\|\bar{\Delta}_t + \bar{e}_t\|^2 \\
&\le q^2(1+\rho)\|\bar{e}_t\|^2 + q^2(1+\frac{1}{\rho})\|\bar{\Delta}_{t,i}\|^2 \\
&\le \frac{1+q^2}{2}\|\bar{e}_t\|^2 + \frac{2q^2}{1-q^2}\|\bar{\Delta}_t\|^2, \qquad\qquad (16)
\end{aligned}
$$

where (16) is derived by choosing $\rho = \frac{1-q^2}{2q^2}$ and the fact that $q < 1$. Now by recursion and the initialization $e_{1,i} = 0, \forall i$, we have

$$
\begin{aligned}
\mathbb{E}\big[\|\bar{e}_{t+1}\|^2\big] &\leq \frac{2q^2}{1-q^2}\sum_{\tau=1}^{t}\big(\frac{1+q^2}{2}\big)^{t-\tau}\mathbb{E}\big[\|\bar{\Delta}_\tau\|^2\big] \\
&\leq \frac{4q^2\eta_l^2 K^2(60\eta_l^2 K^2 L^2 + 1)}{1-q^2}\sum_{\tau=1}^{t}\big(\frac{1+q^2}{2}\big)^{t-\tau}\mathbb{E}\big[\|\nabla f(\theta_\tau)\|^2\big] \\
&\qquad + \frac{16\eta_l^2 q^2 K}{(1-q^2)^2 n}\sigma^2 + \frac{80\eta_l^4 q^2 K^3 L^2}{(1-q^2)^2}(\sigma^2 + 6K\sigma_g^2),
\end{aligned}
$$

which proves the second argument, where we use Lemma C.1 to bound the local local model update. In addition, we know that $\|\Delta_t\| \leq \eta_l KG$ by Assumption 4 for any $t$.

The absolute bound $\|e_{t,i}\|^2 \leq \frac{4\eta_l^2 q_C^2 K^2 G^2}{(1-q_C^2)^2}$ follows from (16) by a similar recursion argument used on local error $e_{t,i}$, and the fact that $q_C \leq \max\{q_C, q_A\} = q$.

$\square$

**Lemma C.5.** *For the moving average error sequence $\mathcal{E}_t$, it holds that*

$$
\begin{aligned}
\|\mathcal{E}_t\|^2 &\leq \frac{4\eta_l^2 q^2 K^2 G^2}{(1-q^2)^2}, \quad \text{for } \forall t, \\
\sum_{t=1}^{T}\mathbb{E}\big[\|\mathcal{E}_t\|^2\big] &\leq \frac{8q^2\eta_l^2 K^2(60\eta_l^2 K^2 L^2 + 1)}{(1-q^2)^2}\sum_{t=1}^{T}\mathbb{E}\big[\|\nabla f(\theta_\tau)\|^2\big] \\
&\qquad + \frac{16T\eta_l^2 q^2 K}{(1-q^2)^2 n}\sigma^2 + \frac{80T\eta_l^4 q^2 K^3 L^2}{(1-q^2)^2}(\sigma^2 + 6K\sigma_g^2).
\end{aligned}
$$

*Proof.* The first argument can be simply deduced by the definition of $\mathcal{E}_t$ that

$$
\begin{aligned}
\|\mathcal{E}_t\| &= (1-\beta_1)\|\sum_{\tau=1}^{t}\beta_1^{t-\tau}\bar{e}_t\| \\
&\leq \|e_{t,i}\| \leq \frac{2\eta_l qKG}{1-q^2}.
\end{aligned}
$$

Denote the quantity

$$
K_t := \sum_{\tau=1}^{t}\big(\frac{1+q^2}{2}\big)^{t-\tau}\mathbb{E}\big[\|\nabla f(\theta_\tau)\|^2\big].
$$

By the same technique as in the proof of Lemma C.3, denoting $\bar{e}_{t,j}$ as the $j$-th coordinate of $\bar{e}_t$, we can bound the accumulated error sequence by

$$
\begin{aligned}
\mathbb{E}\big[\|\mathcal{E}_t\|^2\big] &= \mathbb{E}\big[\|(1-\beta_1)\sum_{\tau=1}^{t}\beta_1^{t-\tau}\bar{e}_\tau\|^2\big] \\
&\leq (1-\beta_1)^2\sum_{j=1}^{d}\mathbb{E}\big[(\sum_{\tau=1}^{t}\beta_1^{t-\tau}\bar{e}_{\tau,j})^2\big] \\
&\overset{(a)}{\leq} (1-\beta_1)^2\sum_{j=1}^{d}\mathbb{E}\big[(\sum_{\tau=1}^{t}\beta_1^{t-\tau})(\sum_{\tau=1}^{t}\beta_1^{t-\tau}\bar{e}_{\tau,j}^2)\big] \\
&\leq (1-\beta_1)\sum_{\tau=1}^{t}\beta_1^{t-\tau}\mathbb{E}\big[\|\bar{e}_\tau\|^2\big] \\
&\overset{(b)}{\leq} \frac{16\eta_l^2 q^2 K}{(1-q^2)^2 n}\sigma^2 + \frac{80\eta_l^4 q^2 K^3 L^2}{(1-q^2)^2}(\sigma^2 + 6K\sigma_g^2) \\
&\qquad + \frac{4(1-\beta_1)q^2\eta_l^2 K^2(60\eta_l^2 K^2 L^2+1)}{1-q^2}\sum_{\tau=1}^{t}\beta_1^{t-\tau}K_\tau,
\end{aligned}
$$

where (a) is due to Cauchy-Schwartz inequality and (b) is a result of Lemma C.4. Summing over $t = 1, ..., T$ and using the technique of geometric series summation leads to

$$
\begin{aligned}
\sum_{t=1}^{T}\mathbb{E}\big[\|\mathcal{E}_t\|^2\big] &\leq \frac{16T\eta_l^2 q^2 K}{(1-q^2)^2 n}\sigma^2 + \frac{80T\eta_l^4 q^2 K^3 L^2}{(1-q^2)^2}(\sigma^2 + 6K\sigma_g^2) \\
&\qquad + \frac{4(1-\beta_1)q^2\eta_l^2 K^2(60\eta_l^2 K^2 L^2+1)}{1-q^2}\sum_{t=1}^{T}\sum_{\tau=1}^{t}\beta_1^{t-\tau}K_\tau \\
&\leq \frac{16T\eta_l^2 q^2 K}{(1-q^2)^2 n}\sigma^2 + \frac{80T\eta_l^4 q^2 K^3 L^2}{(1-q^2)^2}(\sigma^2 + 6K\sigma_g^2) \\
&\qquad + \frac{4q^2\eta_l^2 K^2(60\eta_l^2 K^2 L^2+1)}{1-q^2}\sum_{t=1}^{T}\sum_{\tau=1}^{t}(\frac{1+q^2}{2})^{t-\tau}\mathbb{E}\big[\|\nabla f(\theta_\tau)\|^2\big] \\
&\leq \frac{16T\eta_l^2 q^2 K}{(1-q^2)^2 n}\sigma^2 + \frac{80T\eta_l^4 q^2 K^3 L^2}{(1-q^2)^2}(\sigma^2 + 6K\sigma_g^2) \\
&\qquad + \frac{8q^2\eta_l^2 K^2(60\eta_l^2 K^2 L^2+1)}{(1-q^2)^2}\sum_{t=1}^{T}\mathbb{E}\big[\|\nabla f(\theta_\tau)\|^2\big].
\end{aligned}
$$

The desired result is obtained. $\qquad\square$

**Lemma C.6.** *It holds that $\forall t \in [T]$, $\forall i \in [d]$, $\hat{v}_{t,i} \leq \frac{4\eta_l^2(1+q^2)^3 K^2}{(1-q^2)^2}G^2$.*

*Proof.* For any $t$, by Lemma C.4 and Assumption 4 we have

$$
\begin{aligned}
\|\widetilde{\Delta}_t\|^2 &= \|\mathcal{C}(\Delta_t + e_t)\|^2 \\
&\leq \|\mathcal{C}(\Delta_t + e_t) - (\Delta_t + e_t) + (\Delta_t + e_t)\|^2 \\
&\leq 2(q^2+1)\|\Delta_t + e_t\|^2 \\
&\leq 4(q^2+1)(\eta_l^2 K^2 G^2 + \frac{4\eta_l^2 q^2 K^2 G^2}{(1-q^2)^2}) \\
&= \frac{4\eta_l^2(1+q^2)^3 K^2}{(1-q^2)^2}G^2.
\end{aligned}
$$

Consider the updating rule of $\hat{v}_t = \max\{v_t, \hat{v}_{t-1}\}$. We know that there exists a $j \in [t]$ such that $\hat{v}_t = v_j$. Thus, we have

$$\hat{v}_{t,i} = (1 - \beta_2) \sum_{\tau=1}^{j} \beta_2^{j-\tau} \tilde{g}_{t,i}^2 \leq \frac{4\eta_l^2 (1+q^2)^3 K^2}{(1-q^2)^2} G^2,$$

which proves the claim. $\qquad\square$

**Lemma C.7.** *Let* $D_t := \frac{1}{\sqrt{\hat{v}_{t-1}+\epsilon}} - \frac{1}{\sqrt{\hat{v}_t+\epsilon}}$ *be defined as above. Then,*

$$\sum_{t=1}^{T} \|D_t\|_1 \leq \frac{d}{\sqrt{\epsilon}}, \quad \sum_{t=1}^{T} \|D_t\|^2 \leq \frac{d}{\epsilon}.$$

*Proof.* By the updating rule of Fed-EF-AMS, $\hat{v}_{t-1} \leq \hat{v}_t$ for $\forall t$. Therefore, by the initialization $\hat{v}_0 = 0$, we have

$$\sum_{t=1}^{T} \|D_t\|_1 = \sum_{t=1}^{T} \sum_{i=1}^{d} \left( \frac{1}{\sqrt{\hat{v}_{t-1,i}+\epsilon}} - \frac{1}{\sqrt{\hat{v}_{t,i}+\epsilon}} \right)$$

$$= \sum_{i=1}^{d} \left( \frac{1}{\sqrt{\hat{v}_{0,i}+\epsilon}} - \frac{1}{\sqrt{\hat{v}_{T,i}+\epsilon}} \right)$$

$$\leq \frac{d}{\sqrt{\epsilon}}.$$

For the sum of squared $l_2$ norm, note the fact that for $a \geq b > 0$, it holds that

$$(a-b)^2 \leq (a-b)(a+b) = a^2 - b^2.$$

Thus,

$$\sum_{t=1}^{T} \|D_t\|^2 = \sum_{t=1}^{T} \sum_{i=1}^{d} \left( \frac{1}{\sqrt{\hat{v}_{t-1,i}+\epsilon}} - \frac{1}{\sqrt{\hat{v}_{t,i}+\epsilon}} \right)^2$$

$$\leq \sum_{t=1}^{T} \sum_{i=1}^{d} \left( \frac{1}{\hat{v}_{t-1,i}+\epsilon} - \frac{1}{\hat{v}_{t,i}+\epsilon} \right)$$

$$\leq \frac{d}{\epsilon},$$

which gives the desired result. $\qquad\square$

## C.4 PROOF OF THEOREM 3: PARTIAL PARTICIPATION

*Proof.* We can use a similar proof structure as previous analysis for full participation Fed-EF-SGD as in Section C.2. Like before, we first define the following virtual iterates:

$$x_{t+1} = \theta_{t+1} - \eta \frac{1}{m} \sum_{i=1}^{n} e_{t+1,i}$$

$$= \theta_t - \eta \overline{\tilde{\Delta}}_{t,\mathcal{M}_t} - \eta \frac{1}{m} \sum_{i \in \mathcal{M}_t} e_{t+1,i} - \eta \frac{1}{m} \sum_{i \notin \mathcal{M}_t} e_{t+1,i}$$

$$= \theta_t - \eta \bar{\Delta}_{t,\mathcal{M}_t} - \eta \frac{1}{m} \sum_{i \in \mathcal{M}_t} e_{t,i} - \eta \frac{1}{m} \sum_{i \notin \mathcal{M}_t} e_{t,i} \qquad (17)$$

$$= \theta_t - \eta \bar{\Delta}_{t,\mathcal{M}_t} - \eta \frac{1}{m} \sum_{i=1}^{n} e_{t,i}$$

$$= x_t - \eta \bar{\Delta}_{t,\mathcal{M}_t}.$$

Here, (17) follows from the partial participation setup where there is no error accumulation for inactive clients. The smoothness of loss functions implies

$$f(x_{t+1}) \leq f(x_t) + \langle \nabla f(x_t), x_{t+1} - x_t \rangle + \frac{L}{2}\|x_{t+1} - x_t\|^2.$$

Taking expectation w.r.t. the randomness at round $t$, we have

$$\mathbb{E}[f(x_{t+1})] - f(x_t)$$

$$\leq -\eta\mathbb{E}\big[\langle \nabla f(x_t), \bar{\Delta}_{t,\mathcal{M}_t}\rangle\big] + \frac{\eta^2 L}{2}\mathbb{E}\big[\|\bar{\Delta}_{t,\mathcal{M}_t}\|^2\big]$$

$$= \underbrace{-\eta\mathbb{E}\big[\langle \nabla f(\theta_t), \bar{\Delta}_{t,\mathcal{M}_t}\rangle\big]}_{I} + + \underbrace{\frac{\eta^2 L}{2}\mathbb{E}\big[\|\bar{\Delta}_{t,\mathcal{M}_t}\|^2\big]}_{II} + \underbrace{\eta\mathbb{E}\big[\langle \nabla f(x_t) - \nabla f(\theta_t), \bar{\Delta}_{t,\mathcal{M}_t}\rangle\big]}_{III}. \quad (18)$$

Note that the expectation is also with respect to the randomness in the client sampling procedure. For the first term, since the client sampling is random, we have $\mathbb{E}[\bar{\Delta}_{t,\mathcal{M}_t}] = \mathbb{E}[\bar{\Delta}_t]$. Thus,

$$I = -\eta\mathbb{E}\big[\langle \nabla f(\theta_t), \bar{\Delta}_{t,\mathcal{M}_t}\rangle\big]$$

$$= -\eta\mathbb{E}\big[\langle \nabla f(\theta_t), \bar{\Delta}_t - \eta_l K\nabla f(\theta_t) + \eta_l K\nabla f(\theta_t)\rangle\big]$$

$$= -\eta\eta_l K\mathbb{E}\big[\|\nabla f(\theta_t)\|^2\big] + \eta\eta_l\big\langle \sqrt{K}\nabla f(\theta_t), -\frac{1}{n\sqrt{K}}\sum_{i=1}^{n}\sum_{k=1}^{K}\big(\nabla f_i(\theta_{t,i}^{(k)}) + \nabla f(\theta_t)\big)\big\rangle$$

$$\overset{(a)}{=} -\eta\eta_l K\mathbb{E}\big[\|\nabla f(\theta_t)\|^2\big] + \eta\eta_l\mathbb{E}\Big[\frac{K}{2}\|\nabla f(\theta_t)\|^2$$

$$+ \frac{1}{2Kn^2}\|\sum_{i=1}^{n}\sum_{k=1}^{K}(\nabla f_i(\theta_{t,i}^{(k)}) + \nabla f(\theta_t))\|^2 - \frac{1}{2Kn^2}\|\sum_{i=1}^{n}\sum_{k=1}^{K}\nabla f_i(\theta_{t,i}^{(k)})\|^2\Big]$$

$$\overset{(b)}{\leq} -\eta\eta_l K\mathbb{E}\big[\|\nabla f(\theta_t)\|^2\big] + \frac{\eta\eta_l K}{2}\mathbb{E}\big[\|\nabla f(\theta_t)\|^2\big]$$

$$+ \frac{\eta\eta_l K L^2}{2}\Big[5\eta_l^2 K(\sigma^2 + 6K\sigma_g^2) + 30\eta_l^2 K^2\mathbb{E}\big[\|\nabla f(\theta_t)\|^2\big]\Big] - \frac{\eta\eta_l}{2Kn^2}\mathbb{E}\big[\|\sum_{i=1}^{n}\sum_{k=1}^{K}\nabla f_i(\theta_{t,i}^{(k)})\|^2\big]$$

$$\leq -\frac{\eta\eta_l K}{4}\mathbb{E}\big[\|\nabla f(\theta_t)\|^2\big] + \frac{5\eta\eta_l^3 K^2 L^2}{2}(\sigma^2 + 6K\sigma_g^2) - \frac{\eta\eta_l}{2Kn^2}\mathbb{E}\big[\|\sum_{i=1}^{n}\sum_{k=1}^{K}\nabla f_i(\theta_{t,i}^{(k)})\|^2\big],$$

when $\eta_l \leq \frac{1}{8KL}$, where (a) is a result of the fact that $\langle z_1, z_2 \rangle = \|z_1\|^2 + \|z_2\|^2 - \|z_1 - z_2\|^2$, (b) is because of Lemma C.1.

For term II, we have

$$II \leq \frac{\eta^2\eta_l^2 KL}{2m}\sigma^2 + \frac{\eta^2\eta_l^2 L}{2n(n-1)}\mathbb{E}\big[\|\sum_{i=1}^{n}\sum_{k=1}^{K}\nabla f_i(\theta_{t,i}^{(k)})\|\big]^2$$

$$+ C'\Big[\frac{3\eta^2\eta_l^2 K^2 L(30\eta_l^2 K^2 L^2 + 1)}{2m}\mathbb{E}\big[\|\nabla f(\theta_t)\|^2\big] + \frac{15\eta^2\eta_l^4 K^3 L^3}{2m}(\sigma^2 + 6K\sigma_g^2) + \frac{3\eta^2\eta_l^2 K^2 L}{2m}\sigma_g^2\Big],$$

with $C' = \frac{n-m}{n-1}$. Furthermore, we have that

$$III \leq 2\eta^2 L\mathbb{E}\big[\|\frac{1}{m}\sum_{i=1}^{n}e_{t,i}\|^2\big] + \frac{\eta^2 L}{2}\mathbb{E}\big[\|\bar{\Delta}_{t,\mathcal{M}_t}\|^2\big].$$

The second term is the same as term II. We now bound the first term. Denote $\tilde{e}_{t,i} = e_{t,i} + \Delta_{t,i} - \widetilde{\Delta}_{t,i}$, we have

$$\mathbb{E}\big[\|\frac{1}{m}\sum_{i=1}^{n}e_{t+1,i}\|^2\big] = \frac{1}{m^2}\mathbb{E}_{e_t}\Big[\underbrace{\mathbb{E}_{\mathcal{M}_t}\big[\|\sum_{i=1}^{n}\mathbb{1}\{i \in \mathcal{M}_t\}\tilde{e}_{t,i} + \sum_{i=1}^{n}\mathbb{1}\{i \notin \mathcal{M}_t\}e_{t,i}\|^2\big|e_t\big]}_{A}\Big].$$

By the updating rule of $e_{t,i}$, the inner expectation, conditional on $\boldsymbol{e}_t = (e_{t,1}, ..., e_{t,n})^T$, can be computed as

$$A = \frac{m}{n} \sum_{i=1}^{n} \|\tilde{e}_{i,t}\|^2 + \frac{m(m-1)}{n(n-1)} \sum_{i \neq j}^{n} \tilde{e}_{t,i} \tilde{e}_{t,j} + \frac{n-m}{n} \sum_{i=1}^{n} \|e_{i,t}\|^2$$

$$+ \frac{(n-m)(n-m-1)}{n(n-1)} \sum_{i \neq j}^{n} e_{t,i} e_{t,j} + \frac{m(n-m)}{n(n-1)} \sum_{i \neq j}^{n} \tilde{e}_{t,i} e_{t,j}$$

$$= \frac{m}{n} \|\sum_{i=1}^{n} \tilde{e}_{t,i}\|^2 - \frac{m(n-m)}{n(n-1)} \sum_{i \neq j}^{n} \tilde{e}_{t,i} \tilde{e}_{t,j} + \frac{n-m}{n} \|\sum_{i=1}^{n} e_{i,t}\|^2$$

$$- \frac{m(n-m)}{n(n-1)} \sum_{i \neq j}^{n} e_{t,i} e_{t,j} + \frac{m(n-m)}{n(n-1)} \sum_{i \neq j}^{n} \tilde{e}_{t,i} e_{t,j}$$

$$= \frac{m}{n} \|\sum_{i=1}^{n} \tilde{e}_{t,i}\|^2 + \frac{n-m}{n} \|\sum_{i=1}^{n} e_{i,t}\|^2 - \frac{m(n-m)}{n(n-1)} \sum_{i \neq j}^{n} (\tilde{e}_{t,i} \tilde{e}_{t,j} + e_{t,i} e_{t,j} - \tilde{e}_{t,i} e_{t,j})$$

$$= \frac{m}{n} \|\sum_{i=1}^{n} \tilde{e}_{t,i}\|^2 + \frac{n-m}{n} \|\sum_{i=1}^{n} e_{i,t}\|^2$$

$$- \frac{m(n-m)}{n(n-1)} \|\sum_{i=1}^{n} (\tilde{e}_{t,i} - e_{t,i})\|^2 + \frac{m(n-m)}{n(n-1)} \sum_{i=1}^{n} (\|\tilde{e}_{t,i}\|^2 + \|e_{t,i}\|^2)$$

$$\leq \frac{m}{n} \|\sum_{i=1}^{n} \tilde{e}_{t,i}\|^2 + \frac{n-m}{n} \|\sum_{i=1}^{n} e_{t,i}\|^2 + \frac{m(n-m)}{n(n-1)} \sum_{i=1}^{n} (\|\tilde{e}_{t,i}\|^2 + \|e_{t,i}\|^2).$$

Therefore, by Definition 1 and Assumption 3 we obtain

$$\mathbb{E}\Big[\|\frac{1}{m} \sum_{i=1}^{n} e_{t+1,i}\|^2\Big]$$

$$\leq \frac{mq^2}{n} \mathbb{E}[\|\frac{1}{m} \sum_{i \in \mathcal{G}} (e_{t,i} + \Delta_{t,i})\|^2] + \frac{n-m}{n} \mathbb{E}[\|\frac{1}{m} \sum_{i=1}^{n} e_{t,i}\|^2]$$

$$+ \frac{(n-m)}{mn(n-1)} \sum_{i=1}^{n} ((2q^2+1)\|e_{t,i}\|^2 + 2q^2\|\Delta_{t,i}\|^2)$$

$$\leq \frac{m(1+\gamma)q^2 + (n-m)}{n} \mathbb{E}[\|\frac{1}{m} \sum_{i=1}^{n} e_{t,i}\|^2] + \frac{m(1+1/\gamma)q^2}{n} \mathbb{E}[\|\frac{1}{m} \sum_{i=1}^{n} \Delta_{t,i}\|^2]$$

$$+ \frac{(n-m)}{mn(n-1)} \sum_{i=1}^{n} ((2q^2+1)\|e_{t,i}\|^2 + 2q^2\|\Delta_{t,i}\|^2).$$

We have by Lemma C.1 that

$$\mathbb{E}[\|e_{t,i}\|^2] \leq \frac{20q^2\eta_l^2 K}{(1-q^2)^2}(\sigma^2 + 6K\sigma_g^2) + \frac{60\eta_l^2 q^2 K^2}{1-q^2} \sum_{\tau=1}^{t} (\frac{1+q^2}{2})^{t-\tau} \mathbb{E}[\|\nabla f(\theta_\tau)\|^2],$$

$$\mathbb{E}[\|\Delta_{t,i}\|^2] \leq 5\eta_l^2 K(\sigma^2 + 6K\sigma_g^2) + 30\eta_l^2 K^2 \mathbb{E}[\|\nabla f(\theta_t)\|^2],$$

which implies

$$\frac{(n-m)}{mn(n-1)} \sum_{i=1}^{n} ((2q^2+1)\|e_{t,i}\|^2 + 2q^2\|\Delta_{t,i}\|^2)$$

$$\leq \frac{(n-m)}{m(n-1)} \Big[ \frac{70q^2\eta_l^2 K}{(1-q^2)^2}(\sigma^2 + 6K\sigma_g^2) + \frac{180\eta_l^2 q^2 K^2}{1-q^2} \sum_{\tau=1}^{t} (\frac{1+q^2}{2})^{t-\tau} \mathbb{E}[\|\nabla f(\theta_\tau)\|^2]$$

$$+ \frac{60\eta_l^2 q^2 K^2}{(1-q^2)^2} \mathbb{E}[\|\nabla f(\theta_t)\|^2] \Big].$$

Recall $q = \max\{q_\mathcal{A}, q_\mathcal{C}\}$. Let $\gamma = (1 - q^2)/2q^2$. We have

$$\frac{m(1 + \gamma)q^2 + (n - m)}{n} = 1 - \frac{(1 - q^2)m}{2n} < 1,$$

$$\frac{m(1 + 1/\gamma)q^2}{n} = \frac{m(1 + q^2)q^2}{n(1 - q^2)} \le \frac{2mq^2}{n(1 - q^2)}.$$

By the recursion argument used before, applying Lemma C.2 (adjusted by a $n^2/m^2$ factor) we obtain

$$\mathbb{E}\big[\|\frac{1}{m}\sum_{i=1}^n e_{t+1,i}\|^2\big]$$

$$\le \frac{2mq^2}{n(1 - q^2)}\sum_{\tau=1}^t \big(1 - \frac{(1 - q^2)m}{2n}\big)^{t-\tau}\frac{n^2}{m^2}\big[\frac{\eta_l^2}{n^2}\mathbb{E}\big[\|\sum_{i=1}^n\sum_{k=1}^K \nabla f_i(\theta_{\tau,i}^{(k)})\|^2\big] + \frac{\eta_l^2 K}{n}\sigma^2\big]$$

$$+ \frac{2n(n - m)}{(1 - q^2)m^2(n - 1)}\Big[\frac{70q^2\eta_l^2 K}{(1 - q^2)^2}(\sigma^2 + 6K\sigma_g^2) + \frac{180\eta_l^2 q^2 K^2}{1 - q^2}\sum_{\tau=1}^t(\frac{1 + q^2}{2})^{t-\tau}\mathbb{E}\big[\|\nabla f(\theta_\tau)\|^2\big]$$

$$+ \frac{60\eta_l^2 q^2 K^2}{(1 - q^2)^2}\mathbb{E}\big[\|\nabla f(\theta_t)\|^2\big]\Big]$$

$$\le \frac{2\eta_l^2 q^2}{(1 - q^2)mn}\sum_{\tau=1}^t \big(1 - \frac{(1 - q^2)m}{2n}\big)^{t-\tau}\mathbb{E}\big[\|\sum_{i=1}^n\sum_{k=1}^K \nabla f_i(\theta_{\tau,i}^{(k)})\|^2\big] + \frac{4\eta_l^2 q^2 Kn}{(1 - q^2)^2 m^2}\sigma^2$$

$$+ \frac{280\eta_l^2 q^2(n - m)K}{(1 - q^2)^3 m^2}(\sigma^2 + 6K\sigma_g^2) + \frac{720\eta_l^2 q^2(n - m)K^2}{(1 - q^2)^2 m^2}\sum_{\tau=1}^t(\frac{1 + q^2}{2})^{t-\tau}\mathbb{E}\big[\|\nabla f(\theta_\tau)\|^2\big]$$

$$+ \frac{240\eta_l^2 q^2(n - m)K^2}{(1 - q^2)^3 m^2}\mathbb{E}\big[\|\nabla f(\theta_t)\|^2\big].$$

Summing over $t = 1, ..., T$ gives

$$\sum_{t=1}^T \mathbb{E}\big[\|\frac{1}{m}\sum_{i=1}^n e_{t+1,i}\|^2\big]$$

$$\le \frac{4\eta_l^2 q^2}{(1 - q^2)^2 m^2}\sum_{t=1}^T \mathbb{E}\big[\|\sum_{i=1}^n\sum_{k=1}^K \nabla f_i(\theta_{t,i}^{(k)})\|^2\big] + \frac{4T\eta_l^2 q^2 Kn}{(1 - q^2)^2 m^2}\sigma^2$$

$$+ \frac{280T\eta_l^2 q^2(n - m)K}{(1 - q^2)^3 m^2}(\sigma^2 + 6K\sigma_g^2) + \frac{1680\eta_l^2 q^2(n - m)K^2}{(1 - q^2)^3 m^2}\sum_{t=1}^T \mathbb{E}\big[\|\nabla f(\theta_t)\|^2\big].$$

Now we turn back to (18). By taking the telescoping sum, we have

$$\mathbb{E}[f(x_{t+1})] - f(x_1)$$

$$\le -\frac{\eta\eta_l K}{4}\sum_{t=1}^T \mathbb{E}\big[\|\nabla f(\theta_t)\|^2\big] + \frac{5T\eta\eta_l^3 K^2 L^2}{2}(\sigma^2 + 6K\sigma_g^2) - \frac{\eta\eta_l}{2Kn^2}\sum_{t=1}^T \mathbb{E}\big[\|\sum_{i=1}^n\sum_{k=1}^K \nabla f_i(\theta_{t,i}^{(k)})\|^2\big]$$

$$+ \frac{T\eta^2\eta_l^2 KL}{m}\sigma^2 + \frac{\eta^2\eta_l^2 L}{n(n - 1)}\sum_{t=1}^T \mathbb{E}\big[\|\sum_{i=1}^n\sum_{k=1}^K \nabla f_i(\theta_{t,i}^{(k)})\|\big]^2$$

$$+ \frac{3\eta^2\eta_l^2 C' K^2 L(30\eta_l^2 K^2 L^2 + 1)}{m}\sum_{t=1}^T \mathbb{E}\big[\|\nabla f(\theta_t)\|^2\big] + \frac{15T\eta^2\eta_l^4 C' K^3 L^3}{m}(\sigma^2 + 6K\sigma_g^2) + \frac{3T\eta^2\eta_l^2 C' K^2 L}{m}\sigma_g^2$$

$$+ \frac{8\eta^2\eta_l^2 q^2 L}{(1 - q^2)^2 m^2}\sum_{t=1}^T \mathbb{E}\big[\|\sum_{i=1}^n\sum_{k=1}^K \nabla f_i(\theta_{t,i}^{(k)})\|^2\big] + \frac{8T\eta^2\eta_l^2 q^2 KLn}{(1 - q^2)^3 m^2}\sigma^2$$

$$+ \frac{560T\eta^2\eta_l^2 q^2(n - m)KL}{(1 - q^2)^3 m^2}(\sigma^2 + 6K\sigma_g^2) + \frac{3360\eta^2\eta_l^2 q^2(n - m)K^2 L}{(1 - q^2)^3 m^2}\sum_{t=1}^T \mathbb{E}\big[\|\nabla f(\theta_t)\|^2\big].$$

When the learning rate is chosen such that

$$\eta_l \le \min\left\{\frac{1}{6}, \frac{m}{96C'\eta}, \frac{m^2}{53760(n-m)C_1\eta}, \frac{1}{4\eta}, \frac{1}{32C_1\eta}\right\}\frac{1}{KL},$$

we can get

$$\frac{1}{T}\sum_{t=1}^{T}\mathbb{E}\big[\|\nabla f(\theta_t)\|^2\big] \lesssim \frac{f(\theta_1)-f(\theta^*)}{\eta\eta_l TK} + \left[\frac{\eta\eta_l L}{m} + \frac{8\eta\eta_l C_1 Ln}{m^2}\right]\sigma^2 + \frac{3\eta\eta_l C'KL}{m}\sigma_g^2$$

$$+ \left[\frac{5\eta_l^2 KL^2}{2} + \frac{15\eta\eta_l^3 C'K^2L^3}{m} + \frac{560\eta\eta_l C_1(n-m)L}{m^2}\right](\sigma^2 + 6K\sigma_g^2),$$

where $C_1 = q^2/(1-q^2)^3$. Denote $B = n/m$. When choosing $\eta = \Theta(\sqrt{Km})$, $\eta_l = \Theta(\frac{1}{K\sqrt{TB}})$, the rate can be further bounded by

$$\frac{1}{T}\sum_{t=1}^{T}\mathbb{E}\big[\|\nabla f(\theta_t)\|^2\big] = \mathcal{O}\Big(\frac{\sqrt{B}(f(\theta_1)-f(\theta^*))}{\sqrt{TKm}} + \Big(\frac{1}{\sqrt{TKmB}} + \frac{\sqrt{B}}{\sqrt{TKm}}\Big)\sigma^2 + \frac{\sqrt{K}}{\sqrt{TmB}}\sigma_g^2$$

$$+ \Big(\frac{1}{TKB} + \frac{1}{T^{3/2}B^{3/2}\sqrt{Km}} + \frac{\sqrt{B}}{\sqrt{TKm}}\Big)(\sigma^2 + 6K\sigma_g^2)\Big),$$

which can be further simplified by ignoring smaller terms as

$$\frac{1}{T}\sum_{t=1}^{T}\mathbb{E}\big[\|\nabla f(\theta_t)\|^2\big] = \mathcal{O}\Big(\frac{\sqrt{n}}{\sqrt{m}}\Big(\frac{f(\theta_1)-f(\theta^*)}{\sqrt{TKm}} + \frac{1}{\sqrt{TKm}}\sigma^2 + \frac{\sqrt{K}}{\sqrt{Tm}}\sigma_g^2\Big)\Big).$$

This completes the proof. $\qquad\square$

---

**Algorithm 3** Fed-EF Scheme with Two-Way Compression

---

1: **Input**: learning rates $\eta$, $\eta_l$, hyper-parameters $\beta_1$, $\beta_2$, $\epsilon$

2: **Initialize**: central server parameter $\theta_1 \in \mathbb{R}^d \subseteq \mathbb{R}^d$; $e_{1,i} = \mathbf{0}$ the accumulator for each worker;
$m_0 = \mathbf{0}$, $v_0 = \mathbf{0}$, $\hat{v}_0 = \mathbf{0}$ ; $\phi_1 = \tilde{H}_t = \mathbf{0}$; $\theta_{0,i}^{(1)} = \theta_1$ for all $i \in [n]$

3: **for** $t = 1, \ldots, T$ **do**

4:     **parallel for** worker $i \in [n]$ **do**:

5:         Receive $\tilde{H}_t$ from the server and set $\theta_{t,i}^{(1)} = \theta_{t-1,i}^{(1)} + \tilde{H}_t$        ▷ Download compression

6:         **for** $k = 1, \ldots, K$ **do**

7:             Compute stochastic gradient $g_{t,i}^{(k)}$ at $\theta_{t,i}^{(k)}$

8:             Local update $\theta_{t,i}^{(k+1)} = \theta_{t,i}^{(k)} - \eta_l g_{t,i}^{(k)}$

9:         **end for**

10:         Compute the local model update $\Delta_{t,i} = \theta_{t,i}^{(K+1)} - \theta_t$

11:         Send compressed adjusted local model update $\widetilde{\Delta}_{t,i} = \mathcal{C}(\Delta_{t,i} + e_{t,i})$ to central server

12:         Update the error $e_{t+1,i} = e_{t,i} + \Delta_{t,i} - \widetilde{\Delta}_{t,i}$

13:     **end parallel**

14:     **Central server do:**

15:     Global aggregation $\overline{\widetilde{\Delta}}_t = \frac{1}{n} \sum_{i=1}^{n} \widetilde{\Delta}_{t,i}$

16:     Compress $\tilde{H}_t = \mathcal{C}(\overline{\widetilde{\Delta}}_t + \phi_t)$                        ▷ Fed-EF-SGD

17:     $m_t = \beta_1 m_{t-1} + (1 - \beta_1)\overline{\widetilde{\Delta}}_t$                 ▷ Fed-EF-AMS

18:     $v_t = \beta_2 v_{t-1} + (1 - \beta_2)\overline{\widetilde{\Delta}}_t^2$,   $\hat{v}_t = \max(v_t, \hat{v}_{t-1})$

19:     Compress $\tilde{H}_t = \mathcal{C}(\frac{m_t}{\sqrt{\hat{v}_t + \epsilon}} + \phi_t)$

20:     Update server error accumulator $\phi_{t+1} = \phi_t + (\theta_{t+1} - \theta_t) - \tilde{H}_t$

21:     Update global model $\theta_{t+1} = \theta_t - \eta\tilde{H}_t$ and broadcast $\tilde{H}_t$ to clients

22: **end for**

---

## D TWO-WAY COMPRESSION IN FED-EF

As discussed in the paper, our Fed-EF scheme can also be combined with two-way compression, for both uploading (clients-to-server) and downloading (server-to-clients) channel. This can lead to even more communication reduction in practice. The steps can be found in Algorithm 3. Next, we briefly discuss its impact on the convergence analysis. For simplicity, we will focus on two-way compressed Fed-EF-SGD here, while same arguments hold for Fed-EF-AMS. The general approach is: 1) the clients transmit $\tilde{\Delta}_{t,i}$ to the server which are compressed; 2) the server again compresses the aggregated update $\overline{\widetilde{\Delta}}_t$ and broadcast the compressed $\tilde{H}_t$ to the clients, also using an error feedback at the central node. Note that this approach requires the clients to additionally store the model at the beginning of each round.

To study the convergence of Algorithm 3, we consider a series of virtual iterates as

$$
\begin{aligned}
\tilde{\theta}_{t+1} &= \theta_{t+1} - \eta\phi_{t+1} \\
&= \theta_t - \eta(\tilde{H}_t + \phi_{t+1}) \\
&= \theta_t - \frac{1}{n}\sum_{i=1}^{n}\widetilde{\Delta}_{t,i} - \phi_t \\
&= \tilde{\theta}_t - \overline{\widetilde{\Delta}}_t,
\end{aligned}
$$

where we again use the fact of EF that $\phi_{t+1} + \tilde{H}_t = \phi_t + \overline{\tilde{\Delta}}_t$. Then we can construct a similar sequence $x_t$ as in (13) associated with $\tilde{\theta}_t$ by

$$x_{t+1} = \tilde{\theta}_{t+1} - \eta \bar{e}_{t+1} = x_t - \eta \bar{\Delta}_t.$$

Then we can apply same analysis to derive the convergence bound as in Section C.2. The only difference is in (15), where the second term becomes

$$\frac{\eta^2 L^2}{2} \mathbb{E}\big[\|\bar{e}_t + \phi_t\|^2\big] \le \eta^2 L^2 \mathbb{E}\big[\|\bar{e}_t\|^2\big] + \eta^2 L^2 \mathbb{E}\big[\|\phi_t\|^2\big]. \tag{19}$$

The first term can be bounded in the same way as in (15). Regarding the second term, we can use a similar trick as Lemma C.4 that

$$\begin{aligned}
\|\phi_{t+1}\|^2 &= \|\phi_t + \overline{\tilde{\Delta}}_t - \mathcal{C}(\phi_t + \overline{\tilde{\Delta}}_t)\|^2 \\
&\le q_{\mathcal{C}}^2 \|\phi_t + \overline{\tilde{\Delta}}_t\|^2 \\
&\le \frac{1 + q_{\mathcal{C}}^2}{2} \|\phi_t\|^2 + \frac{2q_{\mathcal{C}}^2}{1 - q_{\mathcal{C}}^2} \|\overline{\tilde{\Delta}}_t\|^2.
\end{aligned}$$

Then, by recursion and the geometric sum, $\|\phi_{t+1}\|^2$ can be bounded by the second term in above up to a constant. We can write

$$\begin{aligned}
\mathbb{E}\big[\|\overline{\tilde{\Delta}}_t\|^2\big] &\le \mathbb{E}\big[\|\bar{\Delta}_t + \bar{e}_t - \bar{e}_{t+1}\|^2\big] \\
&\le 3(\mathbb{E}\big[\|\bar{\Delta}_t\|^2\big] + \mathbb{E}\big[\|\bar{e}_t\|^2\big] + \mathbb{E}\big[\|\bar{e}_{t+1}\|^2\big]).
\end{aligned}$$

As a result, it holds that $\mathbb{E}\big[\|\phi_t\|^2\big] \le \mathcal{O}(\mathbb{E}\big[\|\bar{e}_t\|\big]^2)$ since $\mathbb{E}\big[\|\bar{e}_t\|^2\big] = \mathcal{O}(\mathbb{E}\big[\|\bar{\Delta}_t\|\big]^2)$ by Lemma C.2 and Lemma C.4 under our assumptions. Therefore, (19) has same order as (15). Since other parts of the proof are the same, we know that two-way compression does not change the convergence rate asymptotically of Fed-EF.

