# OpenReview forum: "Analysis of Error Feedback in Compressed Federated Non-Convex Optimization"
_ICLR.cc/2023/Conference — Submitted to ICLR 2023_

### Official Review · Reviewer_GNNs · 2022-10-21

**Confidence:** 4
**Correctness:** 3
**Technical Novelty And Significance:** 2
**Empirical Novelty And Significance:** 2
**Recommendation:** 5

**Clarity, Quality, Novelty And Reproducibility:**

The paper is well written. For novelty and quality, please see the weakness section.

**Strength And Weaknesses:**

Strength: The paper shows linear speedup (a $n$ dependent term in the statistical rate) with the number of participating clients.

Weaknesses:

**Missing Comparison and Prior work:** The paper does not compare with several existing related works. The problem of Error Feedback in FL is studied extensively contrary to the claim of the authors. For example, the following papers

 (a)  `EF21: A New, Simpler, Theoretically Better, and Practically Faster Error Feedback' by Richtarik et. al;
 (b) `EF21 with Bells & Whistles: Practical Algorithmic Extensions of Modern Error Feedback' by Richtarik et. al;
(c) `Communication-Efficient and Byzantine-Robust Distributed Learning With Error Feedback  ' by Ghosh et.al;

address similar distributed/Federated aspects with error feedback. These papers should be discussed and the results should be compared with the ones in these papers. I believe  `EF21 with Bells & Whistles: Practical Algorithmic Extensions of Modern Error Feedback' is the closest one, and a thorough comparison is required.

**Overclaim:** The paper seems to overclaim on certain sections. Ex: `partial-participation has never been considered'--this is not correct.  `EF21 with Bells & Whistles: Practical Algorithmic Extensions of Modern Error Feedback' considers partial client participation. Moreover, the paper claims ` One of the very few, if not only, FL algorithms that uses biased compression with EF is QSparse-local-SGD'--the paper   `Communication-Efficient and Byzantine-Robust Distributed Learning With Error Feedback  ' considers the biased compression setting with error feedback as well.

**Novelty:** The technical analysis, setup has a lot of overlap with `Achieving Linear Speedup with Partial Worker Participation in Non-IID Federated Learning' by Yang et. al, including the choice of step sizes. The proofs use the same arguments as the above mentioned paper, with the additional error feedback term. Please comment on this aspect and address the technical novelty of the work.

**Choice of Step Sizes:** The choice of step size is quite non-standard (they are the same as in  `Achieving Linear Speedup with Partial Worker Participation in Non-IID Federated Learning' by Yang et. al). The local step sizes are very very small (overall with $K$ local iterations, the local iterates do not seem to move/drift enough it seems. On the other hand the global step size is very very big ($\sqrt{K n}$). Please provide some intuitive explanations, on this choice.

**Typo:** I believe the first term in the right hand side of Theorem 1 has a $1/T$ missing; otherwise, the algorithm would diverge given the step size choices.

**Summary Of The Paper:**

The paper analyzes the FL setup with error feedback for non convex objective. It analyzes two variants of the FED-EF algorithm, one with SGD and the other one with AMS. Theoretical analysis is done in both setting, in particular it is shown that a linear speedup is possible, i.e., the error rate goes down with $1/\sqrt{n}$ when $T\geq K n$, where $K$ is the number of local steps and $n$ is the number of machines. The partial participation setting is also analyzed. Experiments are done on MNIST, CIFAR and FMNIST.

**Summary Of The Review:**

Please see the Strengths/Weaknesses section. The paper misses comparison with several existing works and I also have some technical novelty concerns.

---

> ### Author Response · Authors · 2022-11-19
> **Thanks for your review: Part 1**
>
> Dear Reviewer GNNs:
>
> Thanks for your valuable comments on our paper.
>
> **Concern 1: further comparison with recent papers and claims on our results**
>
> Since the topic of our paper covers distributed optimization, federated learning and communication compression, there are extensive related papers in literature and we have tried our best to cite the most relevant ones to our knowledge. Thanks for referring us to two more relevant papers.
>
> [1*] EF21: A New, Simpler, Theoretically Better, and Practically Faster Error Feedback, by Richtarik et. al, NeurIPS 2021.
>
> We have cited [1*] in our original submission. This paper is substantially different from our work: 1) The reference paper proposes a new variant of EF which is different in algorithm design, while we study the convergence of the standard EF when applied to practical FL problems; 2) We consider the federated setting with local steps, data heterogeneity and partial participation, while [1*] studies SGD in the classical distributed learning setting without considering any of these; 3) [1*] analyzed convex loss functions while we study non-convex optimization.
>
> [2*] EF21 with Bells & Whistles: Practical Algorithmic Extensions of Modern Error Feedback, by Richtarik et. al, unpublished.
>
> Thanks for mentioning this manuscript. As we mentioned above, the EF21 method is a different algorithm from the EF studied in our paper, i.e., [Local SGD converges fast and communicates little, Stich, 2019]. Thus, the algorithm design and techniques in theoretical analysis in [2*] are different from ours. In addition, our paper also considers adaptive optimization algorithms besides SGD. Theoretically, our method achieves linear speedup by using two-sided learning rates while the reference paper does not. Therefore, our paper and [1*] are different algorithms with different analysis and results. Particularly, our result on the convergence of standard EF under partial participated FL is independent and new in literature. We have cited [1*] in our revision and added the comments.
>
> [3*] Communication-Efficient and Byzantine-Robust Distributed Learning With Error Feedback, by Ghosh et.al, IEEE SELECTED AREAS IN INFORMATION THEORY 2021.
>
> Thanks also for mentioning this paper. [3*] considers applying EF to compressed distributed training with potential Byzantine workers. Similar to [1*], [3*] falls in the category of classical distribution optimization and does not consider local steps, data heterogeneity and partial participation, which is significantly different from our work. Yet, it is an interesting application of EF in distributed optimization. Thanks for pointing us to the paper, we have added [3*] as a related work on error feedback in the revision.
>
>
> *"[3\*] considers the biased compression setting with error feedback as well."*
>
> As discussed earlier, [3*] did not study EF in federated learning setting. To our knowledge, the only paper which studied standard error feedback strategy in FL is the QSparse-local-SGD [6] which we cited and compared with. In fact, QSparse-local-SGD only combined local steps, but did not consider non-iid data and partial participation. Thus, our work provides improved, more thorough and practical results regarding EF in FL.

---

> > ### Author Response · Authors · 2022-11-19
> > **Thanks for your review: Part 2**
> >
> > **Concern 2: technical novelty and challenges**
> >
> > We would like to note that our non-convex optimization problem setup is standard in FL literature. While we applied the two-sided learning rate schedule same as in, e.g., [57], we think this is an useful algorithmic trick to attain fast convergence rate in distributed optimization, which has also been used in some other recent works. Besides this point (two-side learning rate) in common, we would like to highlight the technical novelties in our analysis compared with prior work.
> >
> > (1) In the Fed-EF-AMS variant, it requires much more effort to design a workable virtual sequence and properly handle the moving average of the first and second moments. We tackle this by constructing an adaptively accumulated error tracker, which introduces additional error term that needs to be bounded. This analysis is novel (compression has not been considered for adaptive FL) and very different from the local SGD analysis.
> >
> > (2) In the partial participation setting, our analysis is more challenging because of the delayed error compensation effect. The analysis requires carefully bounding the error accumulator sequence under partial participation and choosing the stepsizes to factor out the extra $\sqrt{n/m}$ term, which is a new analytical approach that has not been proposed in classical error feedback literature.
> >
> >
> > **Concern 3: intuition on the learning rates**
> >
> > Thanks for pointing out the typo, there should be a $T$ in the denominator of the first term in Theorem 1. The intuition of the decreasing local learning rate is as follows. Firstly, in the setting where the data distribution are highly non-iid, the local models contain "biases" towards minimizing local losses which might be very different from the global loss. In other words, the local client drifts might be in "bad" directions for minimizing the global loss function, in which case too many local steps $K$ would not help the convergence globally (i.e., the "bad" local models may go too far). Therefore, we set the local learning rate to decrease in $K$ to mitigate this effect. Then since the local stepsize is controlled, we can pick a larger global learning rate to make the global model move far enough (recall that the global loss is our final objective). We think this is a good intuition why two-sided learning rates may help in heterogeneous FL.
> >
> > We elaborate by some additional quantitative analysis. In Corollary 1, for example, considering the impact of $K$, we can see that the rate becomes the desired $O(\frac{1}{\sqrt{TKn}})$, which decreases in $K$ (i.e., local steps helps), if $K=O(T\sigma^2/\sigma_g^2)$. That said, when $\sigma_g^2$ is large (high heterogeneity), the asymptotic convergence rate can be $O(\frac{1}{\sqrt{TKn}})$ (i.e., local steps help), when $K$ is relatively small. On the contrary, for small $\sigma_g^2$, we can tolerate larger $K$ to achieve the desired convergence rate.
> >
> > Again, we deeply appreciate your feedback. We hope that our reply and updated paper are able to address your concerns adequately.

---

### Official Review · Reviewer_FJCK · 2022-10-24

**Confidence:** 2
**Correctness:** 4
**Technical Novelty And Significance:** 3
**Empirical Novelty And Significance:** 3
**Recommendation:** 6

**Clarity, Quality, Novelty And Reproducibility:**

The paper is pleasant to read. The ideas in the paper are not new but the application to the setting of Federated learning is novel. Sufficient details of the experiments conducted are included in the paper.

**Strength And Weaknesses:**

- The paper provides the first convergence analysis of compressed adaptive. The paper itself is well-organized.

 - The is limited to an analysis for convergence to a neighborhood around a stationary point and not a global minimizer (or local).

- In page 2: "we present the general compressed FL framework named Fed-EF" seems to suggest the algorithm does not use Error Feedback. Please consider modifying the phrase appropriately. Also, it is not clear what MLP stand for in page 2?
 - How is the stochastic gradient term $g_{i,i}^{(k)}$ introduced in Assumption 2 used? It unclear from Assumptions 1-3 and Algorithm 1.
 - From Figure 2, its not clear if methods with Error Feedback performs better than methods without. In most case (the figure is not big enough to be able to make a good comparison), method with EF and without EF seems to perform on par and the claim in page 7 "test accuracy of Stoc is slightly lower than Fed-EF with hv-Sign" seems selective to that instance of the experiment. It is not clear from the figures shown if the behaviour shown holds on average or is from one instance of the experiment.

**Summary Of The Paper:**

In the paper, the authors consider the federated learning problem. The authors provide an algorithm that compresses the gradient before sending it from client to server (to combat communication cost), accounting for error of compression from previous iteration. Two versions of the algorithm is presented: Fed-EF-SGD and a momentum based FED-EF-AMS.  Theoretical convergence analysis on the convergence is provided (under reasonable conditions) and performance is shown to be on par with uncompressed full Federated Learning algorithm. An analysis on convergence rate under partial participation from the clients is also provided. Some numerical experiments are also provided and validate their theory.

**Summary Of The Review:**

I believe the paper address an open problem in the setting of Federated Learning and provides compelling theorical and numerical evidence to support their claims.

---

> ### Author Response · Authors · 2022-11-19
> **Thanks for your review.**
>
> Dear Reviewer FJCK:
>
> Thank you for your positive feedback and support. We sincerely appreciate your comment that our paper "address an open problem in the setting of Federated Learning and provides compelling theoretical and numerical evidence".
>
> We reply each point under "Strength And Weaknesses:"
>
> **Point 1: The paper provides the first convergence analysis of compressed adaptive. The paper itself is well-organized.**
>
> Thank you.
>
> **Point 2**: We follow the standard in non-convex optimization literature to study the convergence to a stationary point. For general non-convex objective, though highly challenging, there are many active efforts to finding the global minima under various conditions and assumptions. We agree that it would be an interesting direction in the future.
>
> **Point 3**: MLP = Multi-Layer Perceptron. We agree that we should always explain any acronym the first place it is used. Thank you.
>
> **Point 4**: $g_{t,i}^{(k)}$ appears in line 7 of Algorithm 1, which represents the stochastic gradient computed for the $i$-th client in $t$-round, at the $k$-th local iteration.
>
> **Point 5**: In our submission, in order to include more results in the paper, we had to choose to make each plot smaller. In the updated paper, we have included Table 1 - Table 4 with accurate numbers to see the comparison more clearly. We have also polished the sentence in the paper. Our claim is that Fed-EF performs in general comparable with Stoc without EF, sometimes better. Our major focus is not to prove Fed-EF is always better than Stoc (the performances of these methods are typically data and task dependent). Instead, we hope to illustrate that Fed-EF is able to match the performance of full-precision training, with considerably less communication, which is validated by our experiments.
>
> Again, thanks for your encouraging feedback and support. We hope our response answers your questions.

---

### Official Review · Reviewer_Xm1Y · 2022-10-28

**Confidence:** 3
**Correctness:** 3
**Technical Novelty And Significance:** 3
**Empirical Novelty And Significance:** 3
**Recommendation:** 5

**Clarity, Quality, Novelty And Reproducibility:**

Some parts were a bit confusing to me.
Why is $\Delta$ called "the effective gradient"? As far as I can see, it's not a gradient of any function

### Minor issues
On page 2, "which are biased of the true gradients", do you mean "biased compressors"?
On page 5, "the above two terms stays close" => "stay close"
I think it would be good to add the definition of the "Stoc" compressor in the paper


**Strength And Weaknesses:**

## Strengths
1. The work addresses practical issues arising in federated learning: communication efficiency and partial participation of clients, and the authors make an advance in the considered direction.
2. The rates are good in terms of the asymptotic dependence on the number of iterations.
3. The theory is supported by experiments.

## Weaknesses
1. The paper makes some misleading claims about the novelty of the method. In particular, the claim that "Partial participation (PP) has never been considered for error feedback in distributed learning." is not correct, see (Fatkhulin et al., 2021).
2. The authors list as one of their contributions proposing "a unified compressed FL framework". As far as I can see, the unified framework has just two options (SGD and AMS) and calling it "unified" seems to be an overstatement.
3. As far as I can see, Theorem 1 suggests that there is no benefit from the local steps. In particular, if $\eta\eta_l=\mathrm{const}$, then the bound improves as $\eta_l\to 0$, which means that the local steps help only due to sampling more gradients. The same can be said about Theorem 2.
4. All experiments seem to use a single random seed and no confidence intervals are reported.
5. The experiments only present the number of communicated bits as the efficiency metric, which ignores the computation overhead of compressing the updates as well as it does not take into account if it is actually easier to perform communication of the compressed vectors. For instance, TopK does not support synchronization primitives (over batches of clients) such as all-reduce. Moreover, it is not clear if compressing the updates make so much sense since we already ask the clients to perform a lot of local updates, which reduces the communication bottleneck.
6. Clients are required to maintain a state. When partial participation is considered, it is reported by Kairouz et al. "Advances and Open Problems in Federated Learning" that one might not see the same client twice, so stateful methods make less sense with PP.

Ilyas Fatkhullin, Igor Sokolov, Eduard Gorbunov, Zhize Li, Peter Richtárik, 2021, "EF21 with Bells & Whistles: Practical Algorithmic Extensions of Modern Error Feedback"

**Summary Of The Paper:**

## Update
I have read the authors' feedback and I remain confident this paper is not ready for publication for these reasons:
1. The authors argued that their work is different from that of Fatkhulin et al. in that it uses the standard EF. I do not see, however, why standard EF would be better than EF21, I don't think that this difference justifies novelty.
2. The authors also argued that they consider Adam stepsizes on top of that. The theory for Adam stepsizes is, however, underwhelming due to dependence on the dimension, which is caused by the fact that even for Adam itself we do not have a good analysis.
3. The authors argue that $\eta\eta_l$ has to approach 0 as we target higher accuracy, which I agree with. However, once we set the value of $\eta\eta_l$, it is always beneficial to decrease $\eta_l$ and increase $\eta$ while preserving the value of $\eta\eta_l$. This implies that the local steps are not shown to be useful, even if the authors show that local steps are helpful under other (suboptimal) choices of $\eta$ and $\eta_l$.
4. Please note that I **was not** "raising the concern regarding the general research direction of communication compression in FL.". In contrast, I was arguing that the compression used in the experiments might not work well with the communication primitives such as all-reduce. In other words, my comments were toward the experiments and not the research direction.

#######
This paper studies the combination of error feedback for compressed optimization with local method for federated learning. On top of that, the authors consider two potential ways of updating the global model based on local updates: with standard averaging and with Adam-based model update. The main contribution of the paper is to propose Algorithm 1 that combines all these features and to study its convergence with full and partial participation. The theory is supported by numerical experiments that show how different variants of Algorithm 1 based on different compressions and stepsizes perform on training neural networks.

**Summary Of The Review:**

The considered setting might not be very meaningful due to the need for clients to maintain a state, while partial participation is usually employed when the number of clients is large and it is uncommon for them to keep a state. The theory has some small issues (no benefit from local steps). The experiments show only one side of the method's performance–number of communicated bits, which is not the only aspect of time efficiency–and do not illustrate if there might be an actual saving of time.

All in all, this paper will probably have some audience, but the results are not impressive.

---

> ### Author Response · Authors · 2022-11-19
> **Thanks for your review: Part 1**
>
> Dear Reviewer Xm1Y:
>
> Thanks for your feedback on our paper and commenting that we "make an advance in the considered direction". We reply the 6 questions you raised as "weakness".
>
> **Question 1: further comparison with a related paper**
>
> Our paper covers several topics: distributed optimization, federated learning and communication compression. For each topic there are a huge number of papers in the literature. Our original submission cited the original "EF21" paper, and we thank the Referee for suggesting us to compare with
>
> [1*] "EF21 with Bells \& Whistles: Practical Algorithmic Extensions of Modern Error Feedback 2022".
>
> It is known that EF21 has a different algorithm design, not the standard EF strategy [Local SGD converges fast and communicates little, Stich, 2019]. Our submission adopted the standard EF strategy. In other words, our paper studied different algorithms from EF21 (and [1*]), with different analysis techniques and results.
>
> Note that our paper also considered the adaptive federated optimization, and our convergence rate achieves linear speedup by using two-sided learning rates while [1*] does not achieve the linear rate. Our result on the convergence of standard EF under partial participated FL is new in the literature. Both strategies (EF and EF21) are unique contributions.
>
> Again, we thank the Referee for suggesting the reference and we have cited [1*] in the revision and commented on their differences.
>
> **Question 2: a wording issue.**
>
> By "unified" we were trying to say that it combines both SGD-type and Adam-type optimizer, and sure there are more possible variants. We have adjusted the wording as suggested.
>
> **Question 3: We repeat the original question here: "there is no benefit from the local steps. when In particular, if $\eta\eta_l=const$, then the bound improves as $\eta_l\rightarrow 0$, which means that the local steps help only due to sampling more gradients."**
>
> If we set "$\eta\eta_l=const$", then there will be a constant variance term in the convergence rate. Thus, in order for the rate to converge as $T\rightarrow\infty$, $\eta\eta_l$ needs to be decreasing (instead of being a constant). Please let us know if we miss anything here. Thanks.
>
> As we discussed in the paper, under proper conditions, the convergence rates becomes $O(\frac{1}{\sqrt{TKn}})$ which improves with larger $K$, implying that local steps can be helpful to the convergence.
>
> Here we would like to add an additional remark on the condition under which larger $K$ would help. In Corollary 1, for example, we can see that the rate becomes $O(\frac{1}{\sqrt{TKn}})$ (i.e., improving with $K$) when $K=O(T\sigma^2/\sigma_g^2)$. In other words, when the data heterogeneity is high ($\sigma_g^2$ is large), our rate improves with $K$ when $K$ is relatively small. With small $\sigma_g^2$, $K$ can be very large to achieve the desired rate. This is intuitive, since in highly non-iid setting, more local steps may not help the global model to converge faster. This is because in this setting, each local model is actually trained with a "bias" towards minimizing the (possibly very different) local loss. Too many local training would make the local model move too far towards that "biased solution", in which case we will need a smaller local learning rate for mitigation. On the other hand, in more homogeneous case (small $\sigma^2$), we can tolerate much larger $K$ to reduce communication and still get good performance. We hope this remark provides the intuition on the impact of $K$ on the convergence results.
>
> **Question 4: error bars**
>
> In our paper, the curves are averaged over repeated runs. For example, in the original submission, Figure 4 caption mentioned "... averaged over 20 independent runs.". In the revision, we added 4 additional tables (Table 1 to Table 4 in the Appendix) to show the test accuracy along with the standard deviations. Thank you for the suggestion.

---

> > ### Author Response · Authors · 2022-11-19
> > **Thanks for your review: Part 2**
> >
> >
> > **Question 5: motivation and benefit of communication compression in FL in general**
> >
> > Thank you for raising the concern regarding the general research direction of communication compression in FL. While there are numerous papers in this line of research, we certainly agree that ultimately FL needs to be implemented and deployed in real systems. Our paper did not intend to solve the entire problem (including systems and deployment) of FL with communication compression.
> >
> > There are many types of FL systems for many different application scenarios. In wireless edge computing, communication compression appears to be crucial; see for example, "Federated learning for internet of things: Recent advances, taxonomy, and open challenges, Khan et al., 2021" (which we cited in the original submission). Let us quote the following statement from this article:
> >
> > *"we derived that federated learning must use some  effective compression technique due to limited communication resources and a massive number of IoT devices."*
> >
> > On the other hand, we would not be surprised that for some other systems, communication compression might not necessarily be the bottleneck. From the research perspective, communication compression is an very interesting  problem and has recently attracted a lot of attentions in the literature. For us, this is good motivation to invest on this research.
> >
> >
> > **Question 6: Partial participation and stateless system**
> >
> > Thank you for mentioning the survey "Advances and Open Problems in Federated Learning" which mentioned stateless system and described two extreme scenarios:
> >
> > *"In cross-device FL, algorithms generally cannot assume any state is preserved on the clients (Table 1). However, this constraint would typically not be present in the cross-silo FL setting, where the same clients participate repeatedly. Consequently, a wider set of ideas related to error-correction such as [311, 405, 463, 444, 263, 435] are relevant in this setting ..."*
> >
> > In other words, there are systems that always have the states and there are stateless systems. And of course there are systems in between, i.e., partial participation with more stable and frequent clients. Our research does not apply to stateless systems but there are many other FL systems where states can be useful.
> >
> > BTW, this survey also mentioned *"It is now well-understood that communication can be a primary bottleneck for federated learning since wireless links and other end-user internet connections typically operate at lower rates than intra- or inter-data center links and can be potentially expensive and unreliable"* in Section 3.5: Communication and Compression. We hope this also partially answers your question regarding the benefit and necessity of communication compression in FL.
> >
> > **Other questions**:
> >
> > We called $\triangle$ "effective gradient" since it acts like the gradient in standard distributed SGD, which is used by the global optimizer for a gradient-like update. For clarity, we have changed it to "local model update".
> >
> > Thanks also for the suggestion of introducing "Stoc" in the main paper. In the original submission, we placed this section in Appendix A.2 due to lack of space. We are happy to include it in the main paper when more space will be allowed.
> >
> > Again, we sincerely appreciate your feedback. We hope our rebuttal and paper update are able to address your concerns.

---

### Official Review · Reviewer_SCXf · 2022-11-04

**Confidence:** 4
**Correctness:** 3
**Technical Novelty And Significance:** 2
**Empirical Novelty And Significance:** 2
**Recommendation:** 3

**Clarity, Quality, Novelty And Reproducibility:**

#### Clarity:
Overall the paper is well written; However, I feel the figures should be self explanatory from the figure title itself. Often the titles are missing / inadequate weakening readability.

#### Novelty:
I think novelty is limited (see Weakness and Summary of review)

#### Reproducibility:
Needs more clarification and experimental details. (see Weakness)

**Strength And Weaknesses:**

### Strengths:
1. The paper is well motivated and written.

2. The main contribution of the paper is the analysis of SGD and AMSGrad in the Heterogenous Federated setting ( heterogenous client data distribution +  local SGD, partial client participation ) + ( communication compression + EF ).

3. The proofs are clear and rigorous and easy to follow.


### Weaknesses / Clarifications :
1. Definition 1. (qC - deviate compressor) It simply says the compression operator yields a contraction mapping. This is a standard assumption used in most theoretical works in communication compression and error feedback literature: ex see [1-2].  Mentioning this connection with proper citation around Def 1 in the main paper (as opposed to making this connection in Appendix. A.1) will improve clarity,

2. Algorithm 1. is a straightforward ( almost trivial ) extension of distributed SGD with EF to Federated Learning setting;  so overall I feel algorithmic novelty is quite limited.

3. In Fig 18-Title missing bracket inside the expectation - i guess it should be $ E \|\| C(x) - x \|\|^2 \leq q_c^2\|\|x\|\|^2 $ where $ x = \frac{1}{n}\sum_{i=1}^n(\Delta_{t, i} +e_{t,i}) $. What do the subfigures correspond to - not clearly specified : I am *assuming* left one is for normal and right for laplacian data.

4. Let $\epsilon^2 = || C(x) - x ||^2$ Then I believe the points on Fig 18 denote $\frac{\epsilon^2}{||x||^2}$ - However, it is not clear from the fig or B.1 on how C was used : e.g. Top-k what is k values ? In theory the bound trivially holds for $q_c = (1 - \frac{k}{d})$ but empirically how does it correlate on the simulated data. In other words, additional analysis wrt k vs q_c over different heterogeneity (scale) would be more insightful.

5. What is the significant of the offset between Definition 1 and assumption 1 -- as seen in Fig 18.

6. Definition 1 has clear motivation , I am not clear on what is the intuition / how to interpret Assumption 1.  I in fact think that Assumption 1 is unrealistic and Fig 18 shows how in non-iid scenarios it is completely off from bounds derived from Definition 1 ( i.e. from reality ) - Could you please justify this ?

7.  Empirical Evaluation: We know in non-fed EF-SGD error feedback improves the results tremendously - and can give nearly same rate of convergence as full-SGD while using only ~10 % communication. The results in the fed setting should have identical trends - as confirmed by the experiments (not surprising - right ? )

8. How was the communicated bits calculated in Fig 2, 3 etc ? Let's say for Top k one would use different k values to compute log comm bits -- it is not clearly mentioned how these plots are generated. Also, are these plots using EF ? It is not clearly mentioned and hard to read the figures - please update the captions to be self contained.

9. What is the exact mechanism to simulate non-iid clients ? How did you control amount of heterogeneity in experiments ?

10.  Further, how does heterogeneity impact the results especially since we see in Fig 18 with high heterogeneity Assumption 1 has larger offset than Definition 1 ( i.e. from reality ) ?

11. Exp are small scale : done on MNIST, FMNIST. One slightly more challenging task (at least CIFAR 10 / CIFAR 100 /mini-imageNet ) might help with validating the claims - ex.

### References:
1. https://papers.nips.cc/paper/2018/file/b440509a0106086a67bc2ea9df0a1dab-Paper.pdf
2. https://proceedings.mlr.press/v97/karimireddy19a/karimireddy19a.pdf

**Summary Of The Paper:**

### Background

In the distributed optimization paradigm, it is common for the client nodes to communicate only compressed version of the computed local gradients (or parameters) with the server node to reduce communication latency. However, since compression schemes are often lossy (in expectation) and biased (ex Top K) , working with the compressed gradients might result in slower overall convergence. To alleviate this, it is common practice to use error compensation where the error by propagating compressed gradient ||g - C(g)|| is accumulated and used as feedback in subsequent iterations. Error Feedback ensures that the SGD converges at the same rate as the non-compressed counterpart.


### Contribution:

In this paper, EF is applied and analyzed in the federated setting as follows: The server broadcasts the global model; each client (each client maintains an error vector e initialized to 0 ) does k local SGD steps; computes the error compensated gradient g = (gradient + e); computes the C(g); computes the error from compression; updates e by accumulating this additional error.  For the analysis, the authors also considered potentially heterogenous data distribution across clients ; partial client participation, adaptive gradient method at the global server.

**Summary Of The Review:**

Overall, The contribution is quite incremental I feel. Firstly, the proposed algorithm is straightforward extension of distributed SGD with communication compression and error feedback [1,2]. Secondly, The main contribution is the analysis - which also I think is somewhat incremental overall given the tools to analyze already exist and the proof technique is also similar.  Thirdly, empirical evaluation is small scale overall. Fourthly, Assumption 1 is not clear - and seems unrealistic.

---

> ### Author Response · Authors · 2022-11-19
> **Thank you for the review: Part 1**
>
> Dear Reviewer SCXf:
>
> Thank you. We appreciate your positive comments on (e.g.,) the motivation, technical rigor and presentation.  It appears that the main concerns are:
>
>
> **Main concern (1)**: The method is simple.
>
> We actually view this as a compliment because simple method is more likely practical. Starting with this simple method, we are able to achieve fast convergence and obtain new analysis and results under partial participation (which are also main theoretical contributions of this work). We use new techniques to construct an appropriate virtual sequence and bound the additional error sequence in the adaptive optimization algorithm, and to bound the error accumulators under partial participation and factor out the extra slow down factor caused by stale error compensation. These analysis and results are new in EF literature. In addition, we are able to obtain a new algorithm for compressed adaptive federated optimization.
>
> **Main concern (2)**: Need more experiments, e.g., on CIFAR. (we had it in the Appendix).
>
> The experiments on CIFAR are available in Appendix of the original submission. We mentioned the additional experiments in the main paper. The conclusion from CIFAR dataset are consistent with the results in the main paper that Fed-EF is able to match the performance of full-precision training. Assuming that camera-ready may allow additional page (like NeurIPS), we might be able to include the additional experiments in the main paper as opposed to the Appendix. Thank you.
>
> **Main concern (3)**: Clarification questions about the captions of Figure 2 \& 3 and Figure 18 (in the appendix).
>
> We thank you for also reading the details into the Appendix. We agree that in general it is always better to make the caption of the figures as detailed as possible. We have revised the captions of these three figures. Thank you. BTW, we don't quite understand the question about "missing brackets". It looks the original expression is correct. Or do we miss anything?
>
>
> **Main concern (4)**: justification of Assumption 3.
>
> We first would like to remark that such analytical assumption on the compression discrepancy is reasonable and has been used in recent prior works (e.g., [4, 16]) to study compressed distributed optimization methods. We have justified this assumption empirically through simulation (Figure 18) and real-world experiments (Figure 19).
>
>
> **Main concern (5)**: two references in the Appendix should be in the main paper.
>
> Thank you for the suggestion.

---

> > ### Author Response · Authors · 2022-11-19
> > **Thanks for your review: Part 2**
> >
> > **Additional questions about the experiments etc.**
> >
> > *1. More explanation of Figure 18:* Yes, the left is for Gaussian and the right is for Laplace. For TopK, we use top $k=0.1$ coordinates. The points in the figure are the empirical value of $\frac{||C(x)-x||^2}{||x||^2}$. Regarding the relationship between $k$ with $q_C^2$, note that $q_C^2$ is only an upper bound. For TopK, for example, $||C(x)-x||^2=(1-k)||x||^2$ only when all the coordinates have the same magnitude, which is unlikely in practical deep learning. In general, $q_C$ would decrease with larger heavy hitters in the vector. This explains the decreasing trend of TopK as the scale increases in Figure 18.
> >
> > The relationship between Definition 1 and Assumption 3 should be: "we consider compressors satisfying Definition 1, and with these compressors, we additionally assume Assumption 3 holds". As we mentioned on page 5, Assumption 3 intuitively says that "the average of compression'', $\frac{1}{n}\sum_{i=1}^n C\big(\triangle_{t,i}+e_{t,i}\big)$, should be close to "the compression of average'', $C\big(\frac{1}{n}\sum_{i=1}^n (\triangle_{t,i}+e_{t,i})\big)$, during training. In our analysis, the rates include $q=\max(q_C,q_A)$, so the difference between $q_C$ and $q_A$ does not affect the results, as long as $q_A<1$, which is justified numerically through simulation and experiments.
> >
> > *"Fig 18 shows how in non-iid scenarios it is completely off from bounds derived from Definition 1 (i.e. from reality)"*
> >
> > We hope it is more clear now that $q_C$ should not be considered as the "reality" of $q_A$, since they are defined on different quantities in parallel. The solid curves in Figure 18 are the empirical $q_A$ which is the "reality". In Figure 18, we see that $q_A$ is smaller than 1 with various scales which justifies the assumption. In Figure 19, we provide more empirical evidence for training on real-data.
> >
> > *2. New empirical results compared with prior work:* Our empirical evaluation presents new results compared with classical EF-SGD in many aspects. Firstly, we show that EF could perform well in FL under *non-iid data and partial participation* with significant (~100x) compression. These conditions are rarely considered in prior work on EF-SGD. Second, we demonstrate that EF can also be effective for adaptive federated learning algorithm with biased compression (Fed-EF-AMS). Third, we numerically demonstrate the impact of stale error accumulation on the convergence of Fed-EF and justify the theory. All these empirical observations have not been reported in classical EF-SGD literature.
> >
> > *3. Curves presented in Figure 2 \& 3.* In our original main paper, we described at the beginning of Section 5.1 how the compression parameters ($k$ and $b$) are chosen for each compressor in Figure 2 \& 3. For each method, we present the curve with highest compression level that achieves (within $0.1$%) the best full-precision test accuracy; if
> > the method does not match the full-precision performance, we present the curve with the highest test accuracy. We combine the curves of different methods in this way for more straightforward comparison. The full results of each compressor with various compression parameters (ratios) were provided in appendix Section A.4, Figure 8 - Figure 15, in the initial submission.
> >
> > *4. Client heterogeneity:* For $n=200$ clients, we first split the training samples into $2n=400$ shards, where each shard contains samples from only one class. Then, each client is randomly assigned with two shards uniformly. This way, in expectation, about 180 clients would have samples from two classes, and about 20 clients would have samples from only one class. We used this heterogeneous data split in all our experiments, to focus on the important heterogeneous FL setting. We have added this description to the Appendix for clarity. Thank you.
> >
> > The impact on data heterogeneity can be seen from our theoretical results (Theorem 1, 2, 3). As you mentioned, higher heterogeneity leads to larger $q_A$ which makes the convergence bound larger. It also results in larger global variance term $\sigma_g^2$ which would slower down the convergence as well. Empirically, our current setup is already very heterogeneous which reflects the main challenge in practical FL. If the reviewer is interested in the comparisons under less challenging settings (e.g., iid data), we can also show the results under this setting, which basically gives the same conclusions as our current set of highly non-iid experiments. Thank you.
> >
> > Again, we appreciate your positive feedback on our motivation, technical quality and presentation. We feel that many of your questions are about clarifying the figures and experiment settings, etc. We hope that our response and the paper revision address your concerns and clarify some possible misunderstandings of our work, which would be helpful for the reviewer to re-evaluate the contributions of our submission. Thank you very much.

---

### Author Response · Authors · 2022-12-10
**Please provide feedbacks on our rebuttal, thank you**

Dear Reviewers and Area Chair,

Again, we sincerely appreciate your feedback and efforts in reviewing our paper. We have followed Reviewers' suggestions and updated our paper with improved presentation, some more polished wording and clarifications, and more numerical results (Tables 1-4).

We believe our rebuttal and revision have addressed the two main questions asked by the reviewers:

(1) novelty and significance of the new results and analysis;

(2) connection to a recent arxiv paper on another method called "EF21".

Regarding the first question, as we have replied in the rebuttal:

(i) our work is the first to show that error feedback with biased compression can also achieve the fast convergence rate of the non-compressed FL algorithms;

(ii) Fed-EF-AMS is the first compressed adaptive FL algorithm;

(iii) the analysis and result on partial participation with Fed-EF is new in the EF literature.

Moreover, our analysis involves new approaches to formulate the error accumulator in adaptive FL and to extract the extra slow down factor under the partial participation setting. We hope these new algorithms, results and techniques would be beneficial to the FL community.

Regarding the second question, as we explained in our response to Reviewer Xm1Y and Reviewer GNNs, "EF21" is a different algorithm from the standard EF considered in our paper. Consequently, the algorithm and analysis are essentially different. Also, the "EF21" paper only considered SGD as the optimizer and the convergence rate does not achieve linear speedup, while we also proposed Fed-EF-AMS built upon adaptive optimizer, and we provided the linear speedup analysis.

In addition, Reviewer Xm1Y asked about whether the communication cost is indeed the bottleneck of federated learning. One of the important applications of FL is the edge computing, e.g., training machine learning models on mobile devices (or even more aggressively on satellites). In the related literature, communication compression has been recognized as one important part for the success of implementing these FL training systems in practice. We have cited several related papers in our submission, and we are happy to add more references to illustrate the significance of this topic. Thank you.


We hope that our response has addressed your concerns adequately. If there are further questions that need us to explain, please kindly let us know.  Thank you.

---

### Decision · Program_Chairs · 2023-01-20

**Decision:**

Reject

**Justification For Why Not Higher Score:**

The lack of novelty - either in the algorithm or in its analysis - is the primary hinderance to a higher score

**Justification For Why Not Lower Score:**

N/A

**Metareview: Summary, Strengths And Weaknesses:**

This paper studies federated learning in the setting where clients provide compressed error feedback to the server. The setting is standard, and the reviewers are uniform in their assessment that the algorithm in this paper is not particular novel.

If fact the lack of significant novelty seems to be a concurrent assessment of all the reviewers, so that even though the paper is otherwise clearly written and correct, the scores are (on aggregate) the lowest in my collection of papers.

For this reason I would have to recommend rejection from this conference. The reviews are exhaustive and provide a clear roadmap to an improved version for another venue.

**Summary Of Ac-Reviewer Meeting:**

not borderline